# Temporal-dependent effects of rainfall characteristics on inter-/intra-event branch-scale stemflow variability in two xerophytic shrubs

**Chuan Yuan[1, 2, 4], Guangyao Gao[2,3], Bojie Fu[2,3], Daming He[1, 4], Xingwu Duan[1, 4], and Xiaohua Wei[5]**

[1]Institute of International Rivers and Eco–security, Yunnan University, Kunming 650091, China

[2]State Key Laboratory of Urban and Regional Ecology, Research Center for Eco-Environmental Sciences, Chinese Academy of Sciences, Beijing 100085, China

[3]University of Chinese Academy of Sciences, Beijing 100049, China

[4]Yunnan Key Laboratory of International Rivers and Trans-boundary Eco–security, Kunming 650091, China

[5]Department of Earth, Environmental and Geographic Sciences, University of British Columbia (Okanagan campus), Kelowna, British Columbia, V1V 1V7, Canada

**Correspondence:** Guangyao Gao (gygao@rcees.ac.cn)

**Abstract**

Stemflow is important for recharging root-zone soil moisture in arid regions. Previous studies have generally focused on stemflow volume, efficiency and influential factors but have failed to depict stemflow processes and quantify their relations with rainfall characteristics within events, particularly for xerophytic shrubs. Here, we measured the

stemflow volume, intensity, funnelling ratio, and time lags to rain at two dominant shrub
species (*Caragana korshinskii* and *Salix psammophila*) and rainfall characteristics during
events at the semi-arid Liudaogou catchment of the Loess Plateau, China, during the
2014–2015 rainy seasons. Funnelling ratio was calculated as the ratio between stemflow
and rainfall intensities at the inter-/intra-event scales. Our results indicated that the
stemflow of *C. korshinskii* and *S. psammophila* were averagely started 66.2 and 54.8 min,
maximized 109.4 and 120.5 min after rains began, and ended 20.0 and 13.5 min after rains
ceased. The two shrubs had shorter stemflow duration (3.8 and 3.4 h) and significantly
larger stemflow intensities (517.5 and 367.3 mm·h$^{-1}$) than those of rains (4.7 h and 4.5
mm·h$^{-1}$). As branch size increased, both species shared the decreasing funnelling ratios
(97.7–163.7 and 44.2–212.0) and stemflow intensities (333.8–716.2 mm·h$^{-1}$ and 197.2–
738.7 mm·h$^{-1}$). Tested by the multiple correspondence analysis and stepwise regression,
rainfall amount and duration controlled stemflow volume and duration, respectively, at
event scale by linear relations ($p<0.01$). Rainfall intensity and raindrop momentum
controlled stemflow intensity and time lags to rains for both species within event by linear
or power relationships ($p<0.01$). Rainfall intensity was the key factor affecting stemflow
process of *C. korshinskii*, whereas raindrop momentum had the greatest influence on
stemflow process of *S. psammophila*. Therefore, rainfall characteristics had
temporal-dependent influences on corresponding stemflow variables, and the influence also
depended on specific species.
**1 Introduction**

47        Stemflow directs the intercepted rains from canopy to the trunk base. The

funnel-shaped canopy and underground preferential paths, i.e., roots, worm paths and soil
macropores, converge rains to recharge the root-zone moisture (Johnson and Lehmann,
2006; Li et al., 2008). Stemflow is important to concentrate water (Levia and Germer,
2015), nutrients (Dawoe et al., 2018), pathogens (Garbelotto et al., 2003) and bacteria
(Bittar et al., 2018) from the phyllosphere into the pedosphere (Teachey et al., 2018), even
though stemflow accounts for only a minor part of rainfall amount (RA) (6.2%) in contrast
to throughfall (69.8%) and interception loss (24.0%) in dryland ecosystems with annual
mean rainfall ranging in 154–900 mm (Magliano et al., 2019). Stemflow greatly contributes
to the survival of xerophytic plant species (Návar, 2011), the maintenance of patch
structures in arid areas (Kéfi et al., 2007), and the normal functioning of rainfed dryland
ecosystems (Wang et al., 2011).

To quantify the ecohydrological importance of stemflow, numerous studies have been

conducted on stemflow production and efficiency from various aspects, including stemflow
volume (mL), depth (mm), percentage (%), funnelling ratio (unitless), and productivity
(mL·g$^{-1}$, the branch stemflow volume of unit biomass) (Herwitz, 1986; Yuan et al., 2016;
Zabret et al., 2018; Yang et al., 2019). By installing automatic recording devices, the
stemflow process has been gradually determined at 1-h intervals (Spencer and van
Meerveld, 2016), 5-min intervals (André et al., 2008; Levia et al., 2010) and 2-min
intervals (Dunkerley, 2014b). This determination allowed to compute stemflow intensity
(mm·h$^{-1}$) (Germer et al., 2010), flux (mL·min$^{-1}$) (Yang, 2010) and time lag after rain
(Cayuela et al., 2018). Differing from an event-based calculation, the stemflow process
provided insights into the fluctuation of stemflow production at a high temporal resolution.
It permits a better interpretation of the "hot moment" and "hot spot" effects of many
ecohydrological processes (Bundt et al., 2001; McClain et al., 2003). Quantifying the
short-intensity burst and temporal characteristics shed light on the dynamic process and
pulse nature of stemflow (Dunkerley, 2019).

Stemflow cannot be initiated until canopies were saturated by the rains

(Martinez-Meza and Whitford, 1996). The minimal RA needed to start stemflow was
usually calculated by regressing stemflow volume with RA at different plant species (Levia
and Germer, 2015). It also varied with canopy states, i.e., 10.9 and 2.5–3.4 mm for the
leafed oak and beech tress, and 6.0 mm and 1.5–1.9 mm for them in the leafless period
(André et al., 2008; Staelens et al., 2008). Stemflow also frequently continued after rains
ceased due to the rainwater retained on the canopy/branch surface (Iida et al., 2017). *Salix*
*psammophila* and an open tropical forest started stemflow 5–10 min and 15 min later than
the beginning of a rain event in the Mu Us desert of China (Yang, 2010) and the Amazon
basin of Brazil (Germer et al., 2010), respectively. However, 1 h and 1.5 h were needed to
start stemflow after the beginning of a rain event for pine and oak trees in north-eastern
Spain, respectively (Cayuela et al., 2018). For *S. psammophila*, stemflow flux was
maximized 20–210 min after the beginning of a rain event (Yang, 2010), and stemflow
ceased 11 h after rains ceased in an open tropical forest (Germer et al., 2010). Time lags of
stemflow generation, maximization and ending to rains depicted dynamic stemflow process,
and were conducive to better understand the hydrological process occurred at the interface
between the intercepted rains and soil moisture (Sprenger et al., 2019). It was important to
discuss the temporal persistence in spatial patterns of soil moisture particularly at the
intra-event scale (Gao et al., 2019). However, stemflow time lags have not been
systematically studied for xerophytic shrubs.

The preferential paths at the underside of branches for delivering stemflow complicates

stemflow processes within events (Dunkerley, 2014a). The influences of bark microrelief
on stemflow are strongly affected by dynamic rain processes, such as rainfall intensity and
raindrop striking within events (van Stan and Levia, 2010). While exceeding the holding
capacity of branches, high rainfall intensity could overload and interrupt this preferential
path (Carlyle-Mose and Price, 2006). Raindrops hit the canopy surface and create splashes
on the surface. This process is conducive to wetting branches at the lower layers and
accelerating the establishment of the preferential paths of stemflow transportation (Bassette
and Bussière, 2008). Nevertheless, the interaction between the stemflow process and
intra-event rainfall characteristics has not been substantially studied.

This study was designed at the event and process scales to investigate inter-/intra-event

stemflow variability of two dominant xerophytic shrubs. Stemflow volume, intensity,
funnelling ratio and temporal dynamics of *Caragana korshinskii* and *S. psammophila* were
recorded during the 2014–2015 rainy seasons on the Loess Plateau of China. Temporal
dynamics were expressed as stemflow duration and time lags of stemflow generation,
maximization and cessation to rains. Raindrop momentum was introduced to represent the
comprehensive effects of raindrop size, velocity, inclination angle and kinetic energy at the
stemflow process. Funnelling ratio had been calculated at the event base and the 100-s
intervals to assess the convergence effects of stemflow. This study specifically aimed to (1)

depict the stemflow process in terms of stemflow intensity and temporal dynamics, (2) identify the dominant rainfall characteristics influencing inter-/intra-event stemflow variables, and (3) quantify the relationships between stemflow process variables and rainfall characteristics. Achieving these objectives would advance our knowledge of the process-based stemflow production to better understand the pulse nature of stemflow and its interactions with dynamic rain processes.

**2 Materials and Methods**

**2.1 Site description**

This study was conducted in the Liudaogou catchment (110°21′–110°23′E, 38°46′–38°51′N) in Shenmu city, Shaanxi Province, China, during the 2014–2015 rainy seasons. This catchment is 6.9 km$^2$ and 1094–1273 m above sea level (m.a.s.l.). A semiarid continental climate prevails in this area. The mean annual precipitation (MAP) is 414 mm (1971–2013). Most MAP (77%) occurs from July to September (Jia et al., 2013). The mean annual potential evaporation is 1337 mm (Yang et al., 2019). The mean annual temperature is 9.0 °C. The dominant shrubs include *C. korshinskii*, *S. psammophila*, and *Amorpha fruticosa*. The dominant grasses are *Artemisia capillaris*, *Artemisia sacrorum*, *Medicago sativa*, *Stipa bungeana*, etc.

*C. korshinskii* and *S. psammophila* are dominant shrub species at the arid and semi-arid regions of northwestern China (Hu et al., 2016; Liu et al., 2016). They were commonly planted for soil and water conservation, sand fixation and wind barrier, and had extensive distributions at this region (Li et al., 2016). The both species have inverted-cone crowns and no trunks, with multiple branches running obliquely from the base. As modular

organisms and multi-stemmed shrub species, their branches live as independent individuals
and compete with each other for water and light (Firn, 2004). Two plots were established in
the southwestern catchment for these two xerophytic shrubs planted in the 1990s (Fig. 1). *C.*
*korshinskii* and *S. psammophila* plots share similar stand conditions with elevations of 1179
and 1207 m.a.s.l., slopes of 13° and 18°, and sizes of 3294 and 4056 m$^2$, respectively. The
*C. korshinskii* plot has a ground surface of loess and aspect of 224°, while the *S.*
*psammophila* plot has a ground surface of sand and an aspect of 113°.
**2.2 Meteorological measurements and calculations**

A meteorological station was installed at the experimental plot of *S. psammophila* to

record rainfall characteristics and wind speed (WS, m·s$^{-1}$) (Model 03002, R. M. Young
Company, USA), air temperature (T, °C) and relative humidity (H, %) (Model HMP 155,
Vaisala, Finland). They were logged at 10-min intervals by a datalogger (Model CR1000,
Campbell Scientific Inc., USA). Evaporation coefficient (E, unitless) was calculated to
present the evaporation intensity (Equations 1–3) via aerodynamic approaches
(Carlyle-Mose and Schooling, 2015). Tipping-bucket rain gauges (hereinafter referred to as
"TBRG") automatically recorded the volume and timing of rainfall and stemflow (Herwitz,
1986; Germer et al., 2010; Spencer and Meerveld, 2016; Cayuela et al., 2018). To mitigate
the systematic errors for missing the records of inflow during tipping intervals (Groisman
and Legates, 1994), we chose the Onset® (Onset Computer Corp., USA) RG3-M TBRG
with the relatively smaller underestimation for its smaller bucket volume (3.73±0.01 mL)
(Iida et al., 2012). Besides, three 20-cm-diameter standard rain gauges were placed around
TBRG with a 0.5-m distance at the 120° separation (Fig. 1). The regression ($R^2$=0.98,
*p*<0.01) between manual measurements and automatic recording further mitigated the
understanding of inflow water by applying TBRG (Equation 4).
$$e_s = 0.611 \times \exp\left( 17.27 \times T \big/ \left(237.7 + T\right) \right) \tag{1}$$

$$VPD = e_s \times \left(1 - H\right) \tag{2}$$

$$E = WS \times VPD \tag{3}$$

where $e_s$ is the saturation vapor pressure (kPa); T is air temperature (°C); H is air relative
humidity (%); VPD is the vapor pressure deficit (kPa); and E is the evaporation coefficient
(unitless).
$$IW_A = IW_R \times 1.32 + 0.16 \tag{4}$$

where $IW_R$ is the recording of inflow water (including rainfall and stemflow) via TBRG
(mm), and $IW_A$ is the adjusted inflow water (mm).
Discrete rainfall events were defined by a measurable RA of 0.2 mm (the resolution
limit of the TBRG) and the smallest 4-h gap without rains. That was the same period of
time to dry canopies from antecedent rains as reported by Giacomin and Trucchi (1992),
Zhang et al. (2015), Zhang et al., (2017) and Yang et al. (2019). Rainfall interval (RI, h)
was calculated to indirectly represent the bark wetness. Other rainfall characteristics were
also computed, including the RA (mm), rainfall duration (RD, h), the average and 10-min
maximum rainfall intensity of incident rains (I and $I_{10}$, mm·h$^{-1}$), and the 10-min average
rainfall intensity after rain begins ($I_{b10}$, mm·h$^{-1}$) and before rain ends ($I_{e10}$, mm·h$^{-1}$). By
assuming a perfect sphere of a raindrop (Uijlenhoet and Torres, 2006), raindrop momentum
in the vertical direction (F, mg·m·s$^{-1}$) (Equation 8–9) was computed to comprehensively
represent the effects of raindrop size (D, mm) (Equation 5), terminal velocity (v, m·s$^{-1}$)

179 (Equation 6), average inclination angle ($\theta$, °) (Equation 7) affecting stemflow process

180 (Brandt, 1990; Kimble, 1996; van Stan et al., 2011; Carlyle-Moses and Schooling, 2015).

181 The 10-min maximum raindrop momentum ($F_{10}$, mg·m·s$^{-1}$) and the average raindrop

182 momentum at the first and last 10 min ($F_{b10}$ and $F_{e10}$, respectively, mg·m·s$^{-1}$) could be

183 calculated with $I_{10}$, $I_{b10}$ and $I_{e10}$ as indicated at Equation 5–9, respectively. For the 0.8-km

184 distance between the two plots, the meteorological data were used at the *C. korshinskii* plot.

$$D = 2.23 \times (0.03937 \times I)^{0.102} \tag{5}$$

$$v = 3.378 \times \ln(D) + 4.213 \tag{6}$$

$$\tan\theta = \frac{WS}{v} \tag{7}$$

$$F_0 = m \times v = (\frac{1}{6} \times \rho \times \pi \times D^3) \times v \tag{8}$$

$$F = F_0 \times \cos\theta \tag{9}$$

190 where D is raindrop diameter (mm); I is the average rainfall intensity of incident rains

191 (mm·h$^{-1}$); v is raindrop velocity (m·s$^{-1}$); $\theta$ is average inclination angle of raindrops (°); WS

192 is the average wind speed of incident rains (m·s$^{-1}$); $F_0$ is the average raindrop momentum

193 (mg·m·s$^{-1}$); m is the average raindrop mass (g); $\rho$ is the density of freshwater at standard

194 atmospheric pressure and 20°C (0.998 g·cm$^{-3}$).

195 **2.3 Experimental branch selection and measurements**

196  This study focused on the branch-scale stemflow production of the 20-year-old *C.*

197 *korshinskii* and *S. psammophila*. Based on plot investigation, the canopy traits of standard

198 shrubs were determined. Four shrubs were selected accordingly at each species with similar

199 crown areas and heights (5.1±0.3 m$^2$ and 2.1±0.2 m for *C. korshinskii* and 21.4±5.2 m$^2$ and

200 3.5±0.2 m for *S. psammophila*, respectively). The approximately 10-m gap between them

guaranteed shrubs exposing to the similar meteorological conditions (Yuan et al., 2016). We
measured branch morphologies of all 180 and 261 branches at experimental shrubs of *C.*
*korshinskii* and *S. psammophila*, respectively, including BD (Basal diameter, mm) with a
Vernier calliper (Model 7D-01150, Forgestar Inc., Germany), branch length (BL, cm) with
a measuring tape, and branch angle (BA, °) with pocket geologic compass (Model DQL-8,
Harbin Optical Instrument Factory, China), respectively. Thus, BD categories were
determined at 5–10 mm, 10–15 mm, 15–18 mm, 18–25 mm and >25 mm to guarantee the
appropriate branch amounts within categories for meeting the statistical significance. Two
representative branches with median BDs were selected in each category for stemflow
recording. The experimental branches had no intercrossing with neighbouring ones and no
turning point in height from branch tip to base. The outlayer-of-canopy positions avoided
over-shading by the upper layer branches and permitted convenient measurements. Since
the qualified branch with the >25-mm size was not enough for *C. korshinskii* and the
TBRG malfunctioned at the 15–18-mm branches of *S. psammophila*, stemflow data were
not available in these BD categories. In total, 7 branches were selected for stemflow
measurements at each species (Table 1). As the important interface to intercept rains at the
growing season, the well-verified allometric growth equations were performed to estimate
the branch leaf area (LA, cm$^2$) of *C korshinskii* (LA=39.37×BD$^{1.63}$ $R^2$=0.98) (Yuan et al.,
2017) and *S. psammophila* (LA=18.86×BD$^{1.74}$ $R^2$=0.90) (Yuan et al., 2016), respectively.
**2.4 Stemflow measurements and calculations**
A total of 14 TBRGs had been applied to automatically record the branch stemflow
production of *C. korshinskii* and *S. psammophila*. The data of stemflow volume and timing
were automatically recorded at dynamic intervals between neighboring tips. We installed
aluminium foil collars to trap stemflow at branches nearly 40 cm off the ground, higher
than TBRG orifice with height of 25.7 cm (Fig. 1). They were fitted around the entire
branch circumference and sealed by neutral silicone caulking. The limited orifice diameter
of foil collars minimized the accessing of throughfall and rains into them (Yuan et al.,
2017). The 0.5-cm-diameter polyvinyl chloride hoses hung vertically and channelled
stemflow from the collars to TBRGs with a minimum travel time. TBRGs were covered
with the polyethylene films to prevent the accessing of throughfall and splash (Fig. 1).
These apparatuses were periodically checked against leakages or blockages by insects and
fallen leaves. Stemflow variables were computed as follow.
(1) Stemflow volume (SFV, mL): the average stemflow volume of individual branches.

Adjusted with Equation 4 firstly, SFV was computed with the TBRG recordings

($SF_{RG}$, mm) by multiplying its orifice area (186.3 $cm^2$) (Equation 10).

$$SFV = SF_{RG} \times 18.63 \tag{10}$$

(2) Stemflow intensity: the branch stemflow volume per branch basal area per unit

time. SFI ($mm \cdot h^{-1}$) is the average stemflow intensity of incident rains, which is

computed by the event-based SFV (mL), branch basal area (BBA, $mm^2$) and RD

(h) (Equation 11) (Herwitz, 1986; Spencer and Meerveld, 2016). $SFI_{10}$ ($mm \cdot h^{-1}$) is

the 10-min maximum stemflow intensity, which is calculated with the 10-min

maximum stemflow volume ($SFV_{10}$, mL) and BBA ($mm^2$) (Equation 12). $SFI_i$

($mm \cdot h^{-1}$) is the instantaneous stemflow intensity, which is calculated by the tip

volume of TBRG (3.73 mL), BBA ($mm^2$) and time intervals between neighbouring

tips ($t_i$, h) (Equation 13). The comparison between $SFI_i$ and the corresponding

rainfall intensity depicted the synchronicity of stemflow with rains within event.

$$SFI = 1000 \times {SFV} \big/ {(BBA \times RD)} \tag{11}$$

$$SFI_{10} = 6000 \times {SFV_{10}} \big/ {BBA} \tag{12}$$

$$SFI_i = {3730} \big/ {(BBA \times t_i)} \tag{13}$$

(3) Stemflow temporal dynamics: stemflow duration and time lags to rains.

SFD (h): stemflow duration. It is computed by different timings between the first-

and last-tips of stemflow via TBRG.

TLG (min): time lag of stemflow generation after rain begins. It is computed by

different first-tip timings between rainfall and stemflow via TBRG.

TLM (min): time lag of stemflow maximization after rain begins. It is computed

by different timings between the largest-$SFI_i$ and first-rainfall tips via TBRG.

TLE (min): time lag of stemflow ending after rain ceases. It is computed by

different last-tip timings between rainfall and stemflow via TBRG.

(4) Funnelling ratio: the efficiency for capturing and delivering raindrops from the

canopies to trunk/branch base (Siegert and Levia, 2014; Cayuela et al., 2018). By

introducing RD at both numerator and denominator of the original equation

(Herwitz, 1986), FR (unitless) was transformed as the ratio between stemflow and

rainfall intensities at the event base (Equation 14). $FR_{100}$ described the

within-event funnelling ratio at the 100-s interval after rain began (Equation 15).

$$FR = 1000 \times \frac{SFV}{BBA \times RA} = 1000 \times \frac{\dfrac{SFV}{BBA}\big/ RD}{RA \big/ RD} = \frac{SFI}{I} \tag{14}$$

$$\qquad\qquad\qquad FR_{100_i} = \frac{SFI_{100_i}}{I_{100_i}} \qquad\qquad\qquad (15)$$
where $FR_{100i}$, $SFI_{100i}$ and $I_{100i}$ are funnelling ratio, stemflow intensity and rainfall
intensity at the internal $i$ with 100-s pace after rain begins, respectively.
**2.5 Data analysis**
Stemflow variables were averaged at different BD categories to analyse the most
influential rainfall characteristics affecting them. Pearson correlation analyses were firstly
performed to test the relationships between rainfall characteristics (RA, RD, RI, I, $I_{10}$, $I_{b10}$,
$I_{e10}$, F, $F_{10}$, $F_{b10}$, $F_{e10}$ and E) and stemflow variables (SFV, SFI, $SFI_{10}$, FR, TLG, TLM, TLE
and SFD). The significantly related factors were grouped in terms of median value, and
compiled into indicator matrices. They were standardized for a cross-tabulation check as
required by the multiple correspondence analysis (MCA) (Levia et al., 2010; van Stan et al.,
2011, 2016). All qualified data were restructured into orthogonal dimensions (Hair et al.,
1995), where distances between row and column points were maximized (Hill and Lewicki,
2007). As shown at correspondence maps, the clustering rainfall characteristics tightly
related to the centred stemflow variable. Finally, stepwise regressions were operated to
identify the most influential rainfall characteristics (Carlyle-Moses and Schooling, 2015).
The quantitative relations were established in terms of the qualified level of significance ($p$
<0.05) and the highest coefficient of determination ($R^2$). One-way analysis of variance
(ANOVA) with LSD post hoc test was used to determine whether rainfall characteristics,
and stemflow variables significantly differed among event categories, and whether
funnelling ratio and stemflow intensity significantly differed among BD categories for *C.*
*korshinskii* and *S. psammophila*. The level of significance was set at 95% confidence
interval ($p$=0.05). SPSS 21.0 (IBM Corporation, USA), Origin 8.5 (OriginLab Corporation,
USA) and Excel 2019 (Microsoft Corporation, USA) were used for data analysis.
**3 Results**
**3.1 Rainfall characteristics**

A total of 54 rainfall events had been recorded for stemflow measurements at the

2014–2015 rainy seasons (Fig. 2). Thereinto, 20, 8, 10, 8, 4 and 4 events were at the RA
categories of ≤2 mm, 2–5 mm, 5–10 mm, 10–15 mm, 15–20 mm and >20 mm, respectively.
The total RAs at these categories were 22.1 mm, 26.1 mm, 68.8 mm, 93.3 mm, 74.8 mm
and 110.0 mm, respectively. During these events, the average I, $I_{10}$, $I_{b10}$ and $I_{e10}$ were
4.5±1.0 mm·h$^{-1}$, 10.9±2.1 mm·h$^{-1}$, 5.5±1.4 mm·h$^{-1}$ and 2.8±0.7 mm·h$^{-1}$, respectively. The
average F, $F_{10}$, $F_{b10}$ and $F_{e10}$ were 16.1±1.2 mg·m·s$^{-1}$, 24.9±1.4 mg·m·s$^{-1}$, 18.4±1.4
mg·m·s$^{-1}$ and 16.0±1.0 mg·m·s$^{-1}$, respectively. RD, RI and E averaged 4.7±0.8 h, 50.6±6.1
h, and 0.9±0.2, respectively (Table 2).

Rainfall events were further categorized in terms of rainfall-intensity peak amount,

including Events A (the single-peak events), B (the double-peak events) and C (the
multiple-peak events). There were 17, 11 and 15 events at Event A, B and C, respectively.
Because the remaining 11 events had the average RA of 0.6 mm, no more than three
recordings had been observed within event which was limited by 0.2-mm resolution of
TBRGs. Therefore, they could not be categorized and grouped as Event others (Table 2).
Compared with Events A and B, Event C possessed significantly different rainfall
characteristics, e.g., the significantly larger RA (11.7 vs. 4.1 and 5.2 mm) and RD (10.3 vs.
2.5 and 3.6 h) but the significantly smaller $I_{10}$ (9.5 vs. 15.5 and 12.7 mm·h$^{-1}$), $I_{b10}$ (2.8 vs.
7.7 and 9.9 mm·h$^{-1}$), $F_{b10}$ (15.4 vs. 19.7 and 21.7 mg·m·s$^{-1}$) and $F_{e10}$ (13.4 vs. 17.3 and
16.6 mg·m·s$^{-1}$), the non-significantly smaller $I_{e10}$ (2.1 vs. 4.3 and 3.6 mm·h$^{-1}$), $F_{10}$ (24.2 vs.
27.8 and 26.6 mg·m·s$^{-1}$) and E (0.4 vs. 0.9 and 1.0), respectively (Table 2).
In general, rainfall events were skewedly distributed in terms of RA. The occurrences
of events with a RA≤2 mm dominated the experimental period (40.7%), but the events with
RA>20 mm were the greatest contributor to the total RA (28.0%). However, a relatively
equal distribution was noted during events with single (17 events), double (11 events) and
multiple (15 events) rainfall-intensity peaks. Comparatively, the multiple-peak events had
significantly larger rainfall amounts, durations, intensities and raindrop momentums.
**3.2 Inter-/intra-event stemflow variability**
Stemflow variables of *C. korshinskii* and *S. psammophila* showed great inter-event
variations during the experimental period (Fig. 3). *C. korshinskii* had larger SFV, SFI, SFI$_{10}$,
FR, SFD, TLG and TLE (226.6±46.4 mL, 517.5±82.1 mm·h$^{-1}$, 2057.6±399.7 mm·h$^{-1}$,
130.7±8.2, 3.8±0.8 h, 66.2±10.6 min and 20.0±5.3 min, respectively) but smaller TLM
(109.4±20.5 min) than those of *S. psammophila* (172.1±34.5 mL, 367.3±91.1 mm·h$^{-1}$,
1132.2±214.3 mm·h$^{-1}$, 101.6±10.4, 3.4±0.9 h, 54.8±11.7 min, 13.5±17.2 min, and
120.5±22.1 min, respectively) (Table 3). During the 54 events, no negative values were
observed for TLG and TLM but TLE. It indicated that stemflow generally initiated and
maximized after rains started for both species. However, stemflow might be ended before
(negative TLE) and after (positive TLE) rains ceased.
Stemflow well synchronized to rains with similar intensity peak shapes, amounts and
positions for both species. These results were vividly demonstrated at representative rains

with different intensity peak amounts and RAs, including events on July 17, 2015 (Event A, 20.7 mm), July 29, 2015 (Event B, 7.3 mm), and September 10, 2015 (Event C, 13.3 mm) (Fig. 4). *C. korshinskii* had larger $FR_{100}$ (91.7, 76.1 and 94.0, respectively) than those of *S. psammophila* (32.8, 26.3 and 43.7, respectively) during representative events. It indicated a comparatively greater ability of converging rains for *C. korshinskii* within event.

Stemflow variables varied between rainfall event categories. For Event C in comparison to Events A and B, *S. psammophila* had significantly larger SFV (435.2 vs. 102.6 and 145.7 mL), SFD (8.3 vs. 1.2 and 3.4 h), TLM (235.8 vs. 64.3 and 93.4 min), FR (129.1 vs. 77.1 and 91.4), non-significantly larger TLE (20.8 vs. 17.1 and 8.6 min) but significantly smaller SFI (246.6 vs. 648.1 and 421.5 $mm \cdot h^{-1}$) and $SFI_{10}$ (888.4 vs. 1672.7 and 1582.8 $mm \cdot h^{-1}$), respectively (Table 3). SFI decreased at events with increasing intensity peak amounts as shown at Events A–C. The drop of SFI was offset by the decreasing I to some extent (Table 2), which might partly explain the increasing trend of FR from Event A to C. *C. korshinskii* shared similar changing trends of stemflow variables between event categories with those of *S. psammophila*, except for the non-significantly smaller TLE (18.5 min) at Event C in contrast to TLE at Event A and B (22.3 and 18.7 min).

Funnelling ratio and stemflow intensity negatively related with branch size. *C. korshinskii* and *S. psammophila* had significantly greater FR, SFI, and $SFI_{10}$ at the 5–10 mm branches than those at the larger branches (Table 4). For *C. korshinskii*, FR decreased from 163.7±12.2 at the 5–10-mm branches to 97.7±9.2 at the 18–25-mm branches, respectively. It was consistent with decreasing SFI (333.8–716.2 $mm \cdot h^{-1}$) at the corresponding BD categories (Table 4). As branch size increased, *S. psammophila* shared

similar decreasing trends of FR (44.2–212.0) and SFI (197.2–738.7 mm·h$^{-1}$), respectively.

**3.3 Relationships between stemflow variables and rainfall characteristics**

*C. korshinskii* and *S. psammophila* had similar correspondence patterns between
rainfall characteristics and stemflow variables. As shown in Fig. 5, the one-to-one
correspondences were observed for SFV and TLE. The larger (or smaller) SFV and TLE
corresponded to the larger (or smaller) RA and RI, respectively. This result demonstrated
the dominant influences of RA and RI on SFV and TLE, respectively. The one-to-two
correspondences was noted for SFD with RD and E. The larger (or smaller) SFD
corresponded to the larger (or smaller) RD and smaller (or larger) E. RA had been
identified as the dominant rainfall characteristic affecting FR based on the analysis for 53
branches of *C. korshinskii* and 98 branches of *S. psammophila* at the same plots during the
same experimental period (Yuan et al., 2017). It seemed that event-based stemflow
production (the volume, duration and efficiency) were strongly influenced by rainfall
characteristics at inter-event scale (the rainfall amount and duration).
The one-to-more correspondences were observed for TLM, TLG, SFI and SFI$_{10}$ (Fig.
5). The larger (or smaller) TLM corresponded to the smaller (or larger) rainfall
characteristics of I, I$_{10}$, I$_{b10}$, I$_{e10}$, F, F$_{10}$, F$_{b10}$ and F$_{e10}$. The same correspondences were
applied to the larger (or smaller) TLG, and the smaller (or larger) SFI and SFI$_{10}$. It seemed
that the within-event stemflow processes (SFI, SFI$_{10}$, TLG and TLM) were strongly
affected by rainfall characteristics at intra-event scale (the rainfall intensity and raindrop
momentum). Therefore, these results indicated that rainfall characteristics influenced
stemflow variables at the corresponding temporal scales. This influence occurred at the
inter-event scale between SFV and RA, FR and RA, SFD and RD, and at the intra-event
scale for stemflow time lags (TLG and TLM) and intensities (SFI and $SFI_{10}$) with rainfall
intensity (I, $I_{10}$, $I_{b10}$ and $I_{e10}$) and raindrop momentum (F, $F_{10}$, $F_{b10}$ and $F_{e10}$). The only
exception was noted between TLE and RI for the mismatched temporal sales.
Stepwise regression analysis identified the most influential rainfall characteristics
affecting stemflow intensities and temporal dynamics. RD was the dominant rainfall
characteristics affecting SFD. $I_{10}$ significantly affected the TLM of the both species. For *C.*
*korshinskii*, I, $I_{10}$ and F were the most influential factors on SFI, $SFI_{10}$ and TLG,
respectively. However, for *S. psammophila*, F, $F_{10}$ and $F_{b10}$ significantly affected SFI, $SFI_{10}$
and TLG, respectively. The results of multiple regression analysises indicated that there
were linear relationships between SFI and I ($R^2$=0.74, $p$<0.01) and $SFI_{10}$ and $I_{10}$ ($R^2$=0.85,
$p$<0.01) for *C. korshinskii* and between SFD and RD for *C. korshinskii* ($R^2$=0.95, $p$<0.01)
and *S. psammophila* ($R^2$=0.92, $p$<0.01) (Fig. 6). Moreover, power functional relations were
found between SFI and F ($R^2$=0.82, $p$<0.01), $SFI_{10}$ and $F_{10}$ ($R^2$=0.90, $p$<0.01) (Fig. 6), TLG
and $F_{b10}$ ($R^2$=0.55, $p$<0.01) and TLM and $I_{10}$ ($R^2$=0.40, $p$<0.01) (Fig. 7) for *S. psammophila*,
and TLG and F ($R^2$=0.56, $p$ <0.01) and TLM and $I_{10}$ ($R^2$=0.38, $p$<0.01) (Fig. 7) for *C.*
*korshinskii*. However, there was no significant quantitative relationship between TLE and
RI for *C. korshinskii* ($R^2$=0.005, $p$=0.28) or *S. psammophila* ($R^2$=0.002, $p$=0.78) (Fig. 7).
**4 Discussion**
**4.1 Stemflow intensity and funnelling ratio**
Stemflow intensity is generally greater than rainfall intensity at different plant life
forms. The xerophytic shrubs of *C. korshinskii* and *S. psammophila* had larger average
stemflow intensities than the average rainfall intensity (517.5 and 367.3 mm·h$^{-1}$ vs. 4.5
mm·h$^{-1}$). Broadleaf and coniferous species (*Quercus pubescens* Willd. and *Pinus sylvestris*
L., respectively) also have larger maximum stemflow intensities than the maximum rainfall
intensity in north-eastern Spain (Cayuela et al., 2018). The gap between stemflow and
rainfall intensities generally increased as the recording time intervals decreased. While
recording at the 1-h intervals, approximately 20-, 17-, 13- and 2.5-fold greater peak
stemflow intensities had been observed for trees of Cedar, Birch, Douglas Fir and Hemlock,
respectively, at the coastal British Columbia forest (Spencer and Meerveld, 2016). For *C.*
*korshinskii* and *S. psammophila*, in comparison to $I_{10}$ (10.9 mm·h$^{-1}$) at 10-min intervals, the
$SFI_{10}$ (2057.6 and 1132.2 mm·h$^{-1}$, respectively) was over 103.9-fold greater. The
recordings at 6-min interval indicated a 157-fold larger of stemflow intensity (18840 mm·h$^{-1}$
) than rainfall intensity (120 mm·h$^{-1}$) in the cyclone-prone tropical rainforest with
extremely high MAP of 6570 mm (Herwitz, 1986). While calculating the dynamic time
interval between neighbouring tips of TBRG, $SFI_i$ (10816.2 mm·h$^{-1}$) was 150.2-fold
greater than the corresponding rainfall intensity (72 mm·h$^{-1}$). Therefore, stemflow recorded
at a higher temporal resolution might provide more information into the dynamic nature of
stemflow and real-time responses to rainfall characteristics within events.

Greater stemflow intensity than rainfall intensity is hydrologically significant at

terrestrial ecosystems. This scenario indicates the convergence of the canopy-intercepted
rains into the limited area around trunk or branch bases within a certain time period, i.e.,
8.0% and 3.5% of rains being directed to the trunk base only accounting for 0.3% and 0.4%
of plot area in the open rainforest (Germer et al., 2010) and undisturbed lowland tropical
rainforest (Manfroi et al., 2004), respectively. Besides, FR, which compared SFV with RA
that would have been collected at the same area as the basal area at an event scale (Herwitz,
1986), is commonly applied to assess the convergence effect via stemflow volume, rainfall
amount and basal area (Carlyle-Moses et al., 2010; Siegert and Levia, 2014; Fan et al.,
2015; Yang et al., 2019). If FR is greater than 1, more water is collected at the trunk or
branch base than at the clearings. Both methods successfully quantified the convergence
effects of stemflow. However, the former provided a possibility to assess it at high temporal
resolutions within event.
This study established the quantitative connection between FR and stemflow intensity.
As per Equation 14 and the average stemflow and rainfall intensities listed at Table 2 and 3,
FR could be estimated to be 115.0 and 81.6 for *C. korshinskii* and *S. psammophila*,
respectively. Those results approximately agreed with FR of 173.3 and 69.3 (Yuan et al.,
2017) and 124.9 and 78.2 (Yang et al., 2019) for the two species by applying the traditional
calculation based on SFV and RA (Herwitz, 1986). As branch size increased, FR of *C.*
*korshinskii* decreased from 163.7 at the 5–10-mm branches to 97.7 at the 18–25-branches.
The decreasing trend of FR of *S. psammophila* were also noted in the range of 44.2–212.0
with increasing BD. The negative relation between BD and FR agreed with the reports for
trees and babassu palms in an open tropical rainforest in Brazil (Germer et al., 2010), the
mixed-species coastal forest at British Columbia of Canada (Spencer and Meerveld, 2016),
for trees (*Pinus tabuliformis* and *Armeniaca vulgaris*) and shrubs (*C. korshinskii* and *S.*
*psammophila*) on the Loess Plateau of China (Yang et al., 2019). It might be partly
explained by the decreasing stemflow intensities with increasing branch size as per
Equation 14. Our results found that SFI decreased from 716.2 to 333.8 for *C. korshinskii*,
and 738.7 to 197.2 for *S. psammophila* as branch size increased (Table 4). It well justified
the importance of branch size on stemflow intensity. Associated with the infiltration rate,
the stemflow-induced hydrological process might be strongly affected, i.e., soil moisture
recharge, Hortonian overland flow (Herwitz, 1986), saturation overland flow (Germer et al.,
2010), soil erosion (Liang et al., 2011), nutrient leaching (Corti et al., 2019), etc. Therefore,
more attention should be paid to tree/branch size and size-related stand age at future studies
while modeling the stemflow-induced terrestrial hydrological fluxes.

The importance had been addressed to study the funnelling ratio at the stand scale

(Carlyle-Moses et al., 2018); however, it had not been adequately studied at the intra-event
scale. This study calculated the average funnelling ratio at the event base and the 100-s
intervals after rain began. Thus, the convergence effect of stemflow could be better
understood at the inter-/intra-event scales. Our results found that $FR_{100}$ were over 1.8-fold
greater than FR of *C. korshinskii* (282.7 vs. 130.7) and *S. psammophila* (203.4 vs. 101.6),
respectively. It indicated that funnelling ratio fluctuated dramatically within event.
Therefore, computing FR at event and ignoring it at high temporal resolutions within event
might underestimate the eco-hydrological significance of stemflow.

In general, stemflow intensity highly related to funnelling ratio. For addressing its

eco-hydrological importance, stemflow intensity should be precisely defined. It had been
expressed as the stemflow volume per basal area of branches/trunks per unit time with the
unit of mm·h$^{-1}$ (Herwitz, 1986; Spencer and Meerveld, 2016) and mm·5 min$^{-1}$ (Cayuela et
al., 2018). However, stemflow intensity had also been described as stemflow volume per
unit time with the unit of L·week$^{-1}$ (Schimmack et al., 1993) and L·h$^{-1}$ (Liang et al., 2011;
Germer et al., 2013). We highly recommended the former definition. Because of its highly
spatial-related attribution (Herwitz, 1986; Liang et al., 2011; 2014), the eco-hydrological
significance of stemflow would be underestimated by ignoring the basal area, over which
stemflow was received. Moreover, as per this definition, stemflow intensity quantitively
connected with funnelling ratio via Equation 14. Thus, funnelling ratio could be used to
assess the convergence effect of stemflow at both inter- and intra-event scales.
**4.2 Stemflow temporal dynamics**
Stemflow well synchronized to the rains. It agreed with the report of Levia et al.
(2010), who demonstrated a marked synchronicity between SFV and RA in 5-min intervals
for *Fagus. grandifolia*. The duration and time lags to rains were critical to describe
stemflow temporal dynamics. Our results indicated that in comparison to *S. psammophila*,
*C. korshinskii* takes a longer time to initiate (66.2 vs. 54.8 min), end (20.0 vs. 13.5 min)
and produce stemflow (3.8 vs. 3.4 h) but a shorter time to maximize stemflow (109.4 vs.
120.5 min, respectively). Moreover, the TLMs of both species were in the range of the
TLMs for *S. psammophila* (20–210 min) in the Mu Us desert of China (Yang, 2010).
Varying TLGs were documented for different species. Approximately 15 min, 1 h and
1.5 h were needed to initiate the stemflow of palms (Germer, 2010), pine trees and oak
trees (Cayuela et al., 2018), respectively. In addition, an almost instantaneous start of
stemflow had also been observed as rain began for *Quercus rubra* (Durocher, 1990), *Fagus*
*grandifolia* and *Liriodendron tulipifera* (Levia et al., 2010). Compared to the positive TLE
dominating xerophytic shrubs, the TLE greatly varied with tree species. TLE was as much
as 48 h for Douglas fir, oak and redwood in California, USA (Reid and Levia, 2009), and
almost 11 h for palm trees in Brazil (Germer, 2010). However, for sweet chestnut and oak,
almost no stemflow continued when rains ceased in Bristol, England (Durocher, 1990).
These scenarios might occur due to the sponge effect of the canopy surface (Germer, 2010),
which buffered stemflow generation, maximization and cessation before saturation. These
conclusions were consistent with the smaller stemflow intensities of *C. korshinskii* and *S.*
*psammophila* than the rainfall intensity when rain began, as part of the rains was used to
wet canopies (Fig. 4). The hydrophobic bark traits benefited stemflow initiation with the
limited time lags to rains. In contrast, the hydrophilic bark traits were conducive for
continuing stemflow after rain ceased, which kept the preferential flow paths wetter for
longer time periods (Levia and Germer, 2015). As a result, it took time to transfer
intercepted rains from the leaf, branch and trunk to the base. This process strongly affects
the stemflow volume, intensity and loss as evaporation.
The dynamics of intra-event rainfall intensity complicated the stemflow time lags to
rains. A 1-h lag to begin and stop stemflow with the beginning and ending of rains had been
observed for ashe juniper trees during high-intensity events, but no stemflow was generated
at low-intensity storms (Owens et al., 2006). Rainfall intensity was an important dynamic
rainfall characteristic affecting stemflow volume. Owens et al. (2006) found the most
significant difference between various rainfall intensities located in the stemflow patterns
other than throughfall and interception loss. During events with a front-positioned, single
rainfall-intensity peak, *S. psammophila* maximized stemflow in a shorter time than *C.*
*korshinskii* did in the Mu Us desert (30 and 50 min) (Yang, 2010). These results highlighted
the amounts and occurrence time of rainfall-intensity peak affecting the stemflow process,
which was consistent with the finding of Dunkerley (2014b).
Raindrops presented rainfall characteristics at finer temporal-spatial scales. They were
usually ignored because rains were generally regarded as a continuum rather than a discrete
process consisting of individual raindrops of various sizes, velocities, inclination angles
and kinetic energies. Raindrops hit the canopy surface and created splashes at different
canopy layers (Bassette and Bussière, 2008; Li et al., 2016). This process accelerated
canopy wetting and increased water supply for stemflow production. Therefore, raindrop
momentum was introduced in this study to represent the comprehensive effects of raindrop
attributes. Our results indicated that raindrop momentum was sensitive to predicting the
variations in stemflow intensity and temporal dynamics with significant linear or power
functional relations (Figs. 6 and 7). Compared with the importance of rainfall intensity for
*C. korshinskii*, raindrop momentum more significantly affected the stemflow process of *S.*
*psammophila*. This result might be related to the larger canopy size and height of *S.*
*psammophila* (21.4±5.2 m$^2$ and 3.5±0.2 m) than that of *C. korshinskii* (5.1±0.3 m$^2$ and
2.1±0.2 m, respectively). More layers were available within canopies of *S. psammophila* to
intercept the splashes created by raindrop striking (Bassette and Bussière, 2008; Li et al.,
2016), thus shortening the paths and having more water supply for stemflow production.
**4.3 Temporal-dependent influences of rainfall characteristics on stemflow variability**
This study discussed stemflow variables and rainfall characteristics at inter-/intra-event
scales. We found that rainfall characteristics affected stemflow variables at the
corresponding temporal scales. RA and RD controlled SFV, FR and SFD, respectively, at
the inter-event scale. However, stemflow intensity (e.g., SFI and $SFI_{10}$) and temporal
dynamics (e.g., TLG and TLM) were strongly influenced by rainfall intensity (e.g., I, $I_{10}$
and $I_{b10}$) and raindrop momentum (e.g., F, $F_{10}$ and $F_{b10}$) at the intra-event scales. These
results were verified by the well-fitting linear or power functional equations among them
(Figs. 6 and 7). Furthermore, the influences of rainfall intensity and raindrop momentum on
stemflow process were species-specific. In contrast to the significance of rainfall intensity
on the stemflow process of *C. korshinskii*, raindrop momentum imposed a greater influence
on the stemflow process of *S. psammophila*.

In general, rainfall characteristics had temporal-dependent influences on the

corresponding stemflow variables. The only exception was found between TLE and RI. RI
tightly corresponded to TLE for both species tested by the MCA, but there was no
significant quantitative relationship between them ($R^2$=0.005, $p$=0.28 for *C. korshinskii*,
and $R^2$=0.002, $p$=0.78 for *S. psammophila*). This result might be related to the mismatched
temporal scales between TLE and RI. TLE represented stemflow temporal dynamics at the
intra-event scale, while RI was the interval times between neighbouring rains at the
inter-event scale. The mismatched temporal scales might also partly explain the
long-standing debates on the controversial positive, negative and even no significant
influences of rainfall intensity (depicting raining process at 5 min, 10 min, 60 min, etc.) on
event-based stemflow volume (Owens et al., 2006; André et al., 2008; Zhang et al., 2015).
**5 Conclusions**

Stemflow intensity and temporal dynamics are important in depicting the stemflow

process and its interactions with rainfall characteristics within events. We categorized
stemflow variables into the volume, intensity, funnelling ratio and temporal dynamics, thus
to representing the stemflow yield, efficiency and process. Funnelling ratio had been
calculated as the ratio between stemflow and rainfall intensities, which enabled to assess
the convergence of stemflow at the inter-/intra-event scales. Over 1.8-fold greater $FR_{100}$
were noted than FR at representative events for *C. korshinskii* and *S. psammophila*,
respectively. FR decreased with increasing branch size of both species. It could be partly
explained by the decreasing trends of SFI as branch size increased. The rainfall
characteristics had temporal-dependent influences on stemflow variables. RA and RD
controlled SFV, FR and SFD at the inter-event scale. Rainfall intensity and raindrop
momentum significantly affected stemflow intensity and time lags to rains at the intra-event
scale except for TLE. The eco-hydrological significance of stemflow might be
underestimated by ignoring stemflow production at high temporal resolutions within event.
These findings advance our understanding of the stemflow process and its influential
mechanism and help model the critical process-based hydrological fluxes of terrestrial
ecosystems.

*Data availability.* The data collected in this study are available upon request to the authors.

*Author contributions*. GYG and CY set up the research goals and designed field
experiments. CY measured and analyzed the data. GYG and BJF provided the financial
support for the experiments, and supervised the execution. CY created the figures and
wrote the original draft. GYG, BJF, DMH, XWD and XHW reviewed and edited the draft
in serval rounds of revision.

*Competing interests.* The authors declare that they have no conflict of interest.

*Acknowledgements.* This research was sponsored by the National Natural Science
Foundation of China (nos. 41390462, 41822103 and 41901038), the National Key
Research and Development Program of China (no. 2016YFC0501602), the Chinese
Academy of Sciences (no. QYZDY-SSW-DQC025), the Youth Innovation Promotion
Association CAS (no. 2016040), and the China Postdoctoral Science Foundation (no.
2018M633427). We appreciate Prof. D. F. Levia in University of Delaware for reviewing
and improving this manuscript. Thanks to Liwei Zhang for the catchment GIS mapping.
Special thanks are given to Shenmu Erosion and Environment Research Station for
experimental support to this research. We thank Prof. David Dunkerley and two anonymous
reviewers for their professional comments, which greatly improve the quality of this
manuscript.

**Appendix**
List of symbols

| Abbreviation | Descriptions | Unit |
|:---:|:---|:---:|
| a.s.l. | above sea level | NA |
| BA | Branch angle | ° |
| BBA | Branch basal area | $mm^2$ |
| BD | Branch diameter | mm |
| BL | Branch length | cm |
| D | Diameter of rain drop | mm |
| $e_s$ | Saturation vapor pressure | kPa |
| E | Evaporation coefficient | unitless |

| | | |
|---|---|---|
| F | Average raindrop momentum in the vertical direction of incident event | mg·m·s$^{-1}$ |
| F$_0$ | Average raindrop momentum of incident event | mg·m·s$^{-1}$ |
| F$_{10}$ | The 10-min maximum raindrop momentum | mg·m·s$^{-1}$ |
| F$_{b10}$ | Average raindrop momentum at the first 10 min | mg·m·s$^{-1}$ |
| F$_{e10}$ | Average raindrop momentum at the last 10 min | mg·m·s$^{-1}$ |
| FR | Average funnelling ratio of incident event | unitless |
| FR$_{100}$ | Funnelling ratio at the 100-s intervals after rain begins | unitless |
| H | Air relative humidity | % |
| I | Average rainfall intensity of incident event | mm·h$^{-1}$ |
| I$_{10}$ | The 10-min maximum rainfall intensity | mm·h$^{-1}$ |
| I$_{b10}$ | Average rainfall intensity at the first 10-min of incident event | mm·h$^{-1}$ |
| I$_{e10}$ | Average rainfall intensity at the last 10-min of incident event | mm·h$^{-1}$ |
| IW$_A$ | The adjusted inflow water at TBRG | mm |
| IW$_R$ | The recorded inflow water at TBRG | mm |
| LA | Leaf area of individual branch | cm$^2$ |
| MAP | Mean annual precipitation | mm |
| MCA | Multiple correspondence analysis | NA |
| NA | Not applicable | NA |
| $p$ | Level of significance | NA |
| $R^2$ | Coefficient of determination | NA |
| RA | Rainfall amount | mm |
| RD | Rainfall duration | h |
| RI | Rainfall interval | h |
| SE | Standard error | NA |
| SFD | Stemflow duration from its beginning to ending | h |
| SFI | Average stemflow intensity of incident event | mm·h$^{-1}$ |
| SFI$_{10}$ | The 10-min maximum stemflow intensity of incident event | mm·h$^{-1}$ |
| SFI$_i$ | Instantaneous stemflow intensity | mm·h$^{-1}$ |
| SF$_{RG}$ | Stemflow depth recorded by TBRG | mm |
| SFV | Stemflow volume | mL |
| t$_i$ | Time intervals between neighboring tips | h |
| T | Air temperature | °C |
| TBRG | Tipping bucket rain gauge | NA |
| TLE | Time lag of stemflow ending to rainfall ceasing | min |
| TLG | Time lag of stemflow generation to rainfall beginning | min |
| TLM | Time lag of stemflow maximization to rainfall beginning | min |
| v | Terminal velocity of rain drop | m·s$^{-1}$ |
| VPD | Vapor pressure deficit | kPa |
| WS | Wind speed | m·s$^{-1}$ |
| ρ | Density of freshwater at standard atmospheric pressure and 20°C | g·cm$^{-3}$ |
| θ | Inclination angle of rain drop | ° |

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

**Table 1.** Branch morphologies of *C. korshinskii* and *S. psammophila* for stemflow recording.

| Shrub species | BD categories (mm) | Branch amount | BD (mm) | BL (cm) | BA (°) | LA (cm$^2$) |
|---|---|---|---|---|---|---|
| *C. korshinskii* | 5–10 | 2 | 6.6 | 131 | 61 | 837.1 |
| | 10–15 | 2 | 13.1 | 168 | 43 | 2577.3 |
| | 15–18 | 2 | 17.8 | 206 | 72 | 4243.1 |
| | 18–25 | 1 | 22.1 | 242 | 50 | 6394.7 |
| | >25 | NA | NA | NA | NA | NA |
| *S. psammophila* | 5–10 | 2 | 7.5 | 248 | 69 | 626.3 |
| | 10–15 | 2 | 13.2 | 343 | 80 | 1683.5 |
| | 15–18 | NA | NA | NA | NA | NA |
| | 18–25 | 2 | 21.8 | 286 | 76 | 3468.3 |
| | >25 | 1 | 31.3 | 356 | 60 | 7513.7 |

Notes: BD, BL and BA are branch basal diameter, length and inclination angle, respectively; LA is leaf area
of individual branches; NA means not applicable.
**Table 2.** Rainfall characteristics during events with different intensity peak amounts.

| Indicators | Event A | Event B | Event C | Others | Average |
|---|---|---|---|---|---|
| Event amount | 17 | 11 | 15 | 11 | 13.5±1.5 |
| RA (mm) | 4.1 ab | 5.2 b | 11.7 c | 0.6 a | 5.4 ± 0.9 |
| RD (h) | 2.5 a | 3.6 a | 10.3 b | 2.2 a | 4.7 ± 0.8 |
| RI (h) | 48.5 ab | 70.5 b | 57.3 ab | 26.1 a | 50.6 ± 6.1 |
| I (mm·h$^{-1}$) | 5.6 a | 5.5 a | 4.6 a | 2.2 b | 4.5 ± 1.0 |
| I$_{10}$ (mm·h$^{-1}$) | 15.5 a | 12.7 ab | 9.5 b | 6.0 c | 10.9 ± 2.1 |
| I$_{b10}$ (mm·h$^{-1}$) | 7.7 a | 9.9 a | 2.8 b | 1.6 b | 5.5 ± 1.4 |
| I$_{e10}$ (mm·h$^{-1}$) | 4.3 a | 3.6 a | 2.1 ab | 1.2 b | 2.8 ± 0.7 |
| F (mg·m·s$^{-1}$) | 17.1 a | 17.6 a | 17.2 a | 12.5 b | 16.1 ± 1.2 |
| F$_{10}$ (mg·m·s$^{-1}$) | 27.8 a | 26.6 a | 24.2 ab | 21.0 b | 24.9 ± 1.4 |
| F$_{b10}$ (mg·m·s$^{-1}$) | 19.7 ab | 21.7 a | 15.4 b | 16.9 b | 18.4 ± 1.4 |
| F$_{e10}$ (mg·m·s$^{-1}$) | 17.3 a | 16.6 a | 13.4 b | 16.8 a | 16.0 ± 1.0 |
| E (unitless) | 0.9 ab | 1.0 ab | 0.4 a | 1.7 b | 0.9 ± 0.2 |

Note: Event A, Event B and Event C are events with the single, double and multiple rainfall intensity
peaks, respectively; Others are the events that excluded from the categorization; RA, RD and RI are
rainfall amount, duration and interval, respectively; I and I$_{10}$ are the average and 10-min maximum
rainfall intensities, respectively; I$_{b10}$ and I$_{e10}$ are the average rainfall intensities in 10 min after rain begins
and before rain ends, respectively; F and F$_{10}$ are the average and 10-min maximum raindrop momentums,
respectively; F$_{b10}$ and F$_{e10}$ are the average raindrop momentums in 10 min after rain begins and before
rain ends, respectively; E is evaporation coefficient; Different letters indicate significant differences of
rainfall characteristics between event categories ($p<0.05$) (rows at the table).

**Table 3.** Stemflow variables of *C. korshinskii* and *S. psammophila* during rainfall events with different intensity peak amounts.

| Species | Stemflow variables | Event A | Event B | Event C | Others | Average |
|---|---|---|---|---|---|---|
| *C. korshinskii* | SFV (mL) | 134.1 a | 203.7 a | 560.8 b | 7.6 c | 226.6 ± 46.4 |
| | SFI (mm·h$^{-1}$) | 672.9 a | 552.4 b | 527.0 b | 317.8 c | 517.5 ± 82.1 |
| | SFI$_{10}$ (mm·h$^{-1}$) | 2849.0 a | 2399.3 a | 1809.1 b | 1173.2 c | 2057.6 ± 399.7 |
| | FR (unitless) | 109.4 a | 146.6 b | 137.9 b | 128.9 ab | 130.7 ± 8.2 |
| | TLG (min) | 67.3 ab | 56.2 a | 67.0 ab | 74.2 b | 66.2 ± 10.6 |
| | TLM (min) | 81.1 a | 75.5 a | 202.1 b | 78.8 a | 109.4 ± 20.5 |
| | TLE (min) | 22.3 a | 18.7 b | 18.5 b | 20.6 a | 20.0 ± 5.3 |
| | SFD (h) | 1.4 a | 3.1 a | 9.1 b | 1.4 a | 3.8 ± 0.8 |
| *S. psammophila* | SFV (mL) | 102.6 a | 145.7 a | 435.2 b | 4.7 c | 172.1 ± 34.5 |
| | SFI (mm·h$^{-1}$) | 648.1 a | 421.5 b | 246.6 c | 153.2 c | 367.3 ± 91.1 |
| | SFI$_{10}$ (mm·h$^{-1}$) | 1672.7 a | 1582.8 a | 888.4 b | 384.7 c | 1132.2 ± 214.3 |
| | FR (unitless) | 77.1 a | 91.4 a | 129.1 b | 101.6 ab | 101.6 ± 10.4 |
| | TLG (min) | 84.9 a | 46.5 b | 56.1 b | 31.5 b | 54.8 ± 11.7 |
| | TLM (min) | 64.3 a | 93.4 a | 235.8 b | 88.4 a | 120.5 ± 22.1 |
| | TLE (min) | 17.1 a | 8.6 b | 20.8 a | 7.3 b | 13.5 ± 17.2 |
| | SFD (h) | 1.2 a | 3.4 a | 8.3 b | 0.7 a | 3.4 ± 0.9 |

Note: Event A, Event B and Event C are events with the single, double and multiple rainfall intensity peaks, respectively; Others are the events that excluded from the categorization; SFV is stemflow volume; SFI and SFI$_{10}$ are the average and 10-min maximum stemflow intensities at incident rains, respectively; FR is funnelling ratio of stemflow at incident rains; TLG and TLM are time lags of stemflow generating and maximizing after rains begin, respectively; TLE is time lag of stemflow ending after rain ceases; SFD is stemflow duration; Different letters indicate significant differences of stemflow variables between event categories ($p<0.05$) (rows at the table).

**Table 4.** Comparisons of stemflow intensity and funnelling ratio at different basal diameter
categories.

| Species and stemflow variables | | BD categories (mm) | | | | | |
|---|---|---|---|---|---|---|---|
| | | 5–10 | 10–15 | 15–18 | 18–25 | >25 | AVG |
| *C. korshinskii* | FR | 163.7±12.2a | 136±10.9b | 119.5±13.0b | 97.7±9.2b | NA | 131±8.2 |
| | SFI | 716.2±118.7a | 552.5±90.3b | 619±103.3b | 333.8±45.8b | NA | 553.9±82.1 |
| *S. psammophila* | FR | 212±17.4a | 84±6.4b | NA | 44.2±3.0b | 54.9±4.2b | 100.6±7.9 |
| | SFI | 738.7±160.9a | 360.7±82.7a | NA | 197.2±44.9b | 209.9±44.5b | 372.2±79.4 |

Note: SFI and FR are the average stemflow intensity and funnelling ratio at incident rains, respectively; BD is
branch basal diameter (mm); NA means not applicable; Different letters indicate significant differences of
stemflow variables between event categories ($p<0.05$) (rows at the table).

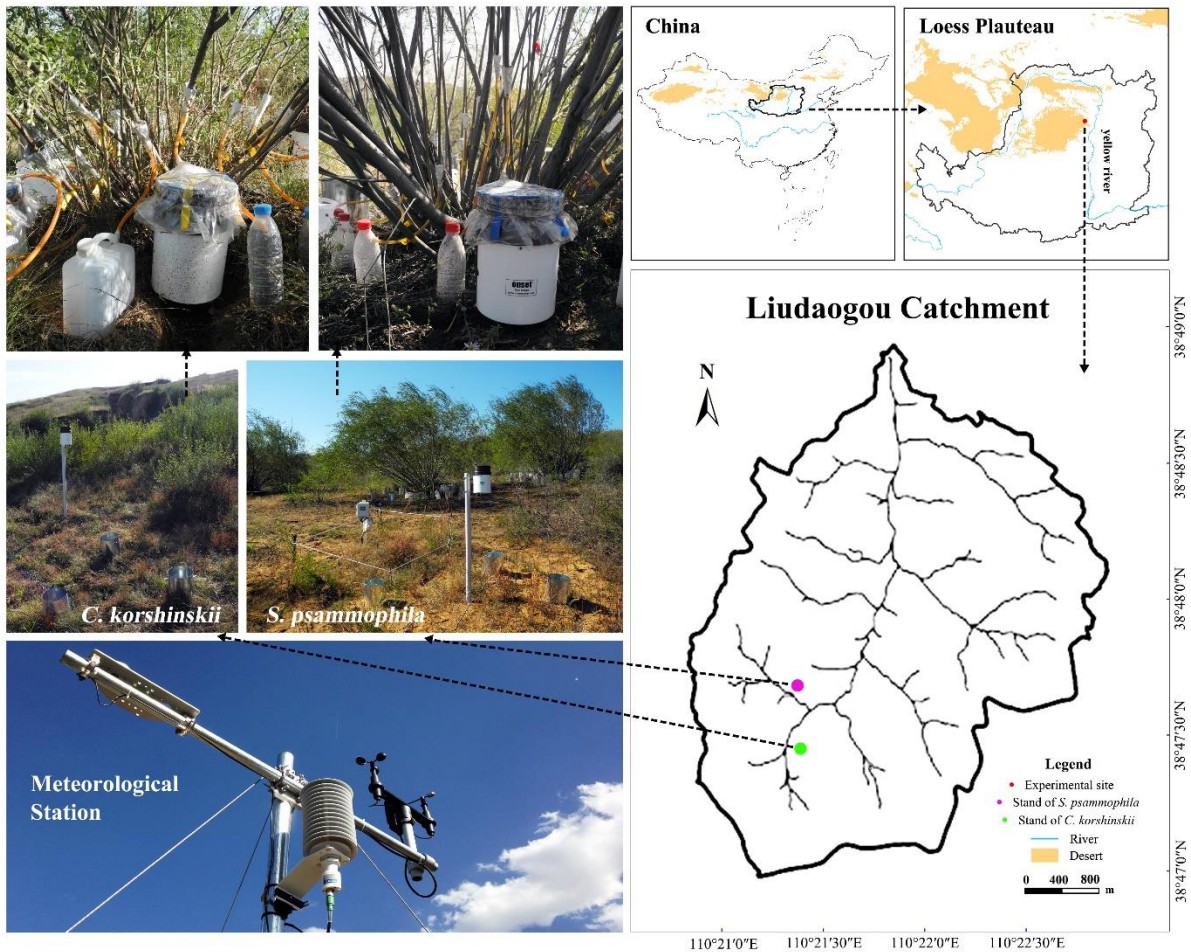


**Figure 1.** Locations and experimental settings in the plots of *C. korshinskii* and *S.*
*psammophila*.

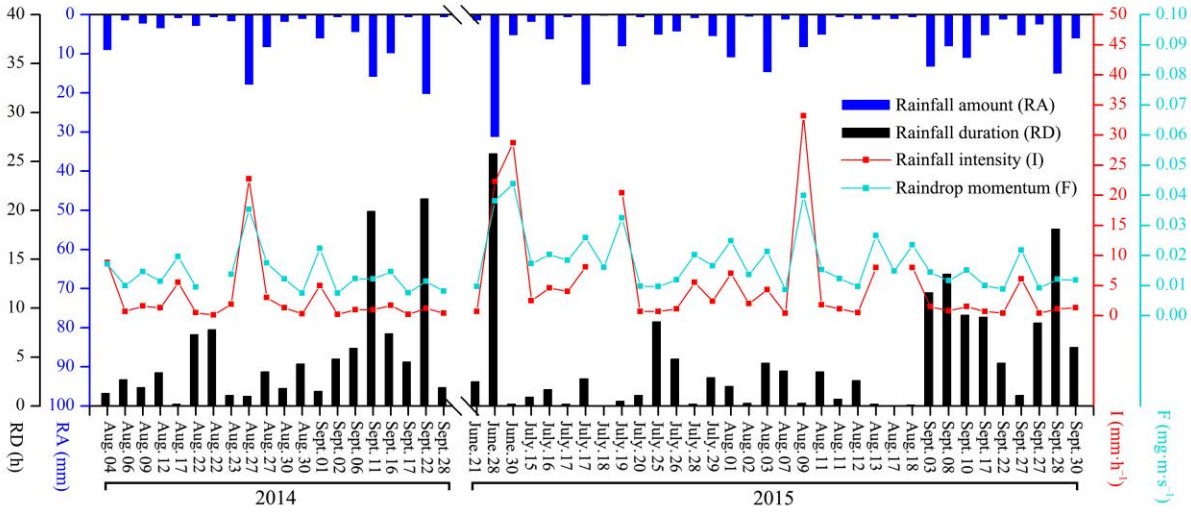


**Figure 2.** Inter-event variations in rainfall characteristics during the experimental period.

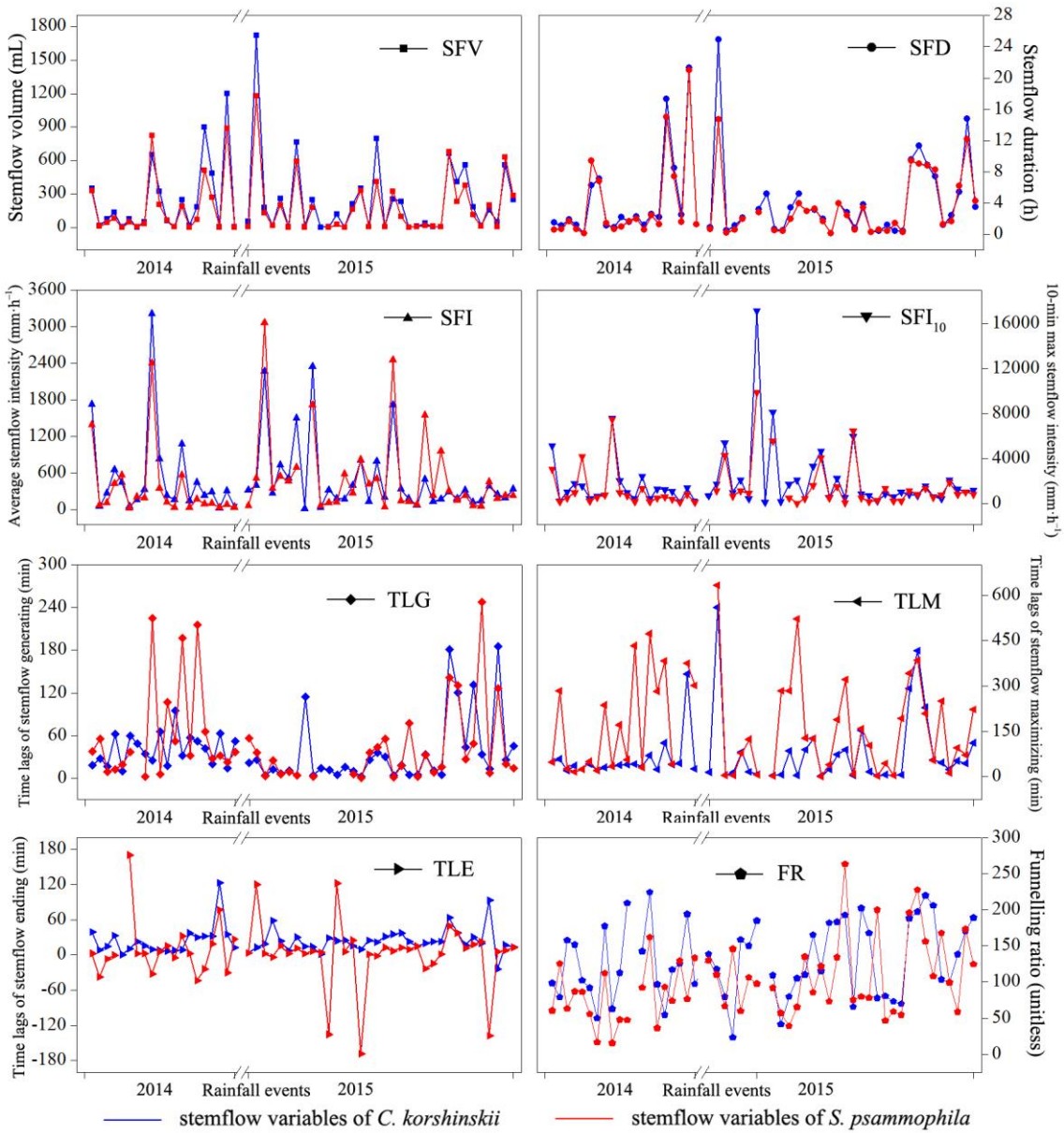


**Figure 3.** Inter-event variations in stemflow variables of *C. korshinskii* and *S. psammophila*

during the experimental period.

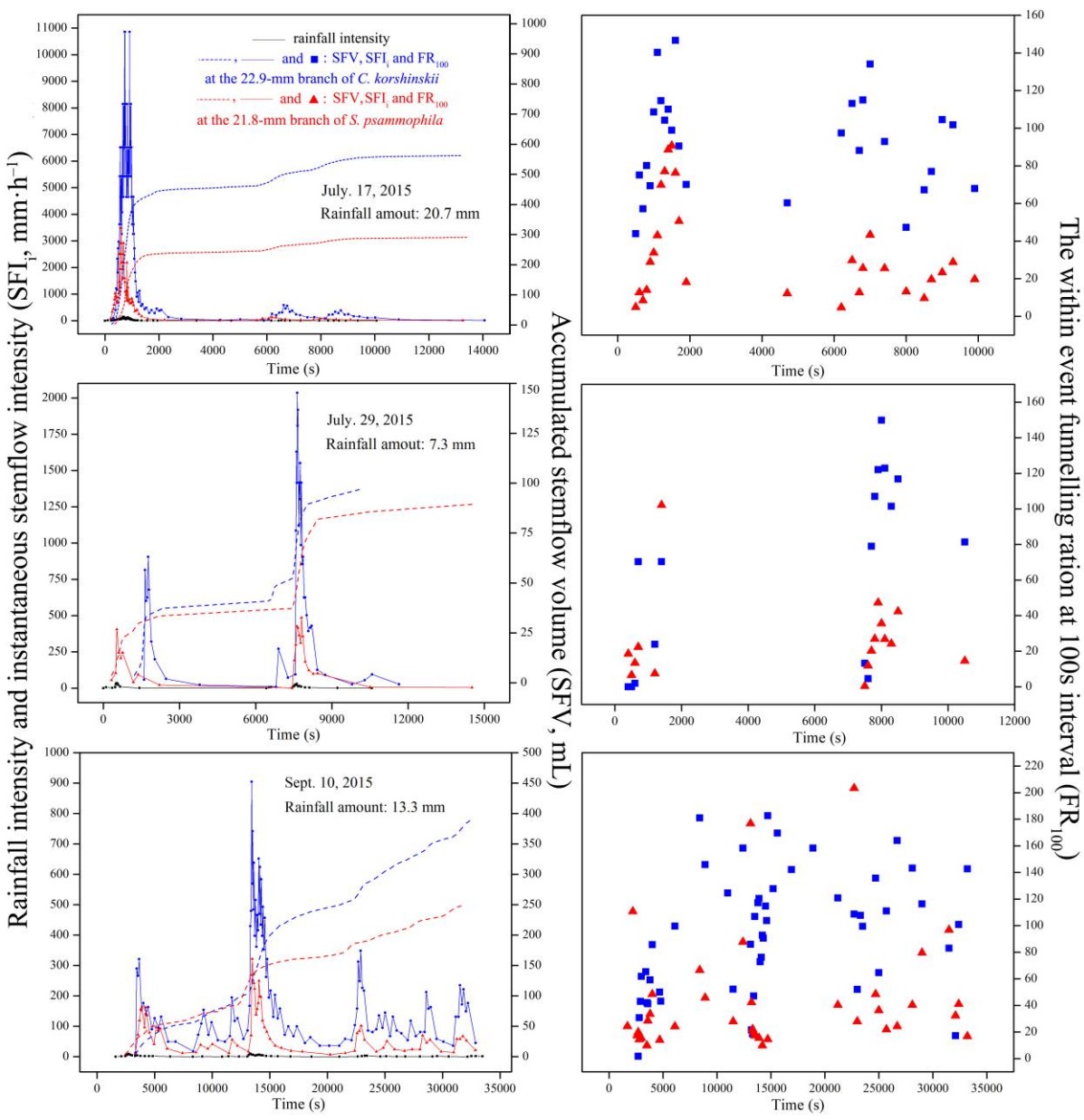


**Figure 4.** Stemflow synchronicity of *C. korshinskii* and *S. psammophila* to rains during

representative events with different rainfall-intensity peak amounts.

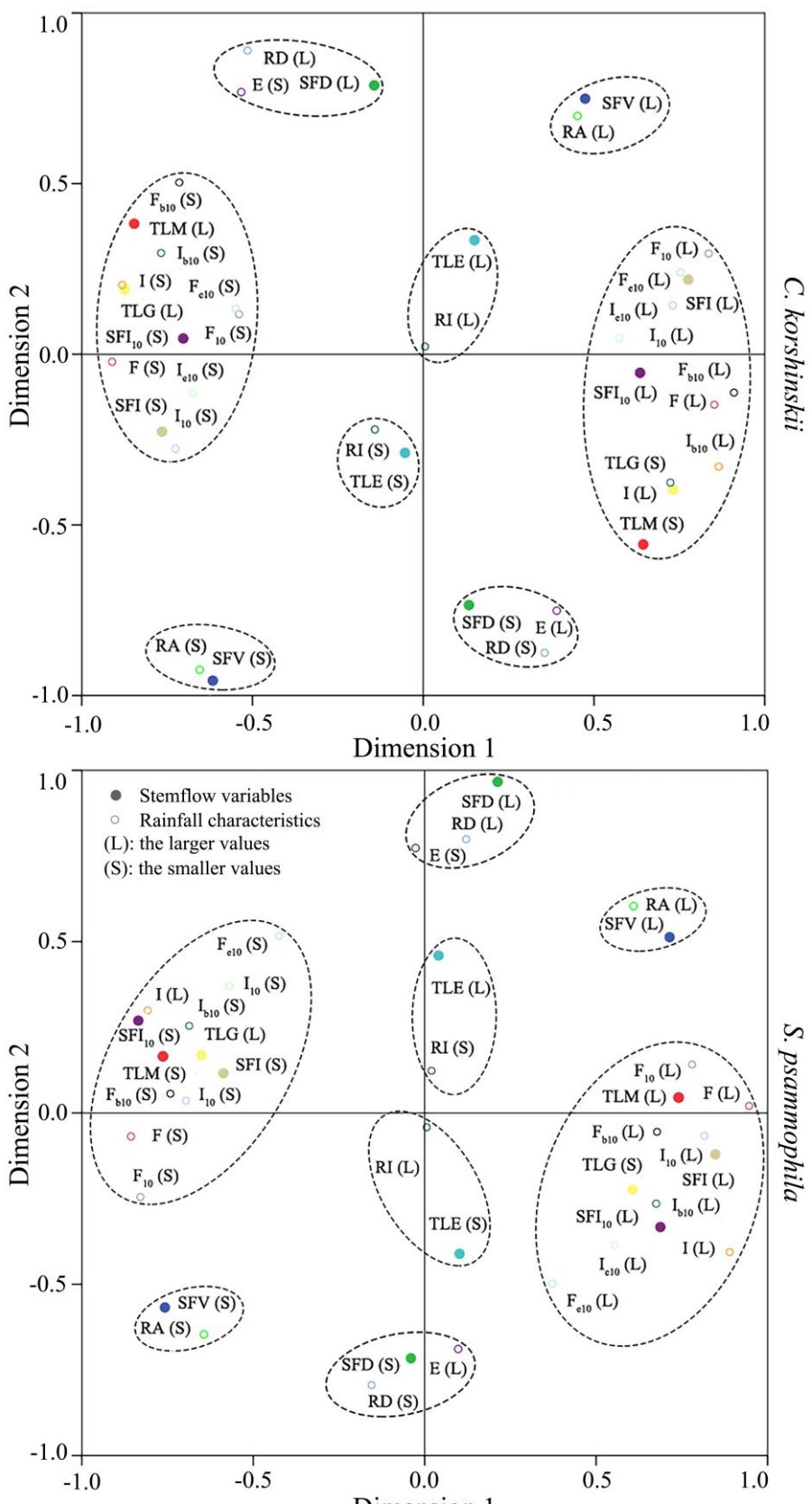


**Figure 5.** Correspondence maps of stemflow variables with rainfall characteristics for *C.*

*korshinskii* and *S. psammophila*.

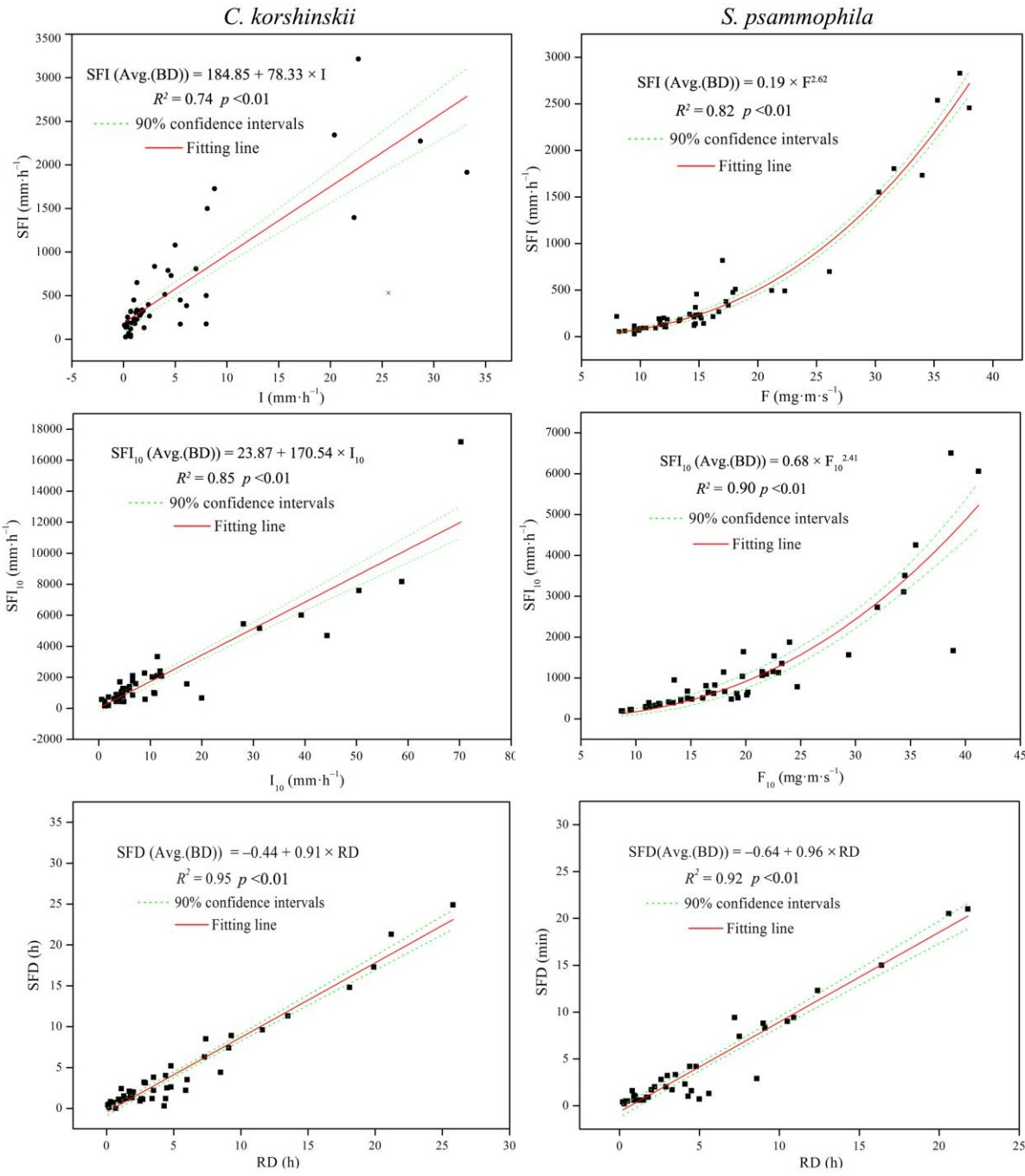


**Figure 6.** Relationships of stemflow intensity and duration with rainfall characteristics.

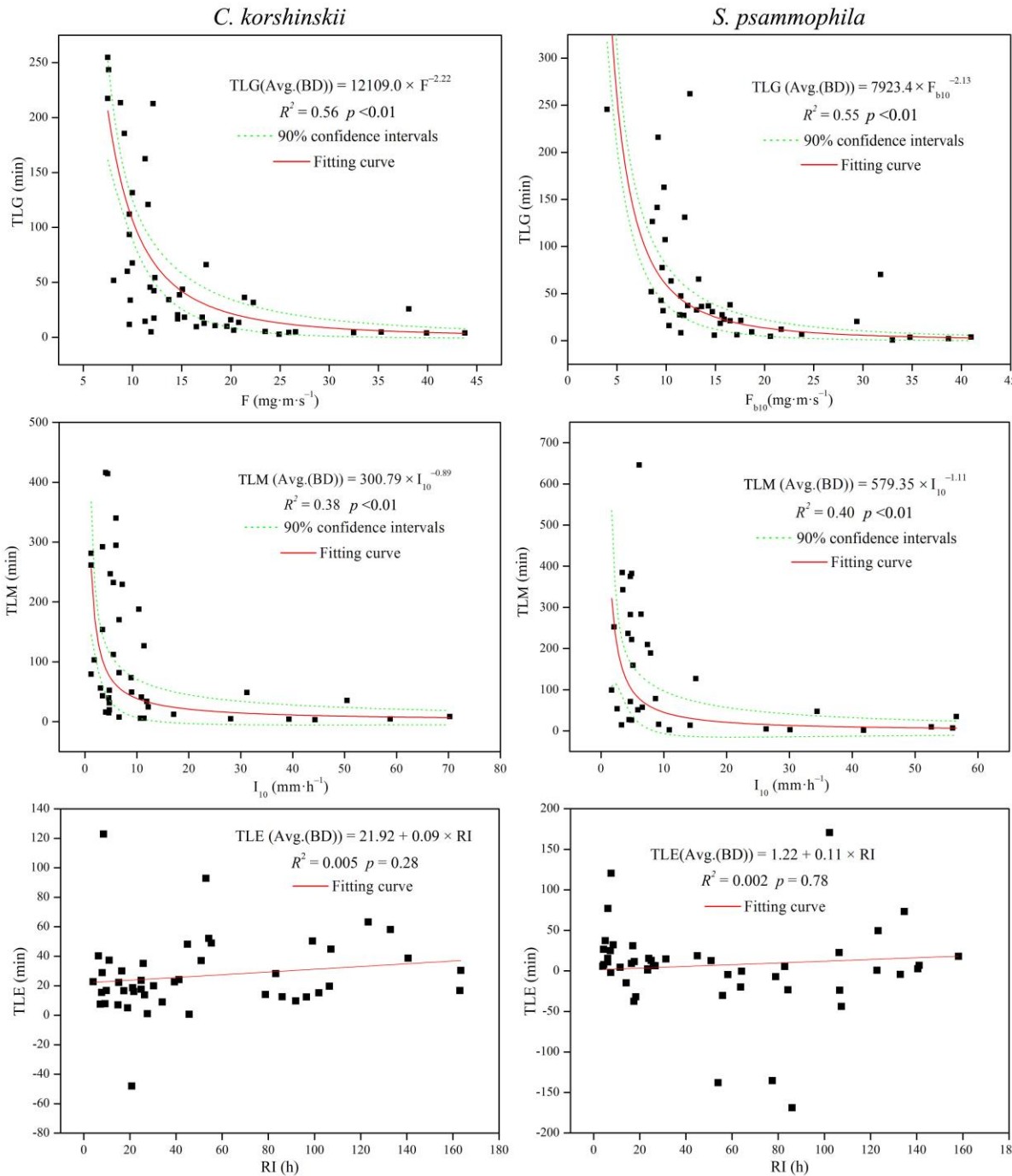


**Figure 7.** Relationships of stemflow time lags with rainfall characteristics.