# Peer review of "Temporal-dependent effects of rainfall characteristics on"

_Hydrology and Earth System Sciences, 2019_

## Referee Comment (RC1) · Anonymous Referee #1 · 9 Jun 2019

The paper by Yuan et al mainly aimed to characterize the inter-/intra-event stemflow dynamics of two xerophytic shrubs and to quantify their relationships with the corresponding inter-/intra-event rainfall characteristics. They concluded that rainfall characteristics had temporal-dependent influences on corresponding stemflow variables. From my point of view, the study has potential to make a contribution to a better understanding of, in particular, the intra-storm stemflow processes and the underlying mechanisms governing its dynamics. The experimental design and data analysis are generally acceptable, while clarity is needed in presenting the design. The figures adequately summarize the results. I recommend this paper for publication in HESS after some moderate revisions had been addressed by the authors.

[Figure]

Moderate/minor comments

1. L 69: Change "initialed" to "initiated".

2. L 72: I would use "leafed period" instead of "leaf period".

3. Section 2.2: What is the time interval for recording rainfall and the stemflow in subsequent section? This needs to be clearly stated.

4. L 184-186: According to Table 1, stemflow data of S. psammophila are not available for branches with a BD of 15-18 mm rather than 18-25 mm. Please verify this.

5. Section 2.4: I miss the information about how many rain gauges the authors used in recording stemflow. Did each branch connect to a rain gauge? It seems to be the case from my view of Fig. 1, which makes a total of 14 rain gauges. Please explicitly state to avoid guessing.

6. L 203: I would change "base area" to "orifice area", which is a more accurate terminology for rain gauge.

7. L 200-210: As for mL of SFV, it should be calculated as: SFV = [mm (branch stem-flow recorded by tipping-bucket rain gauges) / 10] × cm2 (orifice area of a rain gauge). I think the authors missed a 10. Therefore, for the calculation of stemflow volume and stemflow intensity, I suggest that authors provide the corresponding mathematical equations; it would be concise and easier for readers to follow.

8. L 211-215: According to the calculation of TLG, TLM, and TLE, these variables can have either negative or positive values. I encourage the authors to clarify here their respective meanings, i.e., what positive values are suggesting and what negative values are suggesting. Again, it would be easier for readers to better understand their following results.

9. L 258-259: It would be more straightforward to add a row in Table 2 showing how many rainfall events occurred for each category (Event A to C, and others).

10. L 291-298: If it is possible, I would also expect to see some results about the differences of stemflow variables varied among BD categories.

11. Section 4.1: I would like to discuss with the authors about the use and importance of stemflow intensity and RSFI. I admit that stemflow intensity would be a good variable to show the dynamics of intra-event stemflow, while I am not convinced by authors about the importance of comparing the absolute values of stemflow intensity versus rainfall intensity (also demonstrated in L26-30 of Abstract). Their study is based on monitoring branch stemflow, and branch stemflow intensity was a bit higher than rainfall intensity in their study. However, in terms of stemflow's ecological and hydrological importance such as in providing additional soil water and sustaining vegetation growth, we pay more attention to the whole tree/shrub (rather than a single branch). From my understanding this variable is highly dependent on the size of a shrub/tree, because a lager shrub/tree (normally has larger basal diameter or canopy area) would generate substantially higher volume of stemflow, therefore stemflow intensity calculated based on collecting from individual trees/shrubs would be far greater than rainfall intensity, as examples please see Fig. 3 in cayuela et al. (2018, Journal of Hydrology) or Fig. 7 in Germer et al. (2010, Journal of Hydrology). Stemflow and rainfall differs in their paths entering into rain gauges; the orifice area makes sense for rainfall because this area is precisely where rainfall falling into and rainfall depth is then normalized, while stemflow is part of intercepted rainfall by the canopy and then comes down stems, which indicates that infiltrating soil area of stemflow is quite different than that of a rain gauge (i.e., orifice area). Therefore this variable may be prone to underestimate stemflow's eco-hydrological role for small shrubs, as such, in terms of ecological importance this variable seems to be less appropriate to be used for inter-specific comparison or even intra-specific comparison of varying sizes. Moreover, the authors were also recommending a future combination use of funnelling ratio and RSFI in stemflow studies. While I agree with the authors that RSFI is helpful in better understanding of the intra-event convergence effects, funnelling ratio assumes trunk/stem basal area is the true area that stemflow is delivered to the soil, whereas RSFI here is based on stemflow
intensity which I have discussed above. RSFI may also be prone to underestimate stemflow's eco-hydrological role for small trees/shrubs while overestimate that of big trees/shrubs. I encourage authors to discuss both the advantages and limitations of stemflow intensity and RSFI as well as their application.

12. L 433-437: These sentences are somewhat redundant (have been mentioned in above sections) and can be simplified or simply deleted.

13. Figure 3: Data points are average values for 7 branches for each event? Since the authors selected 7 branches of varying BD for each species to measure stemflow, a relative larger difference in stemflow would be expected among branches. It would be an option to adding error bars if they won't make the figure blurring too much.

14. Figure 4: The unit of rainfall stemflow intensity should be mm h-1 rather than m h-1. Also changes should be made in the legend, since both lines and points are included in this figure, it would be misleading by labelling "Lines in blue" or "Lines in red" without mentioning points. Moreover, since 7 branches for each species were selected for monitoring stemflow intra-event dynamics, I am wondering which branches for two species were demonstrated in this figure.

---

## Referee Comment (RC2) · David Dunkerley (Referee) · 10 Jun 2019

The authors report on a detailed study of stemflow in two dryland shrub species, and its relationship with rainfall properties. The data come from field observations of selected branches that were equipped with stemflow collecting collars, and exposed to a number of natural rainfall events. Seven branches were instrumented for each of the two shrub species. The stemflow was recorded by directing the flow into tipping-bucket rain gauges having a 0.2 mm sensitivity.

Although the work appears to be generally thorough, there are some significant issues with it that I consider require clarification before the work could be accepted for

publication.

The authors are concerned with the relative timing of rainfall and of the resulting stem-flow. The difficulty here is that the relative timing is affected by the size of the collecting areas that contribute either rainfall or stemflow to the measuring gauges. The canopy of S. psammophila for instance is reported as 21.4 m2 (line 170), whilst the collecting area of the pluviography TBRG in the open is just 0.018 m2. Thus the canopy area of the shrub is more than 1,000 times larger. Therefore, the tiny tipping bucket (capacity about 3.65 mL, by my estimation) can potentially be filled more rapidly by stemflow than by rainfall in the open. In this way, the time until first tip (regarded by the authors as the onset of stemflow) probably occurs closer to the onset of rainfall as a function of canopy area and its effect in reducing the bucket filling time. Therefore, among the seven instrumented branches, the timing of stemflow initiation should vary, and it might be possible to relate this to the plant morphology. However, the authors do not report the canopy collecting area for the 7 branches that they monitored for each of the two shrub species. Therefore, calculations of the kind just sketched cannot be made nor the results evaluated properly. This imposes uncertainty in the interpretation of the stemflow timing data. The ideal, of course, would be for the collecting area of foliage and branch to be as close as possible to the collecting area of the open-field raingauge.

Indeed, the manuscript lacks any detail of the foliar area on the branches that were monitored for stemflow. For instance, leaf area and leaf wettability are not mentioned or reported. Likewise, there are no data on the shrub canopies as a whole, such as leaf area index (LAI) or canopy gap fraction. The lack of such information again makes the results somewhat difficult to interpret or to compare with results from other taxa and environments.

Data processing is poorly explained. Stemflow intensity, given in mm h-1, requires that the volume of water delivered to the TBRG used to record stemflow (recorded in mL per bucket tip) must be associated with the area over which the equivalent stemflow depth is evaluated. I could not see this explained anywhere in the manuscript, and it needs

to be made clear. If it was the cross-sectional area of the branch being monitored (typically about 3 cm2 by my rough estimation) then this needs to be set out in the manuscript. If the authors did use basal branch cross-sectional area, then of course the stemflow intensity can easily exceed the rainfall intensity, as a function of the very small area over which the stemflow is recorded as arriving - far smaller than the collecting area of the rainfall pluviograph. If this area were to be doubled, then the stemflow intensity would be halved (and so on). Therefore, the area used by the authors in their calculation needs to be stated (and justified by some relationship to plant water availability).

Data processing is also poorly explained in terms of the data on stemflow volume presented by the authors (e.g. in Table 3). Are the stemflow volumes reported there, and discussed at many places in the paper, the sum of the stemflow on the 7 monitored branches, or the arithmetic mean of the stemflow from the 7 branches, or are the figures scaled-up to estimate the stemflow delivered by the entire test shrub? (The test shrubs had a total of 180 and 261 branches (line 173) only 7 of which were monitored for each shrub species (amounting to a sample of $\sim$ 4% and $\sim$2.6% of the branches, the adequacy of which is not discussed by the authors). Whatever the authors did, it is not made clear and this needs to be corrected. Especially in relation to stemflow, all relevant parameters used in data processing must be set out clearly and systematically.

Without knowing the details of the calculation procedure, the relative intensity of the stemflow and the open-field rainfalls are difficult to interpret. No formulae are presented by the authors that would allow this to be checked. My own feeling is that the stemflow flux would be a more useful figure - that is, the flow rate delivered to the base of the branch, expressed for instance in mL/minute or L/hour. If this is accompanied by a clearly-stated area over which the flow is tallied, then a stemflow intensity can be calculated.

In summary, what I find to be missing from the manuscript includes

- some discussion of why 7 stems were studied and whether this is a sufficient sample
- some consideration of the filling time of the buckets in the tipping-bucket gauges used
for rainfall and stemflow measurement, and the effect of this on the lag time before the
start of stemflow (and the cessation of stemflow after rain ends) - more detail on the
shrubs - including the variability of canopy size etc across the population from which
the two sample shrubs were drawn, and some information on leaf area and wettability,
if available - a proper accounting of how stemflow flux was calculated and how the area
over which the intensity was scaled was selected.

David Dunkerley Monash University

More detailed comments:

lines 49-50: it is difficult to generalise from these few data to all "water stressed regions"
(and need to define what a water-stressed region is)

line 57: mL/g of what? biomass?

line 61: a flow in units of mL/min is a flux, not a speed

line 69: should presumably say 'not until AFTER canopies became saturated'

line 70: need to define RA when this contraction is first used. It is used again in line
138 before being defined.

line 76: missing a space before 0.4

lines 77-78: need to include branch surfaces also

line 83: need to state which measure is maximised

line 85: explain why time lags are important: presumably the last stemflow would oc-
cur as a very small (negligible) flux, so why is the timing of the last stemflow impor-
tant? More generally, the authors could say something about why the time variation of
stemflow during rainfall is important. Do peaks of stemflow flux exceed soil infiltration

capacity, perhaps? Otherwise, why is this important?

line 100: no need to repeat the number of rainfall events here, and again in line 222 and again in line 248. Once is sufficient.

line 106: please define 'stemflow intensity' and provide a formula somewhere in the paper

line 139: please explain what 'analogue' means here

lines 147-148: all these timing data are a function of the tipping-bucket filling time (see discussion earlier in this report). When using a TBRG, it is difficult to tell precisely when rain begins or ends, owing to the time that might be required to fill the first tipping-bucket.

line 153: how Is raindrop morphology reflected in this? please explain

line 160: why is mean intensity used here?

line 168: since this paper reports a study of branch stemflow only, the title of the paper should be amended to indicate this clearly (i.e., not a study of stemflow on an entire plant)

line 171: to what extent were the studied shrubs representative of the wider population? please present some data.

lie 181: please explain what is meant by 'canopy skirt locations'. The photos suggest that there were many overhanging leaves and branches. Some of the stemflow collars were placed quite high off the ground (as far as can be judged from the photos, as no quantitative information on this is included in the paper). How do the authors know that the stemflow at these heights would actually reach the ground, and not drip off the branches?

line 189-190: what was the external diameter? this should be included as the dimensions of the stemflow collars are critical - it does not seem sufficient simply to assert

that they caught no rainfall or released drips of throughfall from above.

line 270: how were rainfall intensity peaks identified? What makes one peak an intensity peak?

line 292: is the reference to the volume from a single branch or the total from the 7 branches?

lines 300-310: this is difficult to read, owing to the need to recall the meaning of the very many contractions. Some reminders of what these mean would be useful here.

line 342: a stemflow intensity of 1232 mm h-1 is large. What was the flux? I presume that in the case of the authors own work in the present study, the flux was within the capacity of the tipping-bucket gauges (typically a few hundred mm h-1 at maximum) since the rainfall was not very intense. Some comment on this would be worthwhile.

lines 383-384: but these fluxes would surely depend on the antecedent leaf and branch wetness, and on meteorological conditions such as wind speed and vapour deficit (the latter is not reported, incidentally).

Table 2: why are only 3 rainfall events listed here? More than 40 more are simply lumped under "others" and no details are provided. Why?

Figure 4 shows units of m/h which I presume should be mm/h

---

## Referee Comment (RC3) · Anonymous Referee #3 · 11 Jun 2019

After careful review, I think, in many ways, this is a good manuscript. The work has been well done and the manuscript is well organized. The paper has an appropriate length and the topic is of interest to the general readers of HESS. My major concern is the reasonability of the stemflow variables used in this study. For instance, in Line 207, the authors said that the average (SFI) and 10-min maximum (SFI10) stemflow intensities were calculated by the branch stemflow as recorded by the tipping-bucket rain gauges (mm) and rainfall duration (h). In my opinion, stemflow intensities should be defined as the branch stemflow depth (which can be calculated from branch stemflow volume as divided by branch basal area) in a certain time. In the current form, the authors underestimated stemflow intensities. Also, in Line 216, the ratio of the intra-

event stemflow intensity (RSFI, unitless) should be calculated basing on the suggested calculation of stemflow intensity. I recommend this manuscript for publication after a minor revision. I also state minor comments as follows. L1: Only seven branches were used to measure stemflow for each shrub species (The studied shrubs had a total of 180 and 261 branches), So the suggested title is: Temporal-dependent effects of rainfall characteristics on inter-/intra-event branch-scale stemflow variability in two xerophytic shrubs. L220-226: It could be better if the authors provide the formula for each stemflow variables. L658. Table 1: What is the standard for base diameter (BD) categorization? In the current form, the class interval (5-10, 10-15, 15-18, 18-25, >25 mm) is variable. Why not 5-10, 10-15, 15-20, 20-25, and >25 mm? Please explain it. L662. Table 2: Do the rainfall indicators including RA, RD, RI, I, I10, Ib10 etc differ statically significantly among Event A, Event B, Event C and Others? Please provide the ANVOA results here. L670. Table 3: The comment is the same with the last one. Please provide the statistical results to depict the difference in the stemflow variables among Event A, Event B, Event C and Others.

---

## Author Comment (AC1) · 19 Aug 2019

Please see Response to Reviewer #1 at the attached supplement file for the detailed response by the authors.

General Comments: The paper by Yuan et al mainly aimed to characterize the inter-/intra-event stemflow dynamics of two xerophytic shrubs and to quantify their relationships with the corresponding inter-/intra-event rainfall characteristics. They concluded that rainfall characteristics had temporal-dependent influences on corresponding stemflow variables. From my point of view, the study has potential to make a contribution to a better understanding of, in particular, the intra-storm stemflow processes and the

underlying mechanisms governing its dynamics. The experimental design and data analysis are generally acceptable, while clarity is needed in presenting the design. The figures adequately summarize the results. I recommend this paper for publication in HESS after some moderate revisions had been addressed by the authors.

Reply: We appreciated the anonymous reviewer for the comments and suggestions, which were of great help to improve the overall quality of this manuscript. The manuscript had been carefully revised, and we tried best to submit a qualified manuscript as required.

R1C2: L 69: Change "initialed" to "initiated".

Reply: Done (Line 73, Page 4).

R1C3: L 72: I would use "leafed period" instead of "leaf period".

Reply: Done (Line 77, Page 4).

R1C4: Section 2.2: What is the time interval for recording rainfall and the stemflow in subsequent section? This needs to be clearly stated.

Reply: Sensors were installed at the meteorological station to record wind speed (Model 03002, R. M. Young Company, USA), air temperature and relative humidity (Model HMP 155, Vaisala, Finland). They were logged at 10-min intervals by a data-logger (Model CR1000, Campbell Scientific Inc., USA) (Lines 142–146, Page 7). We recorded stemflow and rainfall via the Onset$^{®}$ (Onset Computer Corp., USA) RG3-M tipping-bucket rain gauges (hereinafter referred to as TBRG). When the bucket (with resolution of 0.2 mm and the equivalent volume of 3.73 mL) was filled and tipped, data of stemflow or rainfall was stored at the dynamic time interval. It depended on rainfall and stemflow intensities. In general, we recorded meteorological features of WS, T and H at 10-min intervals. However, the rainfall and stemflow was recorded at dynamics intervals between neighboring tips with the fixed 0.2-mm resolution (Lines 221–222, Page 10).

R1C5: L 184-186: According to Table 1, stemflow data of S. psammophila are not available for branches with a BD of 15-18 mm rather than 18-25 mm. Please verify this.

Reply: The typo here of "18-25 mm" had been revised to "15-18 mm" at Line 213, Page 10.

R1C6: Section 2.4: I miss the information about how many rain gauges the authors used in recording stemflow. Did each branch connect to a rain gauge? It seems to be the case from my view of Fig. 1, which makes a total of 14 rain gauges. Please explicitly state to avoid guessing.

Reply: TBRGs had been applied in this study to automatically record stemflow volume and timing. Each TBRG connected to one experimental branches of C. korshinskii and S. psammophila. Seven branches were selected at different BD categories for each species. Therefore, we had installed 14 TBRGs for stemflow measuring in this study. It had been clearly described at the revised manuscript (Lines 220, Page 10).

R1C7: L 203: I would change "base area" to "orifice area", which is a more accurate terminology for rain gauge.

Reply: Done (Line 234, Page 11).

R1C8: L 200-210: As for mL of SFV, it should be calculated as: SFV = [mm (branch stem- flow recorded by tipping-bucket rain gauges) / 10] cm2 (orifice area of a rain gauge). I think the authors missed a 10. Therefore, for the calculation of stemflow volume and stemflow intensity, I suggest that authors provide the corresponding mathematical equations; it would be concise and easier for readers to follow.

eply: Thank you for commenting on the poorly explained data processing at this manuscript. At the previous version of this manuscript, we just gave the factors for calculating stemflow volume (SFV, mL), i.e., stemflow depth recorded by TBRG (SFRG, mm) and orifice area (186.3 cm2). The equation for SFV computation had been described at the revised manuscript (Equation 10) (Lines 235, Page 11). Besides, the definitions and calculations of stemflow intensity (Equation 11–13, Lines 246–248, Page 12), time lags to rains (Lines 252–257, Page 12) and other meteorological features (Equation 1–9, Lines 158–160, Line 164, Lines 184–188, Pages 8–9) had also been clearly described at section 2.2 Meteorological measurements and calculations and Section 2.4 Stemflow measurements and calculations.

R1C9: L 211-215: According to the calculation of TLG, TLM, and TLE, these variables can have either negative or positive values. I encourage the authors to clarify here their respective meanings, i.e., what positive values are suggesting and what negative values are suggesting. Again, it would be easier for readers to better understand their following results.

Reply: Thank you for this comment. Associated with the results in this study, the meanings of positive and negative values of TLG, TLE and TLM had been described at the Section 3.2 Stemflow volume, intensity, funnelling ratio and temporal dynamics at the revised manuscript. During the 54 events, no negative values were observed for TLG and TLM but TLE. It indicated that stemflow generally initiated and maximized after rains started for both species. However, stemflow might be ended before (negative TLE) and after (positive TLE) rains ceased. (Lines 326–329, Page 15).

R1C10: L 258-259: It would be more straightforward to add a row in Table 2 showing how many rainfall events occurred for each category (Event A to C, and others).

Reply: Done (Line 808, Page 40).

R1C11: L 291-298: If it is possible, I would also expect to see some results about the differences of stemflow variables varied among BD categories.

Reply: Thank you for this comment. As suggested, we compared SFI and FR at different BD categories of C. korshinskii and S. psammophila. Shown at Table 4, FR of C. korshinskii decreased from 163.7 at the 5–10-mm branches to 97.7 at the 18–25-

mm branches. The decreasing trend of FR were also noted for S. psammophila in the range of 44.2–212.0, as branch size increased. The results were in consistence with the findings for trees and babassu palms in an open tropical rainforest in Brazil (Germer et al., 2010), in the coastal British Columbia forest with mixed species (Spencer and Meerveld, 2016), for trees (Pinus tabuliformis and Armeniaca vulgaris) and shrubs (C. korshinskii and S. psammophila) at Loess Plateau of China (Yang et al., 2019). Because funnelling ratio was calculated as the ratio between stemflow and rainfall intensities, SFI was also compared at different BD categories. It was negatively related with branch size for both species. As indicated at Equation 14–15 (Lines 264–265, Page 12), the decreasing stemflow intensity with branch size might partly explained the negative relations between funnelling ratio and BD. However, we did not compare all the stemflow variables at different BD categories. Because of the high expense of TBRGs (Turner et al., 2019), no more than two branches were selected for stemflow recording at each BD category. The results were much more convincing to analyze the average stemflow variables among BD categories, and compared them at different rainfall amount categories with enough events for meeting the statistical significance.

R1C12: Section 4.1: I would like to discuss with the authors about the use and importance of stemflow intensity and RSFI. I admit that stemflow intensity would be a good variable to show the dynamics of intra-event stemflow, while I am not convinced by authors about the importance of comparing the absolute values of stemflow intensity versus rainfall intensity (also demonstrated in L26-30 of Abstract). Their study is based on monitoring branch stemflow, and branch stemflow intensity was a bit higher than rainfall intensity in their study. However, in terms of stemflow's ecological and hydrological importance such as in providing additional soil water and sustaining vegetation growth, we pay more attention to the whole tree/shrub (rather than a single branch). From my understanding this variable is highly dependent on the size of a shrub/tree, because a lager shrub/tree (normally has larger basal diameter or canopy area) would generate substantially higher volume of stemflow, therefore stemflow intensity calculated based on collecting from individual trees/shrubs would be far greater

than rainfall intensity, as examples please see Fig. 3 in cayuela et al. (2018, Journal of Hydrology) or Fig. 7 in Germer et al. (2010, Journal of Hydrology). Stemflow and rainfall differs in their paths entering into rain gauges; the orifice area makes sense for rainfall because this area is precisely where rainfall falling into and rainfall depth is then normalized, while stemflow is part of intercepted rainfall by the canopy and then comes down stems, which indicates that infiltrating soil area of stemflow is quite different than that of a rain gauge (i.e., orifice area). Therefore this variable may be prone to underestimate stemflow's eco-hydrological role for small shrubs, as such, in terms of ecological importance this variable seems to be less appropriate to be used for inter-specific comparison or even intra-specific comparison of varying sizes. Moreover, the authors were also recommending a future combination use of funnelling ratio and RSFI in stemflow studies. While I agree with the authors that RSFI is helpful in better understanding of the intra-event convergence effects, funnelling ratio assumes trunk/stem basal area is the true area that stemflow is delivered to the soil, whereas RSFI here is based on stemflow intensity which I have discussed above. RSFI may also be prone to underestimate stemflow's eco-hydrological role for small trees/shrubs while overestimate that of big trees/shrubs. I encourage authors to discuss both the advantages and limitations of stemflow intensity and RSFI as well as their application.

Reply: Thank you for commenting on the calculation and importance of stemflow intensity and RSFI at this manuscript. It indeed underestimated the eco-hydrological significance of stemflow by ignoring its receiving area of branch base as suggested. Therefore, we had revised the calculation of stemflow intensity on basis of basal area, and introduced funnelling ratio to assess the convergence effect of stemflow at the revised manuscript. Please see the detailed explanations as below. (1) Stemflow intensity had been re-computed on basis of branch basal area, and quantitatively connected to funneling ratio. The RG3-M TBRGs had been applied to record stemflow in this study. Stemflow depth (SFRG, mm) could be directly computed with tip amounts and tip resolution of 0.2 mm. Similar with the interpretation for rainfall recording, the 0.2-mm per tip represented 200 mL water deposing on the 1-m2 ground surface. Based

at the same receiving areas, we calculated stemflow intensity as the ratio between SFRG and rainfall duration at the previous manuscript. However, it underestimated the eco-hydrological significance of stemflow by ignoring the limited area of trunk/branch base, over which stemflow was received. As suggested at this comment, stemflow intensity should associate with the area over which the equivalent stemflow depth is evaluated. Therefore, we re-calculated stemflow intensity and followed the definition of stemflow volume per basal area per unit time (Herwitz, 1986; Spencer and Meerveld, 2016). In this study, we calculated stemflow intensity at different time intervals, including the event base (SFI), the 10-min (SFI10) and the dynamic intervals between neighboring tips of TBRG (SFIi) (Equation 11–13) (Line 246–248, Page 12). Furthermore, we established the quantitative connections of stemflow intensity with funnelling ratio for the first time as indicated at Equation 14–15 (Lines 264–265, Page 12). RSFI had been deleted at the revised manuscript. By replacing the event-based volume of rainfall and stemflow with their intensities at the traditional expression (Herwitz, 1986), the new method enabled funnelling ratio to be computed at high temporal resolutions within event. (2) Stemflow variables and the meteorological influences were analyzed at branch scale. C. korshinskii and S. psammophila are modular organisms with multiple branches. Each branch of them lives as independent individual which seeks its own survival goals and compete with each other for light and water (Firn, 2004; Allaby, 2010). They provide ideal experimental objects to measure the branch stemflow volume and production processes. By introducing branch basal diameter (BD, mm) as intermediate variable, stemflow volume, intensity and funnelling ratio could be upscaled from branches to shrubs (Yuan et al., 2016; 2017). Therefore, the study on branch stemflow variables was conducive to explain the meteorological influences on stemflow at shrub scale particularly for the modular organisms. To guarantee the representativeness of experimental shrubs and branches, the thorough plot investigation had been carried out. Please see Point (3) at Reply to R2C3 for describing the determination of standard shrubs at the plots of C. korshinskii and S. psammophila, and see Point (4) at Reply to R2C2 for explaining the determination of standard branches

of the two shrubs. To address the branch scaled measurements of stemflow, the title had been revised as "Temporal-dependent effects of rainfall characteristics on inter-/intra-event branch-scaled stemflow variability in two xerophytic shrubs" as suggested by Reviewers 2 and 3.

R1C13: L 433-437: These sentences are somewhat redundant (have been mentioned in above sections) and can be simplified or simply deleted.

Reply: Done.

R1C14: Figure 3: Data points are average values for 7 branches for each event? Since the authors selected 7 branches of varying BD for each species to measure stemflow, a relative larger difference in stemflow would be expected among branches. It would be an option to adding error bars if they won't make the figure blurring too much.

Reply: Stemflow variables were averaged at seven branches of C. korshinskii and S. psammophila, respectively. Inter-event variations of the average stemflow variables during the experimental period had been shown at Figure 3. The relatively high expense of TBRGs limited the number of experimental branches that could be measured (Turner et al., 2019). However, each experimental branch was carefully selected following the strict criteria. Please see Point (4) at Reply to R2C2 for explaining the representativeness of the selected seven branches. A total of seven branches were selected for automatic recording via TBRGs at different BD categories of each species. That was the comprehensive results by balancing the statistical significance and TBRG expenses. To better meeting the statistical significance, we took the average value of stemflow variables at the seven branches at each species, and focused on the comparison of them among different rainfall amount categories. We just discussed the influence of rainfall characteristics in this study, and no analysises were performed to explore the influence of branch traits affecting stemflow volume and process. The variation of stemflow variables had been described as the average±standard error (Iida et al., 2017) at Table 3 (Lines 817–824, Page 41). However, since eight stemflow variables with 54 recording points each were shown at the same figure, the error bars were not drawn at Fig.3 just to keep the intra-event variation of stemflow variables clean and tidy (Lines 835–837, Page 45).

R1C15: Figure 4:The unit of rainfall stemflow intensity should be mm h-1 rather than m h-1. Also changes should be made in the legend, since both lines and points are included in this figure, it would be misleading by labelling "Lines in blue" or "Lines in red" without mentioning points. Moreover, since 7 branches for each species were selected for monitoring stemflow intra-event dynamics, I am wondering which branches for two species were demonstrated in this figure.

Reply: Done. The typo unit (m h-1) had been corrected to mm h-1, and the misleading legends had been revised, and the branch size of C. korshinskii and S. psammophila had been added at Fig.4 (Line 837–840, Page 46).

Please also note the supplement to this comment:
https://www.hydrol-earth-syst-sci-discuss.net/hess-2019-254/hess-2019-254-AC1-supplement.pdf

---

## Author Comment (AC2) · 19 Aug 2019

Please see Response to Reviewer #3 at the attached supplement file for the detailed response by the authors.

General Comments: After careful review, I think, in many ways, this is a good manuscript. The work has been well done and the manuscript is well organized. The paper has an appropriate length and the topic is of interest to the general readers of HESS. . .I recommend this manuscript for publication after a minor revision.

Reply: We appreciated the anonymous reviewer for the comments and suggestions.

[Figure]

This manuscript will be carefully revised as suggested prior to being submitted.

R3C1: My major concern is the reasonability of the stemflow variables used in this study. For instance, in Line 207, the authors said that the average (SFI) and 10-min maximum (SFI10) stemflow intensities were calculated by the branch stemflow as recorded by the tipping-bucket rain gauges (mm) and rainfall duration (h). In my opinion, stemflow intensities should be defined as the branch stemflow depth (which can be calculated from branch stemflow volume as divided by branch basal area) in a certain time. In the current form, the authors underestimated stemflow intensities. Also, in Line 216, the ratio of the intra-event stemflow intensity (RSFI, unitless) should be calculated basing on the suggested calculation of stemflow intensity.

Reply: Thank you for commenting on the calculation of stemflow variables in this study. As suggested at this comment, it indeed underestimated the eco-hydrological significance of stemflow to compute stemflow intensity by ignoring the limited area of branch base, over which stemflow was received. Therefore, we had re-computed stemflow intensity following the definition as stemflow volume per basal area per unit of time (Herwitz, 1986; Spencer and Meerveld, 2016). It had been calculated at different time intervals, including the event (SFI, mm·h−1), 10-min (SFI10, mm·h−1) and dynamic time interval between neighboring tips (SFIi, mm·h−1). Besides, RSFI had been deleted, and funnelling ratio had been introduced to assess the convergence effect of stemflow at the revised manuscript. It had been quantitatively connected with stemflow intensity for the first time as indicated at Equations 14–15 (Lines 264–265, Page 12). Please see the detailed explanation at Point (1) of Reply to R1C12, and Point (1) of Reply to R2C2.

R3C2: I also state minor comments as follows. L1: Only seven branches were used to measure stemflow for each shrub species (The studied shrubs had a total of 180 and 261 branches), So the suggested title is: Temporal-dependent effects of rainfall characteristics on inter-/intra-event branch-scale stemflow variability in two xerophytic shrubs. Reply: Done.

R3C3: L220-226: It could be better if the authors provide the formula for each stemflow variables.

Reply: Done. The detailed descriptions and calculations of stemflow variables had been stated at the revised manuscript, including stemflow volume (SFV, mL) (Equation 10) at Line 235, Page 11, stemflow duration (SFD, h), time lags stemflow generation (TLG, min), maximization and ending (TLE, min) at Lines 249–257, Page 12, stemflow intensities at the event bases (SFI), the 10-min interval (SFI10) and the dynamic intervals between neighboring tips of TBRG (SFIi) (Equation 11–13) at Lines 246–248, Page 12, funnelling ratio at event base (FR) and the 100-s (FR100) intervals (Equation 14–15) at Lines 264–265, Page 12.

R3C4: L658. Table 1: What is the standard for base diameter (BD) categorization? In the current form, the class interval (5–10, 10–15, 15–18, 18–25, >25 mm) is variable. Why not 5-10, 10-15, 15-20, 20-25, and >25 mm? Please explain it.

Reply: Thanks for this comment. Based on the plot investigation for C. korshinskii and S. psammophila, standard shrubs canopies could be determined. Four shrubs and 1 shrub had been selected for stemflow measurements and allometric equations establishments. By measuring branch morphologies at all the branches at these five shrubs of each species, BD categories was determined to guarantee the minimum branch amount at each category for meeting the statistical significance. There was comparatively smaller amount of the 20–25-mm branches of C. korshinskii. Applying the categories interval of 15–18 and 18–25 was aimed to make sure the minimum branches amount between these two neighboring categories for meeting the statistical significance. Please see Point (4) at Reply to R2C2 and Point (3) at Reply to R2C3 for explaining the representativeness of selected 7 branches and 4 shrubs for stemflow recording, respectively.

R3C5: L662. Table 2: Do the rainfall indicators including RA, RD, RI, I, I10, Ib10 etc differ statically significantly among Event A, Event B, Event C and Others? Please
provide the ANVOA results here. L670. Table 3: The comment is the same with the last one. Please provide the statistical results to depict the difference in the stemflow variables among Event A, Event B, Event C and Others.

Reply: Thank you for this comment. The One-way analysis of variance (ANOVA) with LSD post hoc test had been performed to determine whether rainfall characteristics and stemflow variables differed significantly among event categories, and whether funnelling ratio and stemflow intensities differed significantly among BD categories for C. korshinskii and S. psammophila. The level of significance was set at 95% confidence interval (p=0.05) (Lines 284–289, Pages 13–14). The ANOVA results had been stated in the section 3.1 Rainfall characteristics at Lines 307–312, Page 14–15, Section 3.2 Stemflow volume, intensity, funnelling ratio and temporal dynamics at Lines 337–342, Page 16, and Table 2–4 (Lines 808–829, Pages 40–42).

Reference: Herwitz, S.R.: Infiltration-excess caused by Stemflow in a cyclone-prone tropical rainforest, Earth Surf. Proc. Land, 11, 401–412, https://doi.org/10.1002/esp.3290110406, 1986. Spencer, S. A. and van Meerveld, H. J.: Double funnelling in a mature coastal British Columbia forest: spatial patterns of stemflow after infiltration, Hydrol. Process., 30, 4185–4201, https://doi.org/10.1002/hyp.10936, 2016.

Please also note the supplement to this comment:
https://www.hydrol-earth-syst-sci-discuss.net/hess-2019-254/hess-2019-254-AC2-supplement.pdf

**Supplement:**

[Figure]

**中国科学院生态环境研究中心**
**Research Center for Eco-Environmental Science**
**Chinese Academy of Sciences (CAS)**

北京市海淀区双清路 18 号 邮编：100085  Website: www.rcees.ac.cn
**18 Shuangqing Road, Haidian District, Beijing 100085, P. R. China**
Tel: 0086-10-62911239   Email: gygao@rcees.ac.cn

August 19, 2019

Memorandum

To:   Prof. Lixin Wang, Editor of *Hydrology and Earth System Science*

Subject: **Revised manuscript of hess-2019-254**

Dear Prof. Wang,

  We have substantially revised our manuscript entitled as "*Temporal-dependent effects of rainfall characteristics on inter-/intra-event branch-scaled stemflow variability in two xerophytic shrubs*" after considering all the comments of Prof. David Dunkerley and another two anonymous reviewers, which are of great help to improve this manuscript.

  The following are the point-to-point response to all these comments, including (1) Response to the anonymous Reviewer #1, (2) Response to Reviewer #2 (Prof. David Dunkerley), (3) Response to the anonymous Reviewer #3, (4) The revised manuscript, and (5) The revised manuscript with marks in comparation with the previous version, respectively.

[Figure]

**中国科学院生态环境研究中心**
**Research Center for Eco-Environmental Science**
**Chinese Academy of Sciences (CAS)**
北京市海淀区双清路 18 号 邮编：100085  Website: www.rcees.ac.cn
**18 Shuangqing Road, Haidian District, Beijing 100085, P. R. China**
Tel: 0086-10-62911239    Email: gygao@rcees.ac.cn

**Response to Reviewer #1**

**General Comments:** The paper by Yuan et al mainly aimed to characterize the inter-/intra-event stemflow dynamics of two xerophytic shrubs and to quantify their relationships with the corresponding inter-/intra-event rainfall characteristics. They concluded that rainfall characteristics had temporal-dependent influences on corresponding stemflow variables.

From my point of view, the study has potential to make a contribution to a better understanding of, in particular, the intra-storm stemflow processes and the underlying mechanisms governing its dynamics. The experimental design and data analysis are generally acceptable, while clarity is needed in presenting the design. The figures adequately summarize the results. I recommend this paper for publication in HESS after some moderate revisions had been addressed by the authors.

**Reply:**

We appreciated the anonymous reviewer for the comments and suggestions, which were of great help to improve the overall quality of this manuscript. The manuscript had been carefully revised, and we tried best to submit a qualified manuscript as required.

**R1C2:** L 69: Change "initialed" to "initiated".
**Reply:** Done (Line 73, Page 4).

**R1C3:** L 72: I would use "leafed period" instead of "leaf period".
**Reply:** Done (Line 77, Page 4).

**R1C4:** Section 2.2: What is the time interval for recording rainfall and the stemflow in subsequent section? This needs to be clearly stated.
**Reply:**

Sensors were installed at the meteorological station to record wind speed (Model 03002, R. M. Young Company, USA), air temperature and relative humidity (Model HMP 155, Vaisala, Finland). They were logged at 10-min intervals by a datalogger (Model CR1000, Campbell Scientific Inc., USA) (Lines 142–146, Page 7). We recorded stemflow and rainfall via the Onset® (Onset Computer Corp., USA) RG3-M tipping-bucket rain gauges (hereinafter referred to as TBRG). When the bucket (with resolution of 0.2 mm and the equivalent volume of 3.73 mL) was filled and tipped, data of stemflow or rainfall was stored at the dynamic time interval. It depended on rainfall and stemflow intensities. In general, we recorded meteorological features of WS, T and H at 10-min intervals. However, the rainfall and stemflow was recorded at dynamics intervals between neighboring tips with the fixed 0.2-mm resolution (Lines 221–222, Page 10).

**R1C5:** L 184-186: According to Table 1, stemflow data of S. psammophila are not available for branches with a BD of 15-18 mm rather than 18-25 mm. Please verify this.

**Reply:** The typo here of "18-25 mm" had been revised to "15-18 mm" at Line 213, Page 10.

**R1C6:** Section 2.4: I miss the information about how many rain gauges the authors used in recording stemflow. Did each branch connect to a rain gauge? It seems to be the case from my view of Fig. 1, which makes a total of 14 rain gauges. Please explicitly state to avoid guessing.

**Reply:**

TBRGs had been applied in this study to automatically record stemflow volume and timing. Each TBRG connected to one experimental branches of *C. korshinskii* and *S. psammophila*. Seven branches were selected at different BD categories for each species. Therefore, we had installed 14 TBRGs for stemflow measuring in this study. It had been clearly described at the revised manuscript (Lines 220, Page 10).

**R1C7:** L 203: I would change "base area" to "orifice area", which is a more accurate terminology for rain gauge.

**Reply:** Done (Line 234, Page 11).

**R1C8:** L 200-210: As for mL of SFV, it should be calculated as: SFV = [mm (branch stemflow recorded by tipping-bucket rain gauges) / 10] cm2 (orifice area of a rain gauge). I think the authors missed a 10. Therefore, for the calculation of stemflow volume and stemflow intensity, I suggest that authors provide the corresponding mathematical equations; it would be concise and easier for readers to follow.

**Reply:**

Thank you for commenting on the poorly explained data processing at this manuscript. At the previous version of this manuscript, we just gave the factors for calculating stemflow volume (SFV, mL), i.e., stemflow depth recorded by TBRG ($SF_{RG}$, mm) and orifice area (186.3 $cm^2$). The equation for SFV computation had been described at the revised manuscript (Equation 10) (Lines 235, Page 11). Besides, the definitions and calculations of stemflow intensity (Equation 11–13, Lines 246–248, Page 12), time lags to rains (Lines 252–257, Page 12) and other meteorological features (Equation 1–9, Lines 158–160, Line 164, Lines 184–188, Pages 8–9) had also been clearly described at section *2.2 Meteorological measurements and calculations* and *Section 2.4 Stemflow measurements and calculations*.

**R1C9:** L 211-215: According to the calculation of TLG, TLM, and TLE, these variables can have either negative or positive values. I encourage the authors to clarify here their respective meanings, i.e., what positive values are suggesting and what negative values are suggesting. Again, it would be easier for readers to better understand their following results.

**Reply:**

[Figure]

**中国科学院生态环境研究中心**
**Research Center for Eco-Environmental Science**
**Chinese Academy of Sciences (CAS)**

北京市海淀区双清路 18 号 邮编：100085  Website: www.rcees.ac.cn
**18 Shuangqing Road, Haidian District, Beijing 100085, P. R. China**
Tel: 0086-10-62911239  Email: gygao@rcees.ac.cn

Thank you for this comment. Associated with the results in this study, the meanings of positive and negative values of TLG, TLE and TLM had been described at the *Section 3.2 Stemflow volume, intensity, funnelling ratio and temporal dynamics* at the revised manuscript. During the 54 events, no negative values were observed for TLG and TLM but TLE. It indicated that stemflow generally initiated and maximized after rains started for both species. However, stemflow might be ended before (negative TLE) and after (positive TLE) rains ceased. (Lines 326–329, Page 15).

**R1C10:** L 258-259: It would be more straightforward to add a row in Table 2 showing how many rainfall events occurred for each category (Event A to C, and others).

**Reply:** Done (Line 808, Page 40).

**R1C11:** L 291-298: If it is possible, I would also expect to see some results about the differences of stemflow variables varied among BD categories.

**Reply:**

Thank you for this comment. As suggested, we compared SFI and FR at different BD categories of *C. korshinskii* and *S. psammophila*. Shown at Table 4, FR of *C. korshinskii* decreased from 163.7 at the 5–10-mm branches to 97.7 at the 18–25-mm branches. The decreasing trend of FR were also noted for *S. psammophila* in the range of 44.2–212.0, as branch size increased. The results were in consistence with the findings for trees and babassu palms in an open tropical rainforest in Brazil (Germer et al., 2010), in the coastal British Columbia forest with mixed species (Spencer and Meerveld, 2016), for trees (*Pinus tabuliformis* and *Armeniaca vulgaris*) and shrubs (*C. korshinskii* and *S. psammophila*) at Loess Plateau of China (Yang et al., 2019). Because funnelling ratio was calculated as the ratio between stemflow and rainfall intensities, SFI was also compared at different BD categories. It was negatively related with branch size for both species. As indicated at Equation 14–15 (Lines 264–265, Page 12), the decreasing stemflow intensity with branch size might partly explained the negative relations between funnelling ratio and BD.

However, we did not compare all the stemflow variables at different BD categories. Because of the high expense of TBRGs (Turner et al., 2019), no more than two branches were selected for stemflow recording at each BD category. The results were much more convincing to analyze the average stemflow variables among BD categories, and compared them at different rainfall amount categories with enough events for meeting the statistical significance.

**R1C12:** Section 4.1: I would like to discuss with the authors about the use and importance of stemflow intensity and RSFI. I admit that stemflow intensity would be a good variable to show the dynamics of intra-event stemflow, while I am not convinced by authors about the importance of comparing the absolute values of stemflow intensity versus rainfall intensity (also demonstrated in L26-30 of Abstract). Their study is based on monitoring branch stemflow, and branch stemflow intensity was a bit higher than rainfall intensity in their study. However, in terms of stemflow's ecological and hydrological importance such as in providing additional soil water and sustaining vegetation growth, we pay more attention to the whole tree/shrub (rather than a single branch). From my understanding this variable is highly dependent on the size of a shrub/tree, because a lager shrub/tree (normally has larger basal diameter or canopy area) would generate substantially higher volume of stemflow, therefore stemflow intensity calculated based on collecting from individual trees/shrubs would be far greater than rainfall intensity, as examples please see Fig. 3 in cayuela et al. (2018, Journal of Hydrology) or Fig. 7 in Germer et al. (2010, Journal of Hydrology). Stemflow and rainfall differs in their paths entering into rain gauges; the orifice area makes sense for rainfall because this area is precisely where rainfall falling into and rainfall depth is then normalized, while stemflow is part of intercepted rainfall by the canopy and then comes down stems, which indicates that infiltrating soil area of stemflow is quite different than that of a rain gauge (i.e., orifice area). Therefore this variable may be prone to underestimate stemflow's eco-hydrological role for small shrubs, as such, in terms of ecological importance this variable seems to be less appropriate to be used for inter-specific comparison or even intra-specific comparison of varying sizes. Moreover, the authors were also recommending a future combination use of funnelling ratio and RSFI in stemflow studies. While I agree with the authors that RSFI is helpful in better understanding of the intra-event convergence effects, funnelling ratio assumes trunk/stem basal area is the true area that stemflow is delivered to the soil, whereas RSFI here is based on stemflow intensity which I have discussed above. RSFI may also be prone to underestimate stemflow's eco-hydrological role for small trees/shrubs while overestimate that of big trees/shrubs. I encourage authors to discuss both the advantages and limitations of stemflow intensity and

RSFI as well as their application.

**Reply:**

Thank you for commenting on the calculation and importance of stemflow intensity and RSFI at this manuscript. It indeed underestimated the eco-hydrological significance of stemflow by ignoring its receiving area of branch base as suggested. Therefore, we had revised the calculation of stemflow intensity on basis of basal area, and introduced funnelling ratio to assess the convergence effect of stemflow at the revised manuscript.

Please see the detailed explanations as below.

(1) Stemflow intensity had been re-computed on basis of branch basal area, and quantitatively connected to funneling ratio.

The RG3-M TBRGs had been applied to record stemflow in this study. Stemflow depth ($SF_{RG}$, mm) could be directly computed with tip amounts and tip resolution of 0.2 mm. Similar with the interpretation for rainfall recording, the 0.2-mm per tip represented 200 mL water deposing on the 1-$m^2$ ground surface. Based at the same receiving areas, we calculated stemflow intensity as the ratio between $SF_{RG}$ and rainfall duration at the previous manuscript. However, it underestimated the eco-hydrological significance of stemflow by ignoring the limited area of trunk/branch base, over which stemflow was received. As suggested at this comment, stemflow intensity should associate with the area over which the equivalent stemflow depth is evaluated. Therefore, we re-calculated stemflow intensity and followed the definition of stemflow volume per basal area per unit time (Herwitz, 1986; Spencer and Meerveld, 2016). In this study, we calculated stemflow intensity at different time intervals, including the event base (SFI), the 10-min ($SFI_{10}$) and the dynamic intervals between neighboring tips of TBRG ($SFI_i$) (Equation 11–13) (Line 246–248, Page 12). Furthermore, we established the quantitative connections of stemflow intensity with funnelling ratio for the first time as indicated at Equation 14–15 (Lines 264–265, Page 12). RSFI had been deleted at the revised manuscript. By replacing the event-based volume of rainfall and stemflow with their intensities at the traditional expression (Herwitz, 1986), the new method enabled funnelling ratio to be computed at high temporal resolutions within event.

(2) Stemflow variables and the meteorological influences were analyzed at branch scale.

*C. korshinskii* and *S. psammophila* are modular organisms with multiple branches. Each branch of them lives as independent individual which seeks its own survival goals and compete with each other for light and water (Firn, 2004; Allaby, 2010). They provide ideal experimental objects to measure the branch stemflow volume and production processes. By introducing branch basal diameter (BD, mm) as intermediate variable, stemflow volume, intensity and funnelling ratio could be upscaled from branches to shrubs (Yuan et al., 2016; 2017). Therefore, the study on branch stemflow variables was conducive to explain the meteorological influences on stemflow at shrub scale particularly for the modular organisms. To guarantee the representativeness of experimental shrubs and branches, the thorough plot investigation had been carried out. Please see Point (3) at Reply to R2C3 for describing the determination of standard shrubs at the plots of *C. korshinskii* and *S. psammophila*, and see Point (4) at Reply to R2C2 for explaining the determination of standard branches of the two shrubs. To address the branch scaled measurements of stemflow, the title had been revised as "Temporal-dependent effects of rainfall characteristics on inter-/intra-event branch-scaled stemflow variability in two xerophytic shrubs" as suggested by Reviewers 2 and 3.

**R1C13:** L 433-437: These sentences are somewhat redundant (have been mentioned in above sections) and can be simplified or simply deleted.
**Reply:** Done.

**R1C14:** Figure 3: Data points are average values for 7 branches for each event? Since the authors selected 7 branches of varying BD for each species to measure stemflow, a relative larger difference in stemflow would be expected among branches. It would be an option to adding error bars if they won't make the figure blurring too much.
**Reply:**

Stemflow variables were averaged at seven branches of *C. korshinskii* and *S. psammophila*, respectively. Inter-event variations of the average stemflow variables during the experimental period had been shown at Figure 3. The relatively high expense of TBRGs limited the number of experimental branches that could be measured (Turner et al., 2019). However, each experimental branch was carefully selected following the strict criteria. Please see Point (4) at Reply to R2C2 for explaining the representativeness of the selected seven branches. A total of seven branches were selected for automatic recording via TBRGs at different BD categories of each species. That was the comprehensive results by balancing the statistical significance and TBRG expenses.

To better meeting the statistical significance, we took the average value of stemflow variables at the seven branches at each species, and focused on the comparison of them among different rainfall amount categories. We just discussed the influence of rainfall characteristics in this study, and no analysises were performed to explore the influence of branch traits affecting stemflow volume and process. The variation of stemflow variables had been described as the average±standard error (Iida et al., 2017) at Table 3 ==(Lines 817–824, Page 41)==. However, since eight stemflow variables with 54 recording points each were shown at the same figure, the error bars were not drawn at Fig.3 just to keep the intra-event variation of stemflow variables clean and tidy ==(Lines 835–837, Page 45)==.

**R1C15:** Figure 4:The unit of rainfall stemflow intensity should be mm h$^{-1}$ rather than m h$^{-1}$. Also changes should be made in the legend, since both lines and points are included in this figure, it would be misleading by labelling "Lines in blue" or "Lines in red" without mentioning points. Moreover, since 7 branches for each species were selected for monitoring stemflow intra-event dynamics, I am wondering which branches for two species were demonstrated in this figure.

**Reply:**

Done. The typo unit (m h$^{-1}$) had been corrected to mm h$^{-1}$, and the misleading legends had been revised, and the branch size of *C. korshinskii* and *S. psammophila* had been added at Fig.4 ==(Line 837–840, Page 46)==.

**Response to Reviewer #2: Prof. Dunkerley**

**General Comments:** The authors report on a detailed study of stemflow in two dryland shrub species, and its relationship with rainfall properties. The data come from field observations of selected branches that were equipped with stemflow collecting collars, and exposed to a number of natural rainfall events. Seven branches were instrumented for each of the two shrub species. The stemflow was recorded by directing the flow into tipping-bucket rain gauges having a 0.2 mm sensitivity.

Although the work appears to be generally thorough, there are some significant issues with it that I consider require clarification before the work could be accepted for publication.

**Reply:**

  We would like to extend our sincere gratitude to Prof. Dunkerley for these constructive comments and suggestions. They were of great help to improve this manuscript. We have carefully revised this manuscript as required.

**R2C1:** The authors are concerned with the relative timing of rainfall and of the resulting stemflow. The difficulty here is that the relative timing is affected by the size of the collecting areas that contribute either rainfall or stemflow to the measuring gauges. The canopy of S. psammophila for instance is reported as 21.4 m2 (line 170), whilst the collecting area of the pluviography TBRG in the open is just 0.018 m2. Thus the canopy area of the shrub is more than 1,000 times larger. Therefore, the tiny tipping bucket (capacity about 3.65 mL, by my estimation) can potentially be filled more rapidly by stemflow than by rainfall in the open. In this way, the time until first tip (regarded by the authors as the onset of stemflow) probably occurs closer to the onset of rainfall as a function of canopy area and its effect in reducing the bucket filling time.

Therefore, among the seven instrumented branches, the timing of stemflow initiation should vary, and it might be possible to relate this to the plant morphology. However, the authors do not report the canopy collecting area for the 7 branches that they monitored for each of the two shrub species. Therefore, calculations of the kind just sketched cannot be made nor the results evaluated properly. This imposes uncertainty in the interpretation of the stemflow timing data. The ideal, of course, would be for the collecting area of foliage and branch to be as close as possible to the collecting area of the open-field rain gauge.

Indeed, the manuscript lacks any detail of the foliar area on the branches that were monitored for stemflow. For instance, leaf area and leaf wettability are not mentioned or reported. Likewise, there are no data on the shrub canopies as a whole, such as leaf area index (LAI) or canopy gap fraction. The lack of such information again makes the results somewhat difficult to interpret or to compare with results from other taxa and environments.

[Figure]

**Reply:**

Thank you for this comment. As suggested by Prof. Dunkerley, the initiation of rainfall and stemflow, and the time intervals between them were indeed strongly affected by the corresponding areas to collect them. Therefore, we had carefully discussed the influence of interception area affecting stemflow volume, depth, fraction and funnelling ratio at 53 branches of *C. korshinskii* and 98 branches of *S. psammophila* at Yuan et al. (2016; 2017), including the leaf area of individual branches, branch size, the specific surface area of canopy representing by leaves and stems at both the leafed and leafless states, respectively. By installing TBRGs at 7 branches of each species, this study mainly concentrated the branch-scaled inter-/intra-event stemflow variabilities and the influence of rainfall characteristics affecting them. The influence of leaf area index (LAI) and crown area were not discussed at the shrub scale.

The reasons were detailedly explained as below.

(1) Stemflow variables and meteorological influences were analyzed at branch scale.

*C. korshinskii* and *S. psammophila* are modular organisms with multiple branches. Each branch of them lives as independent individual which seeks its own survival goals and compete with each other for light and water (Firn, 2004; Allaby, 2010). They provide ideal experimental objects to measure the branch stemflow volume and production processes, which could be upscaled to stemflow variables of individual shrubs (Yuan et al., 2016; 2017). The branch-scaled study of stemflow process was conducive to better understand stemflow production at shrub scale particularly for the modular organisms. Therefore, this study focused on the branch-scaled stemflow volume, intensity, temporal dynamics and funnelling ratio of the two species, and analyzed the influences of rainfall characteristics affecting them.

(2) Stemflow variables were averaged at seven different-sized branches of each species.

Seven branches were selected to automatically record stemflow via TBRGs at different BD categories of *C. korshinskii* and *S. psammophila*, respectively. The relatively high expense of TBRGs limited the number of experimental branches that could be measured (Turner et al., 2019). However, each experimental branch was carefully selected following the strict criteria as stated at Point (3) of Reply to R2C3 and Point (4) of Reply to R2C2. Thus, we tried best to guarantee the selected experimental branches to represent the experimental shrubs, and the selected shrubs to represent the *C. korshinskii* and *S. psammophila* plots in this study. That was the comprehensive results by balancing the statistical significance and TBRG expenses.

Average stemflow variables were took at these seven branches to present the branch stemflow variables of the representative shrubs at *C. korshinskii* and *S. psammophila* plots. We mainly compared them at different rainfall amount (RA) categories, and discussed the influence of rainfall characteristics affecting them. Therefore, the variances of branch morphologies within species were not relevant to the average branch-scaled stemflow variables. However, they had been described as important background information at Table 1. The canopy traits were also stated at *Section 2.3* (Lines 197–199, Page 9).

(3) Recording stemflow process with the tipping bucket rain gauges had been justified.

Tipping bucket rain gauges (TBRGs) provided the intra-event monitoring of stemflow and had been widely applied (Iida et al., 2012), although they underestimated the inflow water with systematic mechanical errors (Turner et al., 2019). The bigger bucket volume might bring the larger underestimation (Iida et al., 2012). Therefore, RG3-M rain gauges were used in this study with the relatively smaller bucket volume of 0.2 mm (the equivalent volume of 3.73 mL, email-confirmed by the Onset company). Besides, we corrected the TBRG recording via the regressions with manual measurements as per Equation 4 to further mitigate its underestimation (Line 164, Page 8).

TBRGs offered the ability to collect the volume and timing of inflow water throughout an event (Turner et al., 2019). When the bucket was filled by rains and tipped, it was recorded as the beginning of incident rains. Comparatively, stemflow started in a much more complicated manner. Because it could not be initiated until the canopy was saturated. The larger branch leaf area could help to initiate stemflow earlier for trapping more rains, but might also result in a later generation by consuming more rains to wet canopy. Furthermore, stemflow generation also affected by the traveling time from canopy down to branch base, which was strongly affected by the bark roughness. Therefore, compared with the simply positive relation between TBRG orifice area and rains initiation in the clearings, the larger leaf area to intercept rains could not guarantee a quick start of stemflow. Our results indicated *C. korshinskii* and *S. psammophila* averagely initiated stemflow 66.2 and 54.8 min later than rains began during the 2014–2015 rainy seasons. Time lags of stemflow generation to rains was also supported by Germer (2010) and Cayuela et al. (2018). In general, TBRG was not perfect to precisely record stemflow timing, but might be the plausible devices to record stemflow process by far.

**R2C2:** Data processing is poorly explained. Stemflow intensity, given in mm h-1, requires that the volume of water delivered to the TBRG used to record stemflow (recorded in mL per bucket tip) must be associated with the area over which the equivalent stemflow depth is evaluated. I could not see this explained anywhere in the manuscript, and it needs to be made clear. If it was the cross-sectional area of the branch being monitored (typically about 3 cm2 by my rough estimation) then this needs to be set out in the manuscript. If the authors did use basal branch cross-sectional area, then of course the stemflow intensity can easily exceed the rainfall intensity, as a function of the very small area over which the stemflow is recorded as arriving - far smaller than the collecting area of the rainfall pluviograph. If this area were to be doubled, then the stemflow intensity would be halved (and so on). Therefore, the area used by the authors in their calculation needs to be stated (and justified by some relationship to plant water availability).

Data processing is also poorly explained in terms of the data on stemflow volume presented by the authors (e.g. in Table 3). Are the stemflow volumes reported there, and discussed at many places in the paper, the sum of the stemflow on the 7 monitored branches, or the arithmetic mean of the stemflow from the 7 branches, or are the figures scaled-up to estimate the stemflow delivered by the entire test shrub? (The test shrubs had a total of 180 and 261 branches (line 173) only 7 of which were monitored for each shrub species (amounting to a sample of 4% and

2.6% of the branches, the adequacy of which is not discussed by the authors). Whatever the authors did, it is not made clear and this needs to be corrected. Especially in relation to stemflow, all relevant parameters used in data processing must be set out clearly and systematically. Without knowing the details of the calculation procedure, the relative intensity of the stemflow and the open-field rainfalls are difficult to interpret. No formulae are presented by the authors that would allow this to be checked. My own feeling is that the stemflow flux would be a more useful figure - that is, the flow rate delivered to the base of the branch, expressed for instance in mL/minute or L/hour. If this is accompanied by a clearly-stated area over which the flow is tallied, then a stemflow intensity can be calculated.

**Reply:**

Thank you for this comment. The poorly-explained data processing has been carefully revised. We have detailedly described the definitions and calculations of stemflow volume, intensity, time lag to rains and other meteorological features at the revised manuscript. The representativeness of the selected was stated as below.

(1) Stemflow intensity has been computed following the definition as the stemflow volume per basal area per unit of time.

The RG3-M TBRGs had been applied to record stemflow in this study. Stemflow depth ($SF_{RG}$, mm) was computed with tip amounts within event by multiplying tip resolution of 0.2 mm. Similar with the interpretation for rainfall recording, the 0.2-mm per tip represented 200 mL water deposing on the 1-$m^2$ ground surface. Based at the same receiving areas, we calculated stemflow intensity as the ratio between $SF_{RG}$ and rainfall duration at the previous manuscript. However, it underestimated the eco-hydrological significance of stemflow by ignoring the limited area of trunk/branch base, over which stemflow was truly received. Therefore, following the definition of stemflow volume per basal area per unit time (Herwitz, 1986; Spencer and Meerveld, 2016), we re-computed stemflow intensity with the branch base area at different temporal scales, including the event (SFI), the 10-min ($SFI_{10}$) and the intervals between neighboring tips of TBRG ($SFI_i$) (Equation 11–13 at Lines 246–248, Page 12). Furthermore, we established the quantitative connections of stemflow intensity with funnelling ratio for the first time (Equation 14 at Line 264, Page 12). By replacing the event-based volume of rainfall and stemflow with their intensities at the traditional expression, this new method enabled to calculate funnelling ratio at both inter-/intra-event scales (Lines 554–555, Page26).

(2) The detailed definition and calculation had been described for stemflow variables and rainfall characteristics.

The definitions and calculations had been described for stemflow volume (SFV, mL) (Equation 10 at Lines 235, Page 11), stemflow duration (SFD, h), time lags stemflow generation (TLG, min), maximization (TLM, min) and ending (TLE, min) at Lines 249–257, Page 12, the regression for rectifying the TBRG recordings with manual measurements (Equation 4) at Lines 164, Page 8, evaporation coefficient (E, unitless) (Equation 1–3) at Lines 158–160, Page 8, the allometric equations for estimating leaf area of branches at *C. korshinskii* and *S. psammophila* at Lines 215–218, Page 10.

(3) Stemflow variables had been averaged at different BD categories to analyze the most influential rainfall characteristics affecting them.

Stemflow variables were averaged at different-sized branches to present the branch-scaled stemflow variables of the representative shrubs at *C. korshinskii* and *S. psammophila* plots. We carefully checked the results of stemflow variables, and listed the average values of seven branches during rainfall events with different intensity peak amounts at Table 3 (Lines 817–824, Page 41). Please see the detailed description at Point (2) of Reply to R2C1.

(4) Seven representative branches were selected for stemflow recording at each species.

This study selected 4 shrubs for measuring stemflow and 1 shrub for establishing allometric equations of biomass and leaf areas at each species (Yuan et al., 2016; 2017). Please see Point (3) at Reply to R2C3 for a detailed description of the representativeness of selected experimental shrubs. The morphological features had been measured for all the 180 and 261 branches at these 5 shrubs of *C. korshinskii* and *S. psammophila*, respectively, thus to determining the standard branches for stemflow recording in this study. BD categories were grouped to guarantee the minimum branch amount at each category for meeting the statistical significance. The ≤5-mm branches were not included in stemflow measurements, because they were too weak to bear the fossil collars for trapping stemflow. Considering the high meteorological sensitivity of stemflow temporal dynamics, we tried best to select the experimental branches at the same shrub, which were most likely exposed to the similar rainfall characteristics. Moreover, the qualified branches should have the outlayer-of-canopy positions, no intercrossing with neighboring ones and no turning point in height from branch tip to base (Lines 209–210, Page 10). Therefore, apart from the ≤5-mm branches at both species, the >25-mm branches at *C. korshinskii* for not enough qualified individuals, and 15–18-mm branches at *S. psammophila* for TBRG malfunctioning, there are averagely 28 and 41 branches available for stemflow recording per shrub of *C. korshinskii* and *S. psammophila*, respectively (Table R2-1 as below). Finally, 7 branches were selected at each species, which took 25.0% and 17.1% of the available ones per shrub at *C. korshinskii* and *S. psammophila*, respectively. Additionally, the high expense of TBRG was an important reason to limit the amount of experimental shrub and branch for automatic recording of stemflow (Turner et al., 2019).

**Table R2-1.** Branch morphological features of the experimental shrubs of *C. korshinskii* and *S. psammophila*.

| BD categories | *C. korshinskii* | | | | *S. psammophila* | | | |
|---|---|---|---|---|---|---|---|---|
| | BD (mm) | BL (cm) | BA (°) | BN | BD (mm) | BL (cm) | BA (°) | BN |
| ≤5 | 4.1 | 90.4 | 64.1 | 40 | 4.8 | 166 | 66 | 2 |
| 5–10 | 7.3 | 124.9 | 61.8 | 82 | 8.0 | 204 | 64 | 53 |
| 10–15 | 12.5 | 161.1 | 51.7 | 36 | 12.9 | 253 | 58 | 82 |
| 15–18 | 16.3 | 170.6 | 48.7 | 13 | 16.5 | 280 | 52 | 56 |
| 18–25 | 19.3 | 192.3 | 51.3 | 9 | 20.3 | 302 | 50 | 59 |
| >25 | NA | NA | NA | NA | 28.7 | 366 | 50 | 9 |

Note: BD, BL, BA and BN are the basal diameter, length, angle and number of branches.

**R2C3:** In summary, what I find to be missing from the manuscript includes
-some discussion of why 7 stems were studied and whether this is a sufficient sample
-some consideration of the filling time of the buckets in the tipping-bucket gauges used for rainfall and stemflow measurement, and the effect of this on the lag time before the start of stemflow (and the cessation of stemflow after rain ends)
-more detail on the shrubs - including the variability of canopy size etc across the population from which the two sample shrubs were drawn, and some information on leaf area and wettability, if available
-a proper accounting of how stemflow flux was calculated and how the area over which the intensity was scaled was selected.

**Reply:**

(1) Please see Point (4) at Reply to R2C2 and Point (3) at Reply to R2C3 for explaining the representativeness of selected 7 branches and 4 shrubs for stemflow recording, respectively.

(2) Although TBRGs offered the ability to collect stemflow production at high temporal resolution and time lags to rain, they suffered from systematic errors owing to the rate of water delivery to tip buckets (Turner et al., 2019). The TBRGs missed the records of inflow water during tipping intervals, and they consumed water to wet buckets at the beginning (Groisman and Legates, 1994). The calibration was needed to rectify the volume recordings via regressions with the manual measurement results. However, it was difficult for rectifying the temporal data currently. Therefore, applying the TBRG with relative high accuracy was necessary. Iida et al. (2012) reported that the tipping time increased with the bucket volume by comparing different models of TBRG, including the RG3-M (3.73±0.01 mL), OW-34 (15.7±0.3 mL), UIZ-TB20 (198.3±3.3 mL), TXQ-200 (188.7±10.3 mL) and TXQ-400 (403.9±6.9 mL). We chose RG3-M with the small bucket volume of 3.73 mL to mitigate the underestimation in this study. Please see Point (3) at Reply to R2C1 to justify the feasibility of applying TBRGs.

(3) The plot investigations had been carried out at April of 2014 for the 20-year-old *C. korshinskii* and *S. psammophila*. For *C. korshinskii*, three subplots with the size of 5 m×5 m had been selected along the plot diagonal, including subplot A (5 shrubs) and C (6 shrubs) at the ends and subplot B (6 shrubs) at the middle. As indicated at Table R2-2 as below, the average canopy height and area were 1.9±0.1 m and 4.8±0.6 m$^2$, respectively. Because the runoff and sediment plots had already been constructed at the center of *S. psammophila* plot (Fig. R2-1 as below), we selected the subplot (13 shrubs) at northeastern part with the size of 20 m×20 m. The average canopy height and area were 3.5±0.2 m and 19.1±2.2 m$^2$, respectively (Table R2-3 as below). Thus, standard shrub could be determined to represent the two plots. Finally, five experimental shrubs of each species had been selected for stemflow measurements and allometric equation establishments of *C. korshinskii* (2.1±0.2 m and 5.1±0.3 m$^2$) and *S. psammophila* (3.5±0.2 m and 21.4±5.2 m$^2$), respectively.

[Figure]

**中国科学院生态环境研究中心**
**Research Center for Eco-Environmental Science**
**Chinese Academy of Sciences (CAS)**
北京市海淀区双清路 18 号 邮编：100085  Website: www.rcees.ac.cn
**18 Shuangqing Road, Haidian District, Beijing 100085, P. R. China**
Tel: 0086-10-62911239    Email: gygao@rcees.ac.cn

As stated at Point (4) of Reply to R2C2, the standard branches could be determined and seven branches were finally selected for stemflow recording. According to the allometric equations established for estimating leaf area of individual branches (LA, cm$^2$) (Yuan et al., 2016; 2017), LA of experimental shrubs were estimated in the range of 837.7–6394.7 cm$^2$ and 626.3–7513.7 cm$^2$ at different BD categories for *C. korshinskii* and *S. psammophila*, respectively (Table 1 at Lines 805–807, Page 39). Rainfall intervals, the time intervals between neighboring rains (RI, h), was applied to indirectly represent the branch wettability. The drier barks could be estimated when RI was larger. The results of MCA and stepwise regression indicated that RI tightly corresponded to time lags of stemflow ending, but there was no significant quantitative relationship between them for for *C. korshinskii* ($R^2$=0.005, *p*=0.28) or *S. psammophila* ($R^2$=0.002, *p*=0.78) (Fig.7) (Lines 846–847, Page 49).

**Table R2-2.** Investigation of canopy morphology at *C. korshinskii* plot.

| Plots | Shrubs | Canopy heights (m) | Canopy area (m$^2$) |
|---|---|---|---|
| A | 1 | 1.7 | 4.6 |
| | 2 | 1.2 | 2.1 |
| | 3 | 1.9 | 3.7 |
| | 4 | 1.4 | 2.5 |
| | 5 | 2.0 | 5.7 |
| B | 6 | 1.7 | 5.5 |
| | 7 | 1.8 | 4.3 |
| | 8 | 1.8 | 3.8 |
| | 9 | 2.1 | 6.8 |
| | 10 | 2.5 | 11.6 |
| | 11 | 2.3 | 6.7 |
| C | 12 | 1.3 | 3.4 |
| | 13 | 1.9 | 5.9 |
| | 14 | 1.9 | 2.7 |
| | 15 | 1.8 | 2.8 |
| | 16 | 2.0 | 4.0 |
| | 17 | 2.2 | 5.5 |
| Average | | 1.9±0.1 | 4.8±0.6 |

**Table R2-3.** Investigation of canopy morphology at *S. psammophila* plot.

| Shrubs | Canopy heights (m) | Canopy area (m$^2$) |
|---|---|---|
| 1 | 3.8 | 24.0 |
| 2 | 3.8 | 18.5 |
| 3 | 3.6 | 21.8 |
| 4 | 3.7 | 24.0 |

| | | |
|---|---|---|
| 5 | 3.2 | 20.6 |
| 6 | 2.6 | 13.2 |
| 7 | 2.9 | 5.8 |
| 8 | 3.3 | 25.9 |
| 9 | 3.2 | 8.3 |
| 10 | 4.4 | 22.5 |
| 11 | 4.4 | 29.7 |
| 12 | 2.9 | 7.4 |
| 13 | 3.8 | 25.7 |
| Average | 3.5±0.2 | 19.1±2.2 |

[Figure]

**Fig. R2-1.** The established runoff and sediment plots at the *S. psammophila* plot.

(4) Stemflow intensity had been re-calculated on the basis of branch basal area. Please see the detailed description at Point (1) of Reply to R2C2.

**R2C4:** More detailed comments:

lines 49-50: it is difficult to generalise from these few data to all "water stressed regions" (and need to define what a water-stressed region is)

**Reply:** Done. We have revised the "water-stressed regions" into "dryland ecosystems with annual mean rainfall ranging in 154–900 mm" (Line 53, Page 3), which was cited from the reporting of Magliano et al. (2019).

**R2C5:** line 57: mL/g of what? biomass?
**Reply:** It was the unit of stemflow productivity (Yuan et al., 2016; 2017), which represented the stemflow volume of unit biomass. The description has been added at Line 57, Page 3.

**R2C6:** line 61: a flow in units of mL/min is a flux, not a speed
**Reply:** Done. We change the "speed" into "flux" at Line 61, Page 3.

**R2C7:** line 69: should presumably say 'not until AFTER canopies became saturated'
**Reply:** Done (Line 73, Page 4).

**R2C8:** line 70: need to define RA when this contraction is first used. It is used again in line 138 before being defined.
**Reply:** RA has been firstly used and explained at Line 52, Page 3.

**R2C9:** line 76: missing a space before 0.4
**Reply:** Done.

**R2C10:** lines 77-78: need to include branch surfaces also line 83: need to state which measure is maximized
**Reply:** Done. "branch surfaces" has been included at Line 79, and the "stemflow flux" has been stated at Line 84 of Page 4 at the revised manuscript.

**R2C11:** line 85: explain why time lags are important: presumably the last stemflow would occur as a very small (negligible) flux, so why is the timing of the last stemflow important? More generally, the authors could say something about why the time variation of stemflow during rainfall is important. Do peaks of stemflow flux exceed soil infiltration capacity, perhaps? Otherwise, why is this important?
**Reply:** Thank you for this comment. Stemflow might take a minor part of rainfall amount, but it greatly contributes to the survival of xerophytic plant species (Návar, 2011), the maintenance of patch structures in arid areas (Kéfi et al., 2007), and the normal functioning of rainfed dryland ecosystems (Wang et al., 2011) (Lines 52–57, Page 3). Previous studies failed to depict stemflow processes and quantify their relations with rainfall characteristics within events, particularly for xerophytic shrubs (Lines 20–23, Page 1). Time lags of stemflow generation, maximization and ending to rains depicted dynamic stemflow process, and were conducive to better understand the hydrological process occurred at the interface between the intercepted rains and soil moisture (Sprenger et al., 2019). It was important to discuss the temporal persistence in spatial patterns of soil moisture particularly at the intra-event scale (Gao et al.,

2019) (Lines 86–92, Pages 4–5).

**R2C12:** line 100: no need to repeat the number of rainfall events here, and again in line 222 and again in line 248. Once is sufficient.
**Reply:** Done.

**R2C13:** line 106: please define 'stemflow intensity' and provide a formula somewhere in the paper
**Reply:** Done. The definition and formula had been detailedly described at Lines 236–248, Pages 11–12.

**R2C14:** line 139: please explain what 'analogue' means here
**Reply:** Done. The "analogue period of time to dry canopies from antecedent rains" had been revise to "same period of time to dry canopies from antecedent rains as that reported by Giacomin and Trucchi (1992), Zhang et al. (2015), Zhang et al., (2017) and Yang et al. (2019)" at Lines 168–170, Page 8.

**R2C15:** lines 147-148: all these timing data are a function of the tipping-bucket filling time (see discussion earlier in this report). When using a TBRG, it is difficult to tell precisely when rain begins or ends, owing to the time that might be required to fill the first tipping- bucket.
**Reply:** The better understanding of stemflow temporal variables was conducive to address the eco-hydrological importance of stemflow as stated at Reply to R2C11. TBRG was not perfect to precisely record stemflow timing, but might be the plausible devices to record stemflow process by far. Please see Point (3) at Reply to R2C1 for justifying the usage of TBRGs to record stemflow process.

**R2C16:** line 153: how is raindrop morphology reflected in this? please explain
**Reply:** The raindrop momentum was calculated with raindrop size and velocity as indicated at Equation 5–9 (Line 184–188, Page 9), which represent the comprehensive effects of raindrop morphology (size) and kinetic energy (velocity).

**R2C17:** line 160: why is mean intensity used here?
**Reply:** The average rainfall intensity was used here to compute the average raindrop diameter and finally raindrop momentum on event base. The 10-min maximum raindrop momentum ($F_{10}$, mg·m·s$^{-1}$) and the average raindrop momentum at the first and last 10 min ($F_{b10}$ and $F_{e10}$, respectively, mg·m·s$^{-1}$) could be calculated with $I_{10}$, $I_{b10}$ and $I_{e10}$ as indicated at Equation 5–9 (Line 184–188, Page 9), respectively.

**R2C18:** line 168: since this paper reports a study of branch stemflow only, the title of the paper should be amended to indicate this clearly (i.e., not a study of stemflow on an entire plant)

**Reply:** Done. We have revised the title to "Temporal-dependent effects of rainfall characteristics on inter-/intra-event branch-scaled stemflow variability in two xerophytic shrubs" as suggested as Reviewer 3.

**R2C19:** line 171: to what extent were the studied shrubs representative of the wider population? please present some data.

**Reply:** *C. korshinskii* and *S. psammophila* were the dominant shrub species at the arid and semi-arid regions of northwestern China, including Inner Mongolia Autonomous Region, Ningxia Hui Autonomous Region, Xinjiang Uygur Autonomous Region, Qinghai province, Gansu province, Shaanxi province, Shanxi province (Chao and Gong, 1999). Since both species had good drought tolerance, they were commonly planted for soil and water conservation, sand fixation and wind barrier (Li, 2012; Hu et al., 2016; Liu et al., 2016; Zhang et al., 2018). As the typical xerophytic shrub species at this region, they had extensive distributions particularly in arid and desert steppes (Li et al., 2016) at Lines 129–132, Page 6. Besides, please see Point (3) at Reply to R2C3 for explaining the representativeness of the selected 4 experimental shrubs for the *C. korshinskii* and *S. psammophila* plots.

**R2C20:** lie 181: please explain what is meant by 'canopy skirt locations'. The photos suggest that there were many overhanging leaves and branches. Some of the stemflow collars were placed quite high off the ground (as far as can be judged from the photos, as no quantitative information on this is included in the paper). How do the authors know that the stemflow at these heights would actually reach the ground, and not drip off the branches?

**Reply:** The "canopy-skirt locations" has been revised to "the outlayer-of-canopy" at Lines 210,Page 10. The photo shot the lower part of branches to show foil collar and TBRG for stemflow trapping and recording, which might not provide a very clear view of leaves on the upper branches. In contrast to the centered branches, stemflow of branches at the outlayer got less influences from the neighboring ones. We automatically recorded stemflow volume and timing via the RG3-M TBRG with height of 25.7 cm. Therefore, the foil collars were installed at branches nearly 40 cm off the ground (Lines 223–224, Page 11). It might be the minimum height for foil collars so as to keep the hose straight, which channelled stemflow down to TBRGs. The lost by dripping off was believed to be acceptable, compared with the commonly-used method to trap stemflow at breast height (1.2 or 1.3 m off ground) at tress particularly at rainforest, where the stemflow volume was much larger.

**R2C21:** line 189-190: what was the external diameter? this should be included as the dimensions of the stemflow collars are critical - it does not seem sufficient simply to assert that they caught no rainfall or released drips of throughfall from above.

**Reply:** The "external diameter" has been revised to "orifice diameter" at Line 234. The limited orifice diameter of foil collars minimized the accessing of throughfall and rains into them (Yuan et al., 2017) (Lines 225–227, Page 11).

**R2C22:** line 270: how were rainfall intensity peaks identified? What makes one peak an intensity peak?

**Reply:** $SFI_i$, the instantaneous stemflow intensity, was computed in terms of the tip volume (3.73 mL), branch basal area ($mm^2$) and time intervals between neighboring tips recorded by TBRGs as indicated Equation 13 (Line 248, Page 12). The largest $SFI_i$ was defined as the peak intensity at the incident rains.

**R2C23:** line 292: is the reference to the volume from a single branch or the total from the 7 branches?

**Reply:** We focused on the average stemflow variables of 7 experimental branches, and analyzed the most influential rainfall characteristics affecting them. Please see the detailed explanation at Point 2 of Reply to R2C1 and Point 3 of Reply to R2C2.

**R2C24:** lines 300-310: this is difficult to read, owing to the need to recall the meaning of the very many contractions. Some reminders of what these mean would be useful here.

**Reply:** As indicated at the suggestion commenting at Line 70 of R2C5, the contraction was only explained when it was first used. For an easy reading, the list of symbols had been prepared as appendix at the revised manuscript (Lines 592–593, Pages 27–29).

**R2C25:** line 342: a stemflow intensity of 1232 mm h-1 is large. What was the flux? I presume that in the case of the authors own work in the present study, the flux was within the capacity of the tipping-bucket gauges (typically a few hundred mm h-1 at maximum) since the rainfall was not very intense. Some comment on this would be worthwhile.

**Reply:** As indicated at the manual of RG3-M TBRG (https://www.onsetcomp.com/products/data-loggers/rg3-m), data could be automatically recorded at rains with the maximum intensity of 127 mm·h$^{-1}$. The unit depth (mm) of inflow water recorded by TBRG was interpreted to the equivalent 1000 cm$^3$ water on the 1-m$^2$ ground surface. However, stemflow intensity was computed with branch basal areas. It approximately ranged in 34–770 mm$^2$ for *C. korshinskii* and *S. psammophila* in this study, which took less than 0.8‰ of 1 m$^2$. Therefore, it could be estimated that the RG3-M TBRG offers the ability to record stemflow with the maximum intensity greater than 15000 mm·h$^{-1}$.

**R2C26:** lines 383-384: but these fluxes would surely depend on the antecedent leaf and branch wetness, and on meteorological conditions such as wind speed and vapour deficit (the latter is not reported, incidentally).

**Reply:** Thank you for this comment. The evaporation coefficient (E, unitless) had been included at the revised manuscript. E was computed with air temperature, relative humidity and wind speed as indicated at Equation 1–3 (Lines 158–160, Page 8). It represented the comprehensive influences of these meteorological characteristics. By performing the multiple

correspondence analysis (MCA), E and rainfall duration (RD) were tested to closely relate with stemflow duration (Lines 360–362, Page 17). However, the stepwise regression analysis finally confirmed the dominant influence of RD affecting SFD (Lines 381–382, Page 18). Rainfall intervals, the time intervals between neighboring rains (RI, h), was applied to indirectly represent the branch wettability. Please see the detailed description at Point (3) at Reply to R2C3.

**R2C27:** Table 2: why are only 3 rainfall events listed here? More than 40 more are simply lumped under "others" and no details are provided. Why?

**Reply:** Event A, B and C represented three categories of events with the single, double and multiple intensity peak amounts. It had been described at the note of Table 2 (Lines 808–816, Page 40) and *Section 3.1* (Lines 301–303, Pages 14). There were 17, 11 and 15 events at Event A, B and C, respectively. Because the remaining 11 events had the average RA of 0.6 mm, no more than three recordings had been observed within event which was limited by 0.2-mm resolution of TBRGs. Therefore, they could not be categorized and grouped as Event others (Lines 303– 06, Page 14).

**R2C28:** Figure 4 shows units of m/h which I presume should be mm/h
**Reply:** Done.

**Response to Reviewer #3**

**General Comments:** After careful review, I think, in many ways, this is a good manuscript. The work has been well done and the manuscript is well organized. The paper has an appropriate length and the topic is of interest to the general readers of HESS…I recommend this manuscript for publication after a minor revision.

**Reply:**

We appreciated the anonymous reviewer for the comments and suggestions. This manuscript will be carefully revised as suggested prior to being submitted.

**R3C1:** My major concern is the reasonability of the stemflow variables used in this study. For instance, in Line 207, the authors said that the average (SFI) and 10-min maximum (SFI10) stemflow intensities were calculated by the branch stemflow as recorded by the tipping-bucket rain gauges (mm) and rainfall duration (h). In my opinion, stemflow intensities should be defined as the branch stemflow depth (which can be calculated from branch stemflow volume as divided by branch basal area) in a certain time. In the current form, the authors underestimated stemflow intensities. Also, in Line 216, the ratio of the intra-event stemflow intensity (RSFI, unitless) should be calculated basing on the suggested calculation of stemflow intensity.

**Reply:**

Thank you for commenting on the calculation of stemflow variables in this study. As suggested at this comment, it indeed underestimated the eco-hydrological significance of stemflow to compute stemflow intensity by ignoring the limited area of branch base, over which stemflow was received. Therefore, we had re-computed stemflow intensity following the definition as stemflow volume per basal area per unit of time (Herwitz, 1986; Spencer and Meerveld, 2016). It had been calculated at different time intervals, including the event (SFI, mm·h$^{-1}$), 10-min (SFI$_{10}$, mm·h$^{-1}$) and dynamic time interval between neighboring tips (SFI$_i$, mm·h$^{-1}$). Besides, RSFI had been deleted, and funnelling ratio had been introduced to assess the convergence effect of stemflow at the revised manuscript. It had been quantitatively connected with stemflow intensity for the first time as indicated at Equations 14–15 (Lines 264–265, Page 12). Please see the detailed explanation at Point (1) of Reply to R1C12, and Point (1) of Reply to R2C2.

**R3C2:** I also state minor comments as follows. L1: Only seven branches were used to measure stemflow for each shrub species (The studied shrubs had a total of 180 and 261 branches), So the suggested title is: Temporal-dependent effects of rainfall characteristics on inter-/intra-event branch-scale stemflow variability in two xerophytic shrubs.

**Reply:** Done.

**R3C3:** L220-226: It could be better if the authors provide the formula for each stemflow variables.

**Reply:**

Done. The detailed descriptions and calculations of stemflow variables had been stated at the revised manuscript, including stemflow volume (SFV, mL) (Equation 10) at Line 235, Page 11, stemflow duration (SFD, h), time lags stemflow generation (TLG, min), maximization and ending (TLE, min) at Lines 249–257, Page 12, stemflow intensities at the event bases (SFI), the 10-min interval ($SFI_{10}$) and the dynamic intervals between neighboring tips of TBRG ($SFI_i$) (Equation 11–13) at Lines 246–248, Page 12, funnelling ratio at event base (FR) and the 100-s ($FR_{100}$) intervals (Equation 14–15) at Lines 264–265, Page 12.

**R3C4:** L658. Table 1: What is the standard for base diameter (BD) categorization? In the current form, the class interval (5–10, 10–15, 15–18, 18–25, >25 mm) is variable. Why not 5-10, 10-15, 15-20, 20-25, and >25 mm? Please explain it.

**Reply:**

Thanks for this comment. Based on the plot investigation for *C. korshinskii* and *S. psammophila*, standard shrubs canopies could be determined. Four shrubs and 1 shrub had been selected for stemflow measurements and allometric equations establishments. By measuring branch morphologies at all the branches at these five shrubs of each species, BD categories was determined to guarantee the minimum branch amount at each category for meeting the statistical significance. There was comparatively smaller amount of the 20–25-mm branches of *C. korshinskii*. Applying the categories interval of 15–18 and 18–25 was aimed to make sure the minimum branches amount between these two neighboring categories for meeting the statistical significance. Please see Point (4) at Reply to R2C2 and Point (3) at Reply to R2C3 for explaining the representativeness of selected 7 branches and 4 shrubs for stemflow recording, respectively.

**R3C5:** L662. Table 2: Do the rainfall indicators including RA, RD, RI, I, I10, Ib10 etc differ statically significantly among Event A, Event B, Event C and Others? Please provide the ANVOA results here. L670. Table 3: The comment is the same with the last one. Please provide the statistical results to depict the difference in the stemflow variables among Event A, Event B, Event C and Others.

**Reply:**

Thank you for this comment. The One-way analysis of variance (ANOVA) with LSD post hoc test had been performed to determine whether rainfall characteristics and stemflow variables differed significantly among event categories, and whether funnelling ratio and stemflow intensities differed significantly among BD categories for *C. korshinskii* and *S. psammophila*. The level of significance was set at 95% confidence interval ($p$=0.05) (Lines 284–289, Pages 13–14). The ANOVA results had been stated in the section *3.1 Rainfall characteristics* at Lines 307–312, Page 14–15, *Section 3.2 Stemflow volume, intensity, funnelling ratio and temporal dynamics* at Lines 337–342, Page 16, and Table 2–4 (Lines 808–829, Pages 40–42).

[Figure]

**中国科学院生态环境研究中心**
**Research Center for Eco-Environmental Science**
**Chinese Academy of Sciences (CAS)**
北京市海淀区双清路 18 号 邮编：100085  Website: www.rcees.ac.cn
**18 Shuangqing Road, Haidian District, Beijing 100085, P. R. China**
Tel: 0086-10-62911239    Email: gygao@rcees.ac.cn

**Reference:**

[revised manuscript text omitted]

Note: Event A, Event B and Event C are events with the single, double and multiple rainfall intensity peaks, respectively; Others are the events that excluded from the categorization; RA, RD and RI are rainfall amount, duration and interval, respectively; I and $I_{10}$ are the average and 10-min maximum rainfall intensities, respectively; $I_{b10}$ and $I_{e10}$ are the average rainfall intensities in 10 min after rain begins and before rain ends, respectively; F and $F_{10}$ are the average and 10-min maximum raindrop momentums, respectively; $F_{b10}$ and $F_{e10}$ are the average raindrop momentums in 10 min after rain begins and before rain ends, respectively; E is evaporation coefficient; Different letters indicate significant differences of rainfall characteristics between event categories ($p < 0.05$) (rows at the table).

 **Table 3.** Stemflow variables of *C. korshinskii* and *S. psammophila* during rainfall events

 with different intensity peak amounts.

| Species | Stemflow variables | Event A | Event B | Event C | Others | Average |
|---|---|---|---|---|---|---|
| *C. korshinskii* | SFV (mL) | 134.1 a | 203.7 a | 560.8 b | 7.6 c | 226.6 ± 46.4 |
| | SFI (mm·h$^{-1}$) | 672.9 a | 552.4 b | 527.0 b | 317.8 c | 517.5 ± 82.1 |
| | SFI$_{10}$ (mm·h$^{-1}$) | 2849.0 a | 2399.3 a | 1809.1 b | 1173.2 c | 2057.6 ± 399.7 |
| | FR (unitless) | 109.4 a | 146.6 b | 137.9 b | 128.9 ab | 130.7 ± 8.2 |
| | TLG (min) | 67.3 ab | 56.2 a | 67.0 ab | 74.2 b | 66.2 ± 10.6 |
| | TLM (min) | 81.1 a | 75.5 a | 202.1 b | 78.8 a | 109.4 ± 20.5 |
| | TLE (min) | 22.3 a | 18.7 b | 18.5 b | 20.6 a | 20.0 ± 5.3 |
| | SFD (h) | 1.4 a | 3.1 a | 9.1 b | 1.4 a | 3.8 ± 0.8 |
| *S. psammophila* | SFV (mL) | 102.6 a | 145.7 a | 435.2 b | 4.7 c | 172.1 ± 34.5 |
| | SFI (mm·h$^{-1}$) | 648.1 a | 421.5 b | 246.6 c | 153.2 c | 367.3 ± 91.1 |
| | SFI$_{10}$ (mm·h$^{-1}$) | 1672.7 a | 1582.8 a | 888.4 b | 384.7 c | 1132.2 ± 214.3 |
| | FR (unitless) | 77.1 a | 91.4 a | 129.1 b | 101.6 ab | 101.6 ± 10.4 |
| | TLG (min) | 84.9 a | 46.5 b | 56.1 b | 31.5 b | 54.8 ± 11.7 |
| | TLM (min) | 64.3 a | 93.4 a | 235.8 b | 88.4 a | 120.5 ± 22.1 |
| | TLE (min) | 17.1 a | 8.6 b | 20.8 a | 7.3 b | 13.5 ± 17.2 |
| | SFD (h) | 1.2 a | 3.4 a | 8.3 b | 0.7 a | 3.4 ± 0.9 |

Note: Event A, Event B and Event C are events with the single, double and multiple rainfall intensity peaks, respectively; Others are the events that excluded from the categorization; TLG and TLM are time lags of stemflow generating and maximizing after rains begin, respectively; TLE is time lag of stemflow ending after rain ceases; SFD is stemflow duration; SFV is stemflow volume; SFI are the average stemflow intensities at incident rains, respectively; Different letters indicate significant differences of stemflow variables between event categories ($p<0.05$) (rows at the table).

**Table 4.** Comparisons of stemflow intensity and funnelling ratio at different basal diameter categories.

[revised manuscript text omitted]

---

## Author Comment (AC3) · 19 Aug 2019

Please see "Response to Reviewer #2: Prof. Dunkerley" at the attached supplement file for the detailed response by the authors.

General Comments: The authors report on a detailed study of stemflow in two dryland shrub species, and its relationship with rainfall properties. The data come from field observations of selected branches that were equipped with stemflow collecting collars, and exposed to a number of natural rainfall events. Seven branches were instrumented for each of the two shrub species. The stemflow was recorded by directing the flow into tipping-bucket rain gauges having a 0.2 mm sensitivity.

[Figure]

Although the work appears to be generally thorough, there are some significant issues with it that I consider require clarification before the work could be accepted for publication.

Reply:

We would like to extend our sincere gratitude to Prof. Dunkerley for these constructive comments and suggestions. They were of great help to improve this manuscript. We have carefully revised this manuscript as required.

R2C1: The authors are concerned with the relative timing of rainfall and of the resulting stemflow. The difficulty here is that the relative timing is affected by the size of the collecting areas that contribute either rainfall or stemflow to the measuring gauges. The canopy of S. psammophila for instance is reported as 21.4 m2 (line 170), whilst the collecting area of the pluviography TBRG in the open is just 0.018 m2. Thus the canopy area of the shrub is more than 1,000 times larger. Therefore, the tiny tipping bucket (capacity about 3.65 mL, by my estimation) can potentially be filled more rapidly by stemflow than by rainfall in the open. In this way, the time until first tip (regarded by the authors as the onset of stemflow) probably occurs closer to the onset of rainfall as a function of canopy area and its effect in reducing the bucket filling time.

Therefore, among the seven instrumented branches, the timing of stemflow initiation should vary, and it might be possible to relate this to the plant morphology. However, the authors do not report the canopy collecting area for the 7 branches that they monitored for each of the two shrub species. Therefore, calculations of the kind just sketched cannot be made nor the results evaluated properly. This imposes uncertainty in the interpretation of the stemflow timing data. The ideal, of course, would be for the collecting area of foliage and branch to be as close as possible to the collecting area of the open-field rain gauge.

Indeed, the manuscript lacks any detail of the foliar area on the branches that were monitored for stemflow. For instance, leaf area and leaf wettability are not mentioned

or reported. Likewise, there are no data on the shrub canopies as a whole, such as leaf area index (LAI) or canopy gap fraction. The lack of such information again makes the results somewhat difficult to interpret or to compare with results from other taxa and environments.

Reply:

Thank you for this comment. As suggested by Prof. Dunkerley, the initiation of rainfall and stemflow, and the time intervals between them were indeed strongly affected by the corresponding areas to collect them. Therefore, we had carefully discussed the influence of interception area affecting stemflow volume, depth, fraction and funnelling ratio at 53 branches of C. korshinskii and 98 branches of S. psammophila at Yuan et al. (2016; 2017), including the leaf area of individual branches, branch size, the specific surface area of canopy representing by leaves and stems at both the leafed and leaf-less states, respectively. By installing TBRGs at 7 branches of each species, this study mainly concentrated the branch-scaled inter-/intra-event stemflow variabilities and the influence of rainfall characteristics affecting them. The influence of leaf area index (LAI) and crown area were not discussed at the shrub scale. The reasons were detailedly explained as below.

(1) Stemflow variables and meteorological influences were analyzed at branch scale.

C. korshinskii and S. psammophila are modular organisms with multiple branches. Each branch of them lives as independent individual which seeks its own survival goals and compete with each other for light and water (Firn, 2004; Allaby, 2010). They provide ideal experimental objects to measure the branch stemflow volume and production processes, which could be upscaled to stemflow variables of individual shrubs (Yuan et al., 2016; 2017). The branch-scaled study of stemflow process was conducive to better understand stemflow production at shrub scale particularly for the modular organisms. Therefore, this study focused on the branch-scaled stemflow volume, intensity, temporal dynamics and funnelling ratio of the two species, and analyzed the influences of

rainfall characteristics affecting them.

(2) Stemflow variables were averaged at seven different-sized branches of each species.

Seven branches were selected to automatically record stemflow via TBRGs at different BD categories of C. korshinskii and S. psammophila, respectively. The relatively high expense of TBRGs limited the number of experimental branches that could be measured (Turner et al., 2019). However, each experimental branch was carefully selected following the strict criteria as stated at Point (3) of Reply to R2C3 and Point (4) of Reply to R2C2. Thus, we tried best to guarantee the selected experimental branches to represent the experimental shrubs, and the selected shrubs to represent the C. korshinskii and S. psammophila plots in this study. That was the comprehensive results by balancing the statistical significance and TBRG expenses. Average stemflow variables were took at these seven branches to present the branch stemflow variables of the representative shrubs at C. korshinskii and S. psammophila plots. We mainly compared them at different rainfall amount (RA) categories, and discussed the influence of rainfall characteristics affecting them. Therefore, the variances of branch morphologies within species were not relevant to the average branch-scaled stemflow variables. However, they had been described as important background information at Table 1. The canopy traits were also stated at Section 2.3 (Lines 197–199, Page 9).

(3) Recording stemflow process with the tipping bucket rain gauges had been justified.

Tipping bucket rain gauges (TBRGs) provided the intra-event monitoring of stemflow and had been widely applied (Iida et al., 2012), although they underestimated the inflow water with systematic mechanical errors (Turner et al., 2019). The bigger bucket volume might bring the larger underestimation (Iida et al., 2012). Therefore, RG3-M rain gauges were used in this study with the relatively smaller bucket volume of 0.2 mm (the equivalent volume of 3.73 mL, email-confirmed by the Onset company). Besides, we corrected the TBRG recording via the regressions with manual measurements as per

Equation 4 to further mitigate its underestimation (Line 164, Page 8).

TBRGs offered the ability to collect the volume and timing of inflow water throughout an event (Turner et al., 2019). When the bucket was filled by rains and tipped, it was recorded as the beginning of incident rains. Comparatively, stemflow started in a much more complicated manner. Because it could not be initiated until the canopy was saturated. The larger branch leaf area could help to initiate stemflow earlier for trapping more rains, but might also result in a later generation by consuming more rains to wet canopy. Furthermore, stemflow generation also affected by the traveling time from canopy down to branch base, which was strongly affected by the bark roughness. Therefore, compared with the simply positive relation between TBRG orifice area and rains initiation in the clearings, the larger leaf area to intercept rains could not guarantee a quick start of stemflow. Our results indicated C. korshinskii and S. psammophila averagely initiated stemflow 66.2 and 54.8 min later than rains began during the 2014–2015 rainy seasons. Time lags of stemflow generation to rains was also supported by Germer (2010) and Cayuela et al. (2018). In general, TBRG was not perfect to precisely record stemflow timing, but might be the plausible devices to record stemflow process by far.

R2C2: Data processing is poorly explained. Stemflow intensity, given in mm h-1, requires that the volume of water delivered to the TBRG used to record stemflow (recorded in mL per bucket tip) must be associated with the area over which the equivalent stemflow depth is evaluated. I could not see this explained anywhere in the manuscript, and it needs to be made clear. If it was the cross-sectional area of the branch being monitored (typically about 3 cm2 by my rough estimation) then this needs to be set out in the manuscript. If the authors did use basal branch cross-sectional area, then of course the stemflow intensity can easily exceed the rainfall intensity, as a function of the very small area over which the stemflow is recorded as arriving - far smaller than the collecting area of the rainfall pluviograph. If this area were to be doubled, then the stemflow intensity would be halved (and so on). Therefore, the area used by

the authors in their calculation needs to be stated (and justified by some relationship to plant water availability). Data processing is also poorly explained in terms of the data on stemflow volume presented by the authors (e.g. in Table 3). Are the stemflow volumes reported there, and discussed at many places in the paper, the sum of the stemflow on the 7 monitored branches, or the arithmetic mean of the stemflow from the 7 branches, or are the figures scaled-up to estimate the stemflow delivered by the entire test shrub? (The test shrubs had a total of 180 and 261 branches (line 173) only 7 of which were monitored for each shrub species (amounting to a sample of 4% and 2.6% of the branches, the adequacy of which is not discussed by the authors). Whatever the authors did, it is not made clear and this needs to be corrected. Especially in relation to stemflow, all relevant parameters used in data processing must be set out clearly and systematically.

Without knowing the details of the calculation procedure, the relative intensity of the stemflow and the open-field rainfalls are difficult to interpret. No formulae are presented by the authors that would allow this to be checked. My own feeling is that the stemflow flux would be a more useful figure - that is, the flow rate delivered to the base of the branch, expressed for instance in mL/minute or L/hour. If this is accompanied by a clearly-stated area over which the flow is tallied, then a stemflow intensity can be calculated.

Reply:

Thank you for this comment. The poorly-explained data processing has been carefully revised. We have detailedly described the definitions and calculations of stemflow volume, intensity, time lag to rains and other meteorological features at the revised manuscript. The representativeness of the selected was stated as below.

(1) Stemflow intensity has been computed following the definition as the stemflow volume per basal area per unit of time.

The RG3-M TBRGs had been applied to record stemflow in this study. Stemflow depth

(SFRG, mm) was computed with tip amounts within event by multiplying tip resolution of 0.2 mm. Similar with the interpretation for rainfall recording, the 0.2-mm per tip represented 200 mL water deposing on the 1-m2 ground surface. Based at the same receiving areas, we calculated stemflow intensity as the ratio between SFRG and rainfall duration at the previous manuscript. However, it underestimated the eco-hydrological significance of stemflow by ignoring the limited area of trunk/branch base, over which stemflow was truly received. Therefore, following the definition of stemflow volume per basal area per unit time (Herwitz, 1986; Spencer and Meerveld, 2016), we re-computed stemflow intensity with the branch base area at different temporal scales, including the event (SFI), the 10-min (SFI10) and the intervals between neighboring tips of TBRG (SFIi) (Equation 11–13 at Lines 246–248, Page 12). Furthermore, we established the quantitative connections of stemflow intensity with funnelling ratio for the first time (Equation 14 at Line 264, Page 12). By replacing the event-based volume of rainfall and stemflow with their intensities at the traditional expression, this new method enabled to calculate funnelling ratio at both inter-/intra-event scales (Lines 554–555, Page26).

(2) The detailed definition and calculation had been described for stemflow variables and rainfall characteristics.

The definitions and calculations had been described for stemflow volume (SFV, mL) (Equation 10 at Lines 235, Page 11), stemflow duration (SFD, h), time lags stemflow generation (TLG, min), maximization (TLM, min) and ending (TLE, min) at Lines 249–257, Page 12, the regression for rectifying the TBRG recordings with manual measurements (Equation 4) at Lines 164, Page 8, evaporation coefficient (E, unitless) (Equation 1–3) at Lines 158–160, Page 8, the allometric equations for estimating leaf area of branches at C. korshinskii and S. psammophila at Lines 215–218, Page 10.

(3) Stemflow variables had been averaged at different BD categories to analyze the most influential rainfall characteristics affecting them.

Stemflow variables were averaged at different-sized branches to present the branch-scaled stemflow variables of the representative shrubs at C. korshinskii and S. psammophila plots. We carefully checked the results of stemflow variables, and listed the average values of seven branches during rainfall events with different intensity peak amounts at Table 3 (Lines 817–824, Page 41). Please see the detailed description at Point (2) of Reply to R2C1.

(4) Seven representative branches were selected for stemflow recording at each species.

This study selected 4 shrubs for measuring stemflow and 1 shrub for establishing allometric equations of biomass and leaf areas at each species (Yuan et al., 2016; 2017). Please see Point (3) at Reply to R2C3 for a detailed description of the representativeness of selected experimental shrubs. The morphological features had been measured for all the 180 and 261 branches at these 5 shrubs of C. korshinskii and S. psammophila, respectively, thus to determining the standard branches for stemflow recording in this study. BD categories were grouped to guarantee the minimum branch amount at each category for meeting the statistical significance. The ≤5-mm branches were not included in stemflow measurements, because they were too weak to bear the fossil collars for trapping stemflow. Considering the high meteorological sensitivity of stemflow temporal dynamics, we tried best to select the experimental branches at the same shrub, which were most likely exposed to the similar rainfall characteristics. Moreover, the qualified branches should have the outlayer-of-canopy positions, no intercrossing with neighboring ones and no turning point in height from branch tip to base (Lines 209–210, Page 10). Therefore, apart from the ≤5-mm branches at both species, the >25-mm branches at C. korshinskii for not enough qualified individuals, and 15–18-mm branches at S. psammophila for TBRG malfunctioning, there are averagely 28 and 41 branches available for stemflow recording per shrub of C. korshinskii and S. psammophila, respectively (Table R2-1 as below). Finally, 7 branches were selected at each species, which took 25.0% and 17.1% of the available ones per shrub

at C. korshinskii and S. psammophila, respectively. Additionally, the high expense of TBRG was an important reason to limit the amount of experimental shrub and branch for automatic recording of stemflow (Turner et al., 2019).

Table R2-1. Branch morphological features of the experimental shrubs of C. korshinskii and S. psammophila.

R2C3: In summary, what I find to be missing from the manuscript includes

-some discussion of why 7 stems were studied and whether this is a sufficient sample

-some consideration of the filling time of the buckets in the tipping-bucket gauges used for rainfall and stemflow measurement, and the effect of this on the lag time before the start of stemflow (and the cessation of stemflow after rain ends)

-more detail on the shrubs - including the variability of canopy size etc across the population from which the two sample shrubs were drawn, and some information on leaf area and wettability, if available

-a proper accounting of how stemflow flux was calculated and how the area over which the intensity was scaled was selected.

Reply:

(1) Please see Point (4) at Reply to R2C2 and Point (3) at Reply to R2C3 for explaining the representativeness of selected 7 branches and 4 shrubs for stemflow recording, respectively.

(2) Although TBRGs offered the ability to collect stemflow production at high temporal resolution and time lags to rain, they suffered from systematic errors owing to the rate of water delivery to tip buckets (Turner et al., 2019). The TBRGs missed the records of inflow water during tipping intervals, and they consumed water to wet buckets at the beginning (Groisman and Legates, 1994). The calibration was needed to rectify the volume recordings via regressions with the manual measurement results. However, it

was difficult for rectifying the temporal data currently. Therefore, applying the TBRG with relative high accuracy was necessary. Iida et al. (2012) reported that the tipping time increased with the bucket volume by comparing different models of TBRG, including the RG3-M (3.73±0.01 mL), OW-34 (15.7±0.3 mL), UIZ-TB20 (198.3±3.3 mL), TXQ-200 (188.7±10.3 mL) and TXQ-400 (403.9±6.9 mL). We chose RG3-M with the small bucket volume of 3.73 mL to mitigate the underestimation in this study. Please see Point (3) at Reply to R2C1 to justify the feasibility of applying TBRGs.

(3) The plot investigations had been carried out at April of 2014 for the 20-year-old C. korshinskii and S. psammophila. For C. korshinskii, three subplots with the size of 5 m×5 m had been selected along the plot diagonal, including subplot A (5 shrubs) and C (6 shrubs) at the ends and subplot B (6 shrubs) at the middle. As indicated at Table R2-2 as below, the average canopy height and area were 1.9±0.1 m and 4.8±0.6 m2, respectively. Because the runoff and sediment plots had already been constructed at the center of S. psammophila plot (Fig. R2-1 as below), we selected the subplot (13 shrubs) at northeastern part with the size of 20 m×20 m. The average canopy height and area were 3.5±0.2 m and 19.1±2.2 m2, respectively (Table R2-3 as below). Thus, standard shrub could be determined to represent the two plots. Finally, five experimental shrubs of each species had been selected for stemflow measurements and allometric equation establishments of C. korshinskii (2.1±0.2 m and 5.1±0.3 m2) and S. psammophila (3.5±0.2 m and 21.4±5.2 m2), respectively.

As stated at Point (4) of Reply to R2C2, the standard branches could be determined and seven branches were finally selected for stemflow recording. According to the allometric equations established for estimating leaf area of individual branches (LA, cm2) (Yuan et al., 2016; 2017), LA of experimental shrubs were estimated in the range of 837.7–6394.7 cm2 and 626.3–7513.7 cm2 at different BD categories for C. korshinskii and S. psammophila, respectively (Table 1 at Lines 805–807, Page 39). Rainfall intervals, the time intervals between neighboring rains (RI, h), was applied to indirectly represent the branch wettability. The drier barks could be estimated when RI was larger.

The results of MCA and stepwise regression indicated that RI tightly corresponded to time lags of stemflow ending, but there was no significant quantitative relationship between them for for C. korshinskii (R2=0.005, p=0.28) or S. psammophila (R2=0.002, p=0.78) (Fig.7) (Lines 846–847, Page 49).

Table R2-2. Investigation of canopy morphology at C. korshinskii plot.

Table R2-3. Investigation of canopy morphology at S. psammophila plot.

Fig. R2-1. The established runoff and sediment plots at the S. psammophila plot.

(4) Stemflow intensity had been re-calculated on the basis of branch basal area. Please see the detailed description at Point (1) of Reply to R2C2.

R2C4: More detailed comments:

lines 49-50: it is difficult to generalise from these few data to all "water stressed regions" (and need to define what a water-stressed region is)

Reply: Done. We have revised the "water-stressed regions" into "dryland ecosystems with annual mean rainfall ranging in 154–900 mm" (Line 53, Page 3), which was cited from the reporting of Magliano et al. (2019).

R2C5: line 57: mL/g of what? biomass?

Reply: It was the unit of stemflow productivity (Yuan et al., 2016; 2017), which represented the stemflow volume of unit biomass. The description has been added at Line 57, Page 3.

R2C6: line 61: a flow in units of mL/min is a flux, not a speed

Reply: Done. We change the "speed" into "flux" at Line 61, Page 3.

R2C7: line 69: should presumably say 'not until AFTER canopies became saturated'

Reply: Done (Line 73, Page 4).

R2C8: line 70: need to define RA when this contraction is first used. It is used again in line 138 before being defined.

Reply: RA has been firstly used and explained at Line 52, Page 3.

R2C9: line 76: missing a space before 0.4

Reply: Done.

R2C10: lines 77-78: need to include branch surfaces also line 83: need to state which measure is maximized

Reply: Done. "branch surfaces" has been included at Line 79, and the "stemflow flux" has been stated at Line 84 of Page 4 at the revised manuscript.

R2C11: line 85: explain why time lags are important: presumably the last stemflow would occur as a very small (negligible) flux, so why is the timing of the last stemflow important? More generally, the authors could say something about why the time variation of stemflow during rainfall is important. Do peaks of stemflow flux exceed soil infiltration capacity, perhaps? Otherwise, why is this important?

Reply: Thank you for this comment. Stemflow might take a minor part of rainfall amount, but it greatly contributes to the survival of xerophytic plant species (Návar, 2011), the maintenance of patch structures in arid areas (Kéfi et al., 2007), and the normal functioning of rainfed dryland ecosystems (Wang et al., 2011) (Lines 52–57, Page 3). Previous studies failed to depict stemflow processes and quantify their relations with rainfall characteristics within events, particularly for xerophytic shrubs (Lines 20–23, Page 1). Time lags of stemflow generation, maximization and ending to rains depicted dynamic stemflow process, and were conducive to better understand the hydrological process occurred at the interface between the intercepted rains and soil moisture (Sprenger et al., 2019). It was important to discuss the temporal persistence in spatial patterns of soil moisture particularly at the intra-event scale (Gao et al., 2019) (Lines 86–92, Pages 4–5).

R2C12: line 100: no need to repeat the number of rainfall events here, and again in line 222 and again in line 248. Once is sufficient.

Reply: Done.

R2C13: line 106: please define 'stemflow intensity' and provide a formula somewhere in the paper

Reply: Done. The definition and formula had been detailedly described at Lines 236–248, Pages 11–12.

R2C14: line 139: please explain what 'analogue' means here

Reply: Done. The "analogue period of time to dry canopies from antecedent rains" had been revise to "same period of time to dry canopies from antecedent rains as that reported by Giacomin and Trucchi (1992), Zhang et al. (2015), Zhang et al., (2017) and Yang et al. (2019)" at Lines 168–170, Page 8.

R2C15: lines 147-148: all these timing data are a function of the tipping-bucket filling time (see discussion earlier in this report). When using a TBRG, it is difficult to tell precisely when rain begins or ends, owing to the time that might be required to fill the first tipping- bucket.

Reply: The better understanding of stemflow temporal variables was conducive to address the eco-hydrological importance of stemflow as stated at Reply to R2C11. TBRG was not perfect to precisely record stemflow timing, but might be the plausible devices to record stemflow process by far. Please see Point (3) at Reply to R2C1 for justifying the usage of TBRGs to record stemflow process.

R2C16: line 153: how is raindrop morphology reflected in this? please explain

Reply: The raindrop momentum was calculated with raindrop size and velocity as indicated at Equation 5–9 (Line 184–188, Page 9), which represent the comprehensive effects of raindrop morphology (size) and kinetic energy (velocity).

R2C17: line 160: why is mean intensity used here?

Reply: The average rainfall intensity was used here to compute the average raindrop diameter and finally raindrop momentum on event base. The 10-min maximum raindrop momentum (F10, mg·m·s−1) and the average raindrop momentum at the first and last 10 min (Fb10 and Fe10, respectively, mg·m·s−1) could be calculated with I10, Ib10 and Ie10 as indicated at Equation 5–9 (Line 184–188, Page 9), respectively.

R2C18: line 168: since this paper reports a study of branch stemflow only, the title of the paper should be amended to indicate this clearly (i.e., not a study of stemflow on an entire plant)

Reply: Done. We have revised the title to "Temporal-dependent effects of rainfall characteristics on inter-/intra-event branch-scaled stemflow variability in two xerophytic shrubs" as suggested as Reviewer 3.

R2C19: line 171: to what extent were the studied shrubs representative of the wider population? please present some data.

Reply: C. korshinskii and S. psammophila were the dominant shrub species at the arid and semi-arid regions of northwestern China, including Inner Mongolia Autonomous Region, Ningxia Hui Autonomous Region, Xinjiang Uygur Autonomous Region, Qing-hai province, Gansu province, Shaanxi province, Shanxi province (Chao and Gong, 1999). Since both species had good drought tolerance, they were commonly planted for soil and water conservation, sand fixation and wind barrier (Li, 2012; Hu et al., 2016; Liu et al., 2016; Zhang et al., 2018). As the typical xerophytic shrub species at this region, they had extensive distributions particularly in arid and desert steppes (Li et al., 2016) at Lines 129–132, Page 6. Besides, please see Point (3) at Reply to R2C3 for explaining the representativeness of the selected 4 experimental shrubs for the C. korshinskii and S. psammophila plots.

R2C20: lie 181: please explain what is meant by 'canopy skirt locations'. The photos

suggest that there were many overhanging leaves and branches. Some of the stemflow collars were placed quite high off the ground (as far as can be judged from the photos, as no quantitative information on this is included in the paper). How do the authors know that the stemflow at these heights would actually reach the ground, and not drip off the branches?

Reply: The "canopy-skirt locations" has been revised to "the outlayer-of-canopy" at Lines 210ïijŇ Page 10. The photo shot the lower part of branches to show foil collar and TBRG for stemflow trapping and recording, which might not provide a very clear view of leaves on the upper branches. In contrast to the centered branches, stemflow of branches at the outlayer got less influences from the neighboring ones. We automatically recorded stemflow volume and timing via the RG3-M TBRG with height of 25.7 cm. Therefore, the foil collars were installed at branches nearly 40 cm off the ground (Lines 223–224, Page 11). It might be the minimum height for foil collars so as to keep the hose straight, which channelled stemflow down to TBRGs. The lost by dripping off was believed to be acceptable, compared with the commonly-used method to trap stemflow at breast height (1.2 or 1.3 m off ground) at tress particularly at rainforest, where the stemflow volume was much larger.

R2C21: line 189-190: what was the external diameter? this should be included as the dimensions of the stemflow collars are critical - it does not seem sufficient simply to assert that they caught no rainfall or released drips of throughfall from above.

Reply: The "external diameter" has been revised to "orifice diameter" at Line 234. The limited orifice diameter of foil collars minimized the accessing of throughfall and rains into them (Yuan et al., 2017) (Lines 225–227, Page 11).

R2C22: line 270: how were rainfall intensity peaks identified? What makes one peak an intensity peak?

Reply: SFIi, the instantaneous stemflow intensity, was computed in terms of the tip volume (3.73 mL), branch basal area (mm2) and time intervals between neighboring

tips recorded by TBRGs as indicated Equation 13 (Line 248, Page 12). The largest SFIi was defined as the peak intensity at the incident rains.

R2C23: line 292: is the reference to the volume from a single branch or the total from the 7 branches?

Reply: We focused on the average stemflow variables of 7 experimental branches, and analyzed the most influential rainfall characteristics affecting them. Please see the detailed explanation at Point 2 of Reply to R2C1 and Point 3 of Reply to R2C2.

R2C24: lines 300-310: this is difficult to read, owing to the need to recall the meaning of the very many contractions. Some reminders of what these mean would be useful here.

Reply: As indicated at the suggestion commenting at Line 70 of R2C5, the contraction was only explained when it was first used. For an easy reading, the list of symbols had been prepared as appendix at the revised manuscript (Lines 592–593, Pages 27–29).

R2C25: line 342: a stemflow intensity of 1232 mm h-1 is large. What was the flux? I presume that in the case of the authors own work in the present study, the flux was within the capacity of the tipping-bucket gauges (typically a few hundred mm h-1 at maximum) since the rainfall was not very intense. Some comment on this would be worthwhile.

Reply: As indicated at the manual of RG3-M TBRG (https://www.onsetcomp.com/products/data-loggers/rg3-m), data could be automatically recorded at rains with the maximum intensity of 127 mm·h−1. The unit depth (mm) of inflow water recorded by TBRG was interpreted to the equivalent 1000 cm3 water on the 1-m2 ground surface. However, stemflow intensity was computed with branch basal areas. It approximately ranged in 34–770 mm2 for C. korshinskii and S. psammophila in this study, which took less than 0.8‰ of 1 m2. Therefore, it could be estimated that the RG3-M TBRG offers the ability to record stemflow with the

maximum intensity greater than 15000 mm·h–1.

R2C26: lines 383-384: but these fluxes would surely depend on the antecedent leaf and branch wetness, and on meteorological conditions such as wind speed and vapour deficit (the latter is not reported, incidentally).

Reply: Thank you for this comment. The evaporation coefficient (E, unitless) had been included at the revised manuscript. E was computed with air temperature, relative humidity and wind speed as indicated at Equation 1–3 (Lines 158–160, Page 8). It represented the comprehensive influences of these meteorological characteristics. By performing the multiple correspondence analysis (MCA), E and rainfall duration (RD) were tested to closely relate with stemflow duration (Lines 360–362, Page 17). However, the stepwise regression analysis finally confirmed the dominant influence of RD affecting SFD (Lines 381–382, Page 18). Rainfall intervals, the time intervals between neighboring rains (RI, h), was applied to indirectly represent the branch wettability. Please see the detailed description at Point (3) at Reply to R2C3.

R2C27: Table 2: why are only 3 rainfall events listed here? More than 40 more are simply lumped under "others" and no details are provided. Why?

Reply: Event A, B and C represented three categories of events with the single, double and multiple intensity peak amounts. It had been described at the note of Table 2 (Lines 808–816, Page 40) and Section 3.1 (Lines 301–303, Pages 14). There were 17, 11 and 15 events at Event A, B and C, respectively. Because the remaining 11 events had the average RA of 0.6 mm, no more than three recordings had been observed within event which was limited by 0.2-mm resolution of TBRGs. Therefore, they could not be categorized and grouped as Event others (Lines 303– 06, Page 14).

R2C28: Figure 4 shows units of m/h which I presume should be mm/h

Reply: Done.

Reference: Allaby, M.: A Dictionary of Ecology, 4th Edition., Oxford University Press,

Oxford, 2010.

Cayuela, C., Llorens, P., Sánchez-Costa, E., Levia, D.F. and Latron, J.: Effect of biotic and abiotic factors on inter- and intra-event variability in stemflow rates in oak and pine stands in a Mediterranean mountain area, J. Hydrol., 560, 396–406, https://doi.org/10.1016/j.jhydrol.2018.03.050, 2018.

Chao, P. N. and Gong, G. T.: Salix (Salicaceae), in: Flora of China, edited by: Wu, Z. Y., Raven, P. H., and Hong, D. Y., Science Press, Beijing and Missouri Botanical Garden Press, St. Louis, 162–274, 1999.

Dunkerley, D.: Stemflow on the woody parts of plants: dependence on rainfall intensity and event profile from laboratory simulations, Hydrol. Process., 28, 5469–5482, http://dx.doi.org/10.1002/hyp.10050, 2014a.

Dunkerley, D.: Stemflow production and intrastorm rainfall intensity variation: an experimental analysis using laboratory rainfall simulation, Earth Surf. Proc. Land., 39, 1741–1752, http://dx.doi.org/10.1002/esp.3555, 2014b.

[revised manuscript text omitted]

Zhang, Y.F., Wang, X.P., Hu, R. and Pan, Y.X.: Meteorological influences on process-based spatial-temporal pattern of throughfall of a xerophytic shrub in arid lands of northern China, Sci. Total. Environ., 619, 1003–1013, https://doi.org/10.1016/j.scitotenv.2017.11.207, 2018.

Please also note the supplement to this comment:
https://www.hydrol-earth-syst-sci-discuss.net/hess-2019-254/hess-2019-254-AC3-supplement.pdf

[Figure]

**Fig. 1.** Fig. R2-1. The established runoff and sediment plots at the S. psammophila plot.

[Figure]

**中国科学院生态环境研究中心**
**Research Center for Eco-Environmental Science**
**Chinese Academy of Sciences (CAS)**
北京市海淀区双清路 18 号 邮编：100085  Website: www.rcees.ac.cn
**18 Shuangqing Road, Haidian District, Beijing 100085, P. R. China**
Tel: 0086-10-62911239   Email: gygao@rcees.ac.cn

**Table R2-1.** Branch morphological features of the experimental shrubs of *C. korshinskii* and *S. psammophila*.

| BD categories | *C. korshinskii* | | | | *S. psammophila* | | | |
|---|---|---|---|---|---|---|---|---|
| | BD (mm) | BL (cm) | BA (°) | BN | BD (mm) | BL (cm) | BA (°) | BN |
| ≤5 | 4.1 | 90.4 | 64.1 | 40 | 4.8 | 166 | 66 | 2 |
| 5–10 | 7.3 | 124.9 | 61.8 | 82 | 8.0 | 204 | 64 | 53 |
| 10–15 | 12.5 | 161.1 | 51.7 | 36 | 12.9 | 253 | 58 | 82 |
| 15–18 | 16.3 | 170.6 | 48.7 | 13 | 16.5 | 280 | 52 | 56 |
| 18–25 | 19.3 | 192.3 | 51.3 | 9 | 20.3 | 302 | 50 | 59 |
| >25 | NA | NA | NA | NA | 28.7 | 366 | 50 | 9 |

Note: BD, BL, BA and BN are the basal diameter, length, angle and number of branches.

**Fig. 2.** Table R2-1. Branch morphological features of the experimental shrubs of C. korshinskii and S. psammophila.

[Figure]

**中国科学院生态环境研究中心**
**Research Center for Eco-Environmental Science**
**Chinese Academy of Sciences (CAS)**
北京市海淀区双清路 18 号 邮编：100085  Website: www.rcees.ac.cn
**18 Shuangqing Road, Haidian District, Beijing 100085, P. R. China**
Tel: 0086-10-62911239  Email: gygao@rcees.ac.cn

**Table R2-2.** Investigation of canopy morphology at *C. korshinskii* plot.

| Plots | Shrubs | Canopy heights (m) | Canopy area (m$^2$) |
|---|---|---|---|
| A | 1 | 1.7 | 4.6 |
| | 2 | 1.2 | 2.1 |
| | 3 | 1.9 | 3.7 |
| | 4 | 1.4 | 2.5 |
| | 5 | 2.0 | 5.7 |
| B | 6 | 1.7 | 5.5 |
| | 7 | 1.8 | 4.3 |
| | 8 | 1.8 | 3.8 |
| | 9 | 2.1 | 6.8 |
| | 10 | 2.5 | 11.6 |
| | 11 | 2.3 | 6.7 |
| C | 12 | 1.3 | 3.4 |
| | 13 | 1.9 | 5.9 |
| | 14 | 1.9 | 2.7 |
| | 15 | 1.8 | 2.8 |
| | 16 | 2.0 | 4.0 |
| | 17 | 2.2 | 5.5 |
| Average | | 1.9±0.1 | 4.8±0.6 |

**Fig. 3.** Table R2-2. Investigation of canopy morphology at C. korshinskii plot.

[Figure]

中国科学院生态环境研究中心
**Research Center for Eco-Environmental Science**
**Chinese Academy of Sciences (CAS)**
北京市海淀区双清路 18 号 邮编：100085  Website: www.rcees.ac.cn
**18 Shuangqing Road, Haidian District, Beijing 100085, P. R. China**
Tel: 0086-10-62911239  Email: gygao@rcees.ac.cn

**Table R2-3.** Investigation of canopy morphology at *S. psammophila* plot.

| Shrubs | Canopy heights (m) | Canopy area (m$^2$) |
|---|---|---|
| 1 | 3.8 | 24.0 |
| 2 | 3.8 | 18.5 |
| 3 | 3.6 | 21.8 |
| 4 | 3.7 | 24.0 |
| 5 | 3.2 | 20.6 |
| 6 | 2.6 | 13.2 |
| 7 | 2.9 | 5.8 |
| 8 | 3.3 | 25.9 |
| 9 | 3.2 | 8.3 |
| 10 | 4.4 | 22.5 |
| 11 | 4.4 | 29.7 |
| 12 | 2.9 | 7.4 |
| 13 | 3.8 | 25.7 |
| Average | 3.5±0.2 | 19.1±2.2 |

**Fig. 4.** Table R2-3. Investigation of canopy morphology at S. psammophila plot.